# Nucleation-promoting and growth-limiting synthesis of disordered rock-salt Li-ion cathode materials

Hoda Ahmed [1], Moohyun Woo [1], Nicolas Dumaresq [1], Pablo Trevino Lara [1], Richie Fong [1], Sang-Jun Lee [2], Gregory Lazaris [1], Nauman Mubarak [1], Nicolas Brodusch [1], Dong-Hwa Seo [3], Raynald Gauvin [1], George P. Demopoulos [1] & Jinhyuk Lee [1] ✉

Disordered rock-salt oxides and oxyfluorides are promising positive electrode materials for high-performance lithium-ion batteries free of nickel and cobalt. However, conventional synthesis methods rely on post-synthesis pulverization to achieve cycling-appropriate particle sizes, offering limited control over particle microstructure and crystallinity. This accelerates degradation and complicates secondary particle processing. Here we present a synthesis strategy that enhances nucleation while suppressing particle growth and agglomeration across various disordered rock-salt compositions, including lithium–manganese–titanium oxide, lithium–manganese–niobium oxide, and lithium–nickel–titanium oxide systems. Applied to $Li_{1.2}Mn_{0.4}Ti_{0.4}O_2$, this method yields highly crystalline, well-dispersed sub-200 nm particles that form homogeneous electrode films with stable cycling behavior. Tested in cells with lithium metal as the counter electrode, these electrodes deliver ~200 mAh/g with 85% capacity retention relative to the first cycle after 100 cycles (20 mA/g, 1.5–4.8 V), and an average discharge voltage loss of 4.8 mV per cycle, compared to 38.6% retention and 7.5 mV loss per cycle for electrodes derived from pulverized solid-state particles. This approach suggests a route to enhance the performance and durability of disordered rock-salt electrodes for sustainable lithium-ion batteries.

Decades of advancements in lithium-ion batteries (LIBs) have propelled the global transition toward electric vehicles and grid energy storage systems, which complement intermittent renewable power generation efforts[1]. This surge in LIB adoption has led to the increase in cell prices in recent years, marking a reversal of the decade-long trend of declining prices due to strains on key battery inputs[2]. To address the escalating demand, forecasted to reach TWh-scale levels in the immediate future and beyond, LIBs must enhance their performance while simultaneously becoming more cost-effective. This necessity is underscored by prominent international battery strategies that highlight the urgent need for alternatives to nickel (Ni) and cobalt (Co) in commercial LIB cathodes (positive electrodes), such as layered oxide $LiNi_{0.6}Mn_{0.2}Co_{0.2}O_2$[3].

Disordered rock-salt Li-excess cathode materials (DRXs), especially those rich in Mn as the redox-active transition metal (Mn-DRXs), such as $Li_{1.2}Mn_{0.4}Ti_{0.4}O_2$ and $Li_{1.68}Mn_{1.60}O_{3.7}F_{0.3}$, represent one of the few Co/Ni-free cathode materials to reshape the current market landscape[4–13]. Consequently, there has been a surge of

[1]Department of Mining and Materials Engineering, McGill University, Montréal, QC, Canada. [2]Stanford Synchrotron Radiation Lightsource, SLAC National Accelerator Laboratory, Menlo Park, CA, USA. [3]Department of Materials Science and Engineering, Korea Advanced Institute of Science and Technology (KAIST), Daejeon, Republic of Korea. ✉e-mail: jinhyuk.lee@mcgill.ca

interest in these Mn-DRXs, driven by the quest for compositions capable of delivering both stable cycling performance and improved energy density. Various strategies have emerged in this pursuit, including fluorination[14,15], metal substitution[16–18], phosphate doping[19], and electrolyte engineering[20,21].

Despite considerable efforts directed towards these materials, progress in controlling their particle microstructure has been minimal, largely hindered by the limited synthesis methods available. This problem is significant for Mn-DRXs because they need to have a small enough particle size (<200 nm) to achieve high capacity to overcome their limited intrinsic Li diffusivity ($10^{-16}$ to $10^{-14}$ cm$^2$/s), but the current synthesis methods produce nanoparticles with low crystallinity, barely controlled particle sizes, and particle agglomeration, accelerating electrode degradation and electrolyte decomposition[5].

Currently, Mn-DRX compounds are primarily synthesized using the solid-state synthesis method or mechanochemistry. Solid-state synthesis involves the mixing of precursors (e.g., $Li_2CO_3$, $Mn_2O_3$, $TiO_2$, $Nb_2O_5$, LiF) followed by calcination at high temperatures typically exceeding 900 °C[5]. This high-temperature calcination of the agglomerated precursor mixture results in several-micrometer-sized particles with uncontrolled necking, necessitating aggressive post-synthesis particle pulverization (such as via ball milling) before using them as cathode active materials. Mechanochemical synthesis involves precursor mixing and chemical synthesis in a single step induced by mechanical ball milling[5]. This method is often utilized to produce metastable Mn-DRXs, including highly fluorinated compounds[17,18], nanocomposite spinel-DRX compounds[8,22], or polyanion-doped DRX compounds[19,23]. Unfortunately, mechanochemistry, being inherently a ball-milling method, results in secondary particles with low crystallinity without control over their size and shape[5].

Meanwhile, Mn-DRXs have also been synthesized using molten-salt[24–26], sol-gel[27], and microwave heating methods[28] in recent studies. However, while all these methods could produce phase-pure Mn-DRXs, they also fell short in producing morphology-controlled sub-200 nm particles, requiring post-synthesis particle pulverization to cycle the compounds. In summary, the majority of the reported methods thus far have relied on aggressive post-synthesis particle pulverization for cycling, leading to a loss of control over particle morphology and size, and often introducing crystal defects[5].

Knowing that the success of any battery material relies on a synthesis method that can control the particle size and morphology while maintaining the particle crystallinity, such as the co-precipitation method for the precursors of layered Li-Ni-Mn-Co oxides and their derivatives[29,30], the absence of synthesis methods capable of doing so for Mn-DRX presents a significant drawback. This limitation hampers the uniform coating of particles and the even distribution of active materials within the cathode film for enhanced capacity retention, thereby impeding the practical advancement of Ni- and Co-free LIBs based on the Mn-DRX cathode[5].

In this article, we present the direct synthesis of highly crystalline Mn-DRX particles of $Li_{1.2}Mn_{0.4}Ti_{0.4}O_2$ (LMTO), the representative Mn-DRX compound, with an average primary particle size of less than 200 nm and suppressed particle agglomeration. Additionally, we demonstrate the versatility of this method by applying it to other DRX compositions: $Li_{1.1}Mn_{0.7}Ti_{0.2}O_2$, $Li_{1.2}Mn_{0.6}Nb_{0.2}O_2$, and $Li_{1.2}Ni_{0.2}Ti_{0.6}O_2$. This achievement was accomplished through a nucleation-promoting and growth-limiting molten-salt synthesis method (NM synthesis). We provide fundamental insights into how this method controls the particle size of LMTO (NM-LMTO), the effects of annealing and washing protocols, and how these sub-200 nm, highly crystalline single particles significantly enhance cycling stability compared to LMTO prepared via the solid-state method followed by particle pulverization (85% vs. 38.6% capacity retention over 100 cycles; 4.8 mV vs. 7.5 mV voltage loss per cycle in Li||LMTO cells). This improvement is facilitated by the uniform distribution of LMTO particles within the cathode

film. Overall, the results and insights provided by our study regarding a particle-morphology-controllable synthesis method herald an exciting direction for the research and development of DRXs for advanced Ni/Co-free LIBs.

## Results and Discussion
### Introduction of the NM method

We chose LMTO as the representative Mn-DRX to demonstrate our NM synthesis, a modified molten-salt synthesis method. For comparison, we also prepared LMTO synthesized through the solid-state method (S-LMTO) and its pulverized counterpart (PS-LMTO).

Molten-salt syntheses of Mn-DRXs were previously demonstrated using the same metal precursors (e.g., $Li_2CO_3$, $Mn_2O_3$, $TiO_2$, $Nb_2O_5$) used in the solid-state method while additionally incorporating a non-reactive salt, such as KCl, which has a lower melting point (770 °C for KCl) than the calcination temperature of Mn-DRX (e.g., 950 °C for 12 h) (Fig. 1a). During calcination, this molten salt serves as a solvent to facilitate nucleation and growth of Mn-DRX particles, while the corrosive nature of the molten-salt reduces the particle agglomeration[24–26,31], suggesting the opportunity to produce size- and morphology-controlled Mn-DRX particles with reduced agglomeration.

However, previous demonstrations of the molten-salt synthesis of Mn-DRXs resulted in large particles ranging from 5 to 20 μm[24–26], which are hardly cyclable due to slow Li diffusion in the Mn-DRX structure[32]. Thus, despite suppressed particle agglomeration and uniform shape, those particles required aggressive post-synthesis pulverization for electrochemical testing, making previous attempts irrelevant for producing directly cyclable small Mn-DRX particles using the molten-salt method. Forming large Mn-DRX particles via the molten-salt method is not surprising considering that high calcination temperature is generally necessary for Mn-DRXs to be thermodynamically stabilized against lower-temperature competing phases (e.g., orthorhombic $LiMnO_2$)[33]. At high temperatures, molten salts significantly promote particle growth.

Nevertheless, since the molten salt can enhance nucleation kinetics through a solvent-mediated reaction[31], we hypothesized that with brief heating time, it would be possible to nucleate Mn-DRXs without significant particle growth even at high temperatures. Subsequently, any incomplete reactions can be resolved during a second annealing step at a lower temperature, which serves to limit particle growth while improving the Mn-DRX crystallinity.

Accordingly, we developed a modified molten-salt method (NM synthesis) (Fig. 1a), employing $Li_2CO_3$, $Mn_2O_3$, and $TiO_2$ as LMTO precursors and CsBr as the molten-salt flux. CsBr was selected primarily for the following reasons: First, CsBr has a melting point of 636 °C, which is lower than KCl (770 °C) used by Chen et al. and KBr (734 °C), allowing for sufficiently low-temperature molten-salt calcination of LMTO in the first stage[24–26]. Secondly, the NM method includes an annealing step on the LMTO/salt mixture at a temperature below the melting point of the salt. For this step, we require a salt with a reasonably high melting point to enable annealing at a high enough temperature, which CsBr provides. If the salt flux melts during the annealing step, it would effectively become an extended molten-salt synthesis, leading to significant particle growth. In the Supplementary Information (SI), we present the NM synthesis of LMTO using various salts (KCl, KBr, KI, CsCl, CsBr, and CsI). We show that Cs-based salts yield higher LMTO purity than K-based salts under the same heating protocol (Fig. S1). This improvement is attributed to the lower melting points and higher dielectric constants of Cs salts, which enhance ion solvation and improve the solubility of precursors or intermediates during molten-salt synthesis, thereby promoting more homogeneous reactant distribution and reducing impurity formation[31].

With CsBr as the molten salt source, the first step of the NM synthesis involves ramping up the furnace temperature (e.g., 1 °C/s) to

a predetermined temperature (*e.g.*, 800 °C, 900 °C), high enough to melt CsBr and to drive the LMTO nucleation. For the results presented in the main body of this manuscript, we maintained a precursor-to-salt ratio of 1:1.5 by weight (Fig. S2), as it leads to the purest LMTO phase under the optimized NM heating protocol for LMTO, to be discussed in later sections. A comprehensive study on the effect of this ratio on NM synthesis, including its influence on the degree of supersaturation of precursor reactants in the molten salt, will be presented in a separate publication. Calcination is then applied to the mixture of CsBr and LMTO precursors at the set temperature for a desired duration (*e.g.*, 5 min, 20 min), followed by furnace cooling to room temperature. In the second step, the mixture of the salt flux and LMTO is annealed at a temperature (*e.g.*, 600 °C) below the melting point of the molten salt flux for a specified duration (*e.g.*, 2 h, 10 h, 20 h). During this stage, the solid CsBr in the mixture serves as a barrier to prevent LMTO particle agglomeration, while the annealing process completes the LMTO formation and enhances crystallinity. After this, the mixture is washed with deionized water (DIW) to remove the CsBr. A subsequent Li-reinsertion step is performed to restore the Li that was extracted from the LMTO particles through $Li^+/H^+$ exchange during the DIW washing, followed by filtration and drying of the powder in a vacuum oven. The

significance of these steps will be further discussed in the following paragraphs.

## Direct synthesis of size-controlled and dispersed DRX single particles

Figure 1b–e compare the crystallinity and microstructure of LMTO particles synthesized using the typical solid-state method followed by post-synthesis pulverization and our NM method. Figure 1b displays the X-ray diffraction (XRD) patterns of LMTO after the solid-state method (S-LMTO, 1000 °C for 2 h under Ar) and after particle pulverization (PS-LMTO) via a planetary ball mill (Fig. S3). Both samples exhibit characteristic XRD patterns of DRXs, yet PS-LMTO demonstrates significantly broadened XRD peaks, indicative of diminished crystallinity due to particle pulverization. Similar to the S-LMTO, XRD patterns of NM-LMTO synthesized using different initial molten-salt calcination conditions (*e.g.*, 800 °C for 5 or 20 min, 900 °C for 5 or 20 min) exhibit the DRX phases with sharply defined peaks, indicating high crystallinity (Figs. 1c, S4). It is noteworthy that these NM-LMTO samples underwent the same annealing step at 600 °C for 20 h with the LMTO precursor-to-CsBr weight ratio of 1:1.5, followed by the same washing and Li-reinsertion steps to be discussed later.

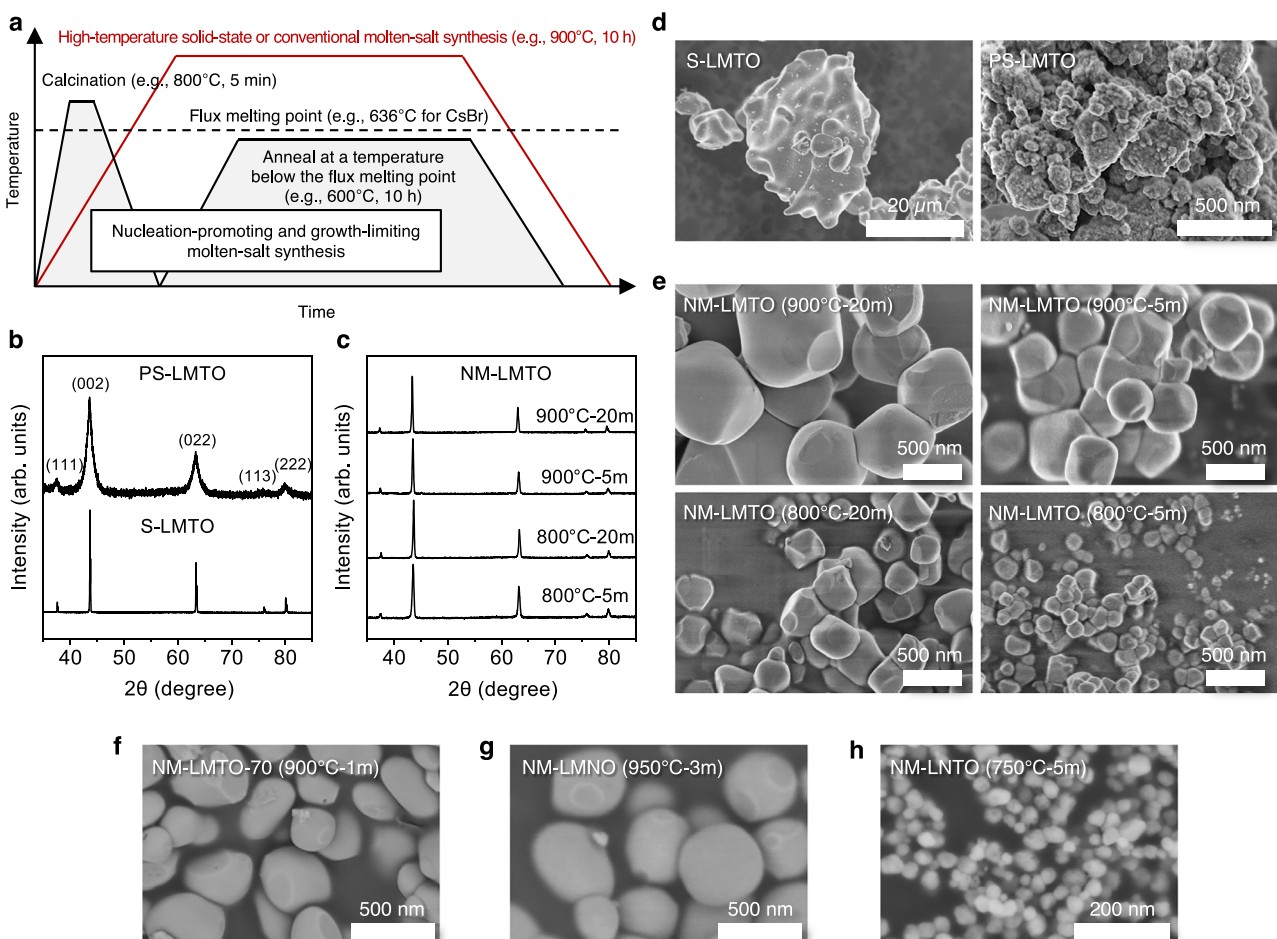

**Fig. 1 | Nucleation-promoting and growth-limiting molten-salt synthesis method (NM synthesis). a** The schematics of the heat treatment profile of the NM synthesis vs. traditional solid-state or molten-salt synthesis. **b** XRD patterns of the as-synthesized $Li_{1.2}Mn_{0.4}Ti_{0.4}O_2$ (LMTO) made through a solid-state method (S-LMTO) and after particle pulverization (PS-LMTO). **c** XRD patterns of the LMTO synthesized via the NM method (NM-LMTO) with different initial calcination temperatures and times (900 °C-20 min, 900 °C-5 min, 800 °C-20 min, and 800 °C-5 min, precursor-to-salt ratio of 1:1.5 by weight). These NM-LMTO particles

underwent the annealing procedure at 600 °C for 20 h under argon after the initial calcination step. **d**–**h** The SEM images of the **d** S-LMTO and PS-LMTO, **e** NM-LMTO with different initial calcination steps, **f** NM-$Li_{1.1}Mn_{0.7}Ti_{0.2}O_2$ (NM-LMTO-70, 900 °C-1 min calcination, 600 °C-5 h annealing under argon), **g** NM-$Li_{1.2}Mn_{0.6}Nb_{0.2}O_2$ (NM-LMNO, 950 °C-3 min calcination, 600 °C-5 h annealing under argon), and (**h**) NM-$Li_{1.2}Ni_{0.2}Ti_{0.6}O_2$ (NM-LNTO, 750 °C-5 min calcination, 600 °C-5 h annealing in air). Source data for **b**, **c** are provided as a Source Data file.

The scanning electron microscope (SEM) images presented in Fig. 1d, e offer a visual insight into how different synthesis and processing methods impact the microstructure of the LMTO particles. When employing the typical solid-state synthesis method, S-LMTO particles exhibit significant agglomeration, forming large particles with a diameter exceeding 10 μm (Fig. 1d). Subsequently, particle pulverization yields PS-LMTO particles with agglomerated polycrystalline structures, composed of sub-50 nm primary particles and exhibiting a broad distribution of secondary particle sizes. These microstructure features are consistent with findings reported in the Mn-DRX literature[5]. Furthermore, the nanosizing of particles contributes to XRD peak broadening, aligning with the broadening observed in Fig. 1b.

On the other hand, the SEM images of NM-LMTO in Fig. 1e illustrate the effectiveness of our NM method in producing LMTO single particles with controlled sizes by adjusting the initial calcination temperature and duration. Transitioning from 900 °C for 20 min to 900 °C for 5 min, 800 °C for 20 min, and 800 °C for 5 min results in variations in primary particle size. After 20 min of calcination at 900 °C, the average primary particle size is ~701 nm (standard deviation ~268 nm, Fig. S5), while after 5 min at 900 °C, it decreases to ~429 nm (standard deviation ~250 nm, Fig. S5). These outcomes underscore the rapid growth of LMTO particles during high-temperature molten-salt calcination. However, for NM-LMTO particles calcined at 800 °C for 20 min, their average size decreases to ~266 nm (standard deviation ~36 nm, Fig. S5). Subsequently, the NM-LMTO particles produced at 800 °C for 5 min exhibit primary particle sizes below 200 nm (average primary particle size ~118 nm, standard deviation ~35 nm, Fig. S5). This result demonstrates direct synthesis of LMTO particles with sub-200 nm primary particle size with high crystallinity, as evidenced by the sharp XRD peaks in Fig. 1c. Furthermore, since the NM method relies on molten-salt synthesis, particle agglomeration can be suppressed by the corrosive nature of the molten salt, which etches nascent necks and thus limits their consolidation. However, the second annealing step at 600 °C for 20 h (to be discussed below) results in some remaining agglomeration.

This NM method demonstrates broad applicability for synthesizing small single DRX particles, including $Li_{1.1}Mn_{0.7}Ti_{0.2}O_2$ (LMTO-70), $Li_{1.2}Mn_{0.6}Nb_{0.2}O_2$ (LMNO), and $Li_{1.2}Ni_{0.2}Ti_{0.6}O_2$ (LNTO), as shown in the SEM images in Fig. 1f, g, h. NM-LMTO-70, NM-LMNO, and NM-LNTO were calcined at 900 °C for 1 min under argon, 950 °C for 3 min under argon, and 750 °C for 5 min in air, respectively, followed by annealing at 600 °C for 5 h under argon for NM-LMTO-70 and NM-LMNO and in air for NM-LNTO. Detailed synthesis procedures are provided in the Methods section. All samples successfully formed DRX phases with target compositions (Fig. S6).

NM-LMTO-70 (average primary particle size ~308 nm, standard deviation ~119 nm, Fig. S7) and NM-LMNO (average primary particle size ~420 nm, standard deviation ~179 nm, Fig. S7) exhibit larger particle sizes compared to NM-LMTO, attributed to their higher calcination temperatures required for pure-phase production. Despite this, these particles remain among the smallest single particles reported for each composition, significantly smaller than the >10 μm particle sizes of as-synthesized LMTO-70 and LMNO reported in the literature[32,34]. Notably, unlike highly agglomerated polycrystalline LNTO particles reported in the literature[35], NM-LNTO exhibits small and well-dispersed single particles (average primary particle size ~36 nm, standard deviation ~6 nm, Fig. S7), underscoring the effectiveness of the NM method in producing small DRX single particles.

## Effects of annealing, washing, and Li-reinsertion steps in the NM method

Following the demonstration of the NM method's ability to directly synthesize size-controlled, highly crystalline DRX single particles, we now elucidate the importance of the annealing, washing, and Li-reinsertion steps in ensuring that the produced material achieves high capacity.

While short molten-salt calcination at a relatively low temperature of 800 °C can produce sub-200 nm LMTO primary particles in the first step of the NM method, we observe compositional heterogeneity in the particle after the initial step due to the limited reaction time. Figure 2a illustrates the scanning transmission electron microscopy (STEM) high-angle annular dark-field (HAADF) images and the Electron Energy Loss Spectroscopy (STEM-EELS) mapping of Mn, Ti, and O from the cross-section of an NM-LMTO particle before and after annealing under argon at 600 °C for 10 h. The initial calcination condition for this sample was 830 °C for 7 min. The cross-sectional EELS mapping reveals that the NM-LMTO particle without (before) the second annealing develops a core-shell structure with a Ti-rich surface and Mn-rich core. However, this compositional inhomogeneity is alleviated after annealing at 600 °C for 10 h, as observed by the smoothed-out Mn and Ti signals from the mapping. Moreover, the XRD peaks of NM-LMTO (initially calcined at 800 °C for 5 min) become sharper, with each peak shifting to a higher angle after annealing at 600 °C for 2, 10, and 20 h. This shift indicates a reduction in lattice parameters, likely associated with improved compositional homogeneity (Fig. 2b). Eventually, the lattice parameter of the NM-LMTO decreases to ~4.1517 Å after 20 h of annealing (Figs. S8, S9), aligning with the typical lattice parameter of ~4.15 Å reported for LMTO in the literature[36].

We believe that the observed compositional inhomogeneity, as well as its removal upon annealing, is linked to the synthesis pathway of LMTO. Specifically, the Mn-rich orthorhombic $LiMnO_2$-like phase serves as an intermediate that reacts with the Ti-rich rock-salt phase to form LMTO (Fig. S2), a process that requires metal diffusion between these phases. In this context, STEM-EELS likely captures incomplete metal diffusion in certain particles due to short calcination times. Subsequent annealing facilitates the completion of the reaction between the two phases, resolving the inhomogeneity.

Note that for this second-step annealing, we anneal the LMTO-CsBr mixture together at a temperature (e.g., 600 °C) below the melting point of CsBr (636 °C). If CsBr melts again during the (second) annealing stage, it will result in an extended molten-salt reaction, significantly increasing the LMTO particle size, as can be inferred from the rapid particle growth seen in Fig. 1e. Also, by annealing the entire LMTO-CsBr mixture, we can give more time for any unreacted LMTO precursors in the LMTO-CsBr mixture to complete the LMTO formation. Finally, the solidified CsBr in the LMTO-CsBr mixture can limit the direct contact between LMTO particles to slow down LMTO particle agglomeration, compared to when CsBr is washed before annealing. Overall, this second-step annealing protocol is essential for decoupling the LMTO's crystallinity improvement from particle growth. However, the extended annealing time necessarily increases particle agglomeration, as seen in the right-shifted particle size distribution (measured with a Zetasizer) after annealing at 600 °C for 10 h or 20 h (Fig. 2c) and the more agglomerated particles observed in the SEM image after annealing at 600 °C for 10 h (Fig. 2d). It is important to note that despite the slight particle agglomeration, the primary particle size after annealing does not increase a lot, remaining sub-200 nm.

The significance of the washing and Li-reinsertion steps in the NM method is now discussed. Following the second annealing step, it is necessary to remove the molten-salt flux precursor, such as CsBr in our case, from the salt-DRX mixture. These salts are typically chosen to be highly soluble in water, facilitating their removal through washing with water. For example, Chen et al. utilized (neutral) DIW to remove the salt flux after synthesizing Mn-DRXs via molten-salt synthesis[24–26]. However, DIW washing can significantly impair the resulting Mn-DRX particles. This is primarily attributed to the $Li^+/H^+$ exchange between Mn-DRX particles and DIW, represented as $Li^+(DRX) + H^+(aq) \rightarrow H^+(DRX) + Li^+(aq)$. This process can lead to the extraction of $Li^+$ from

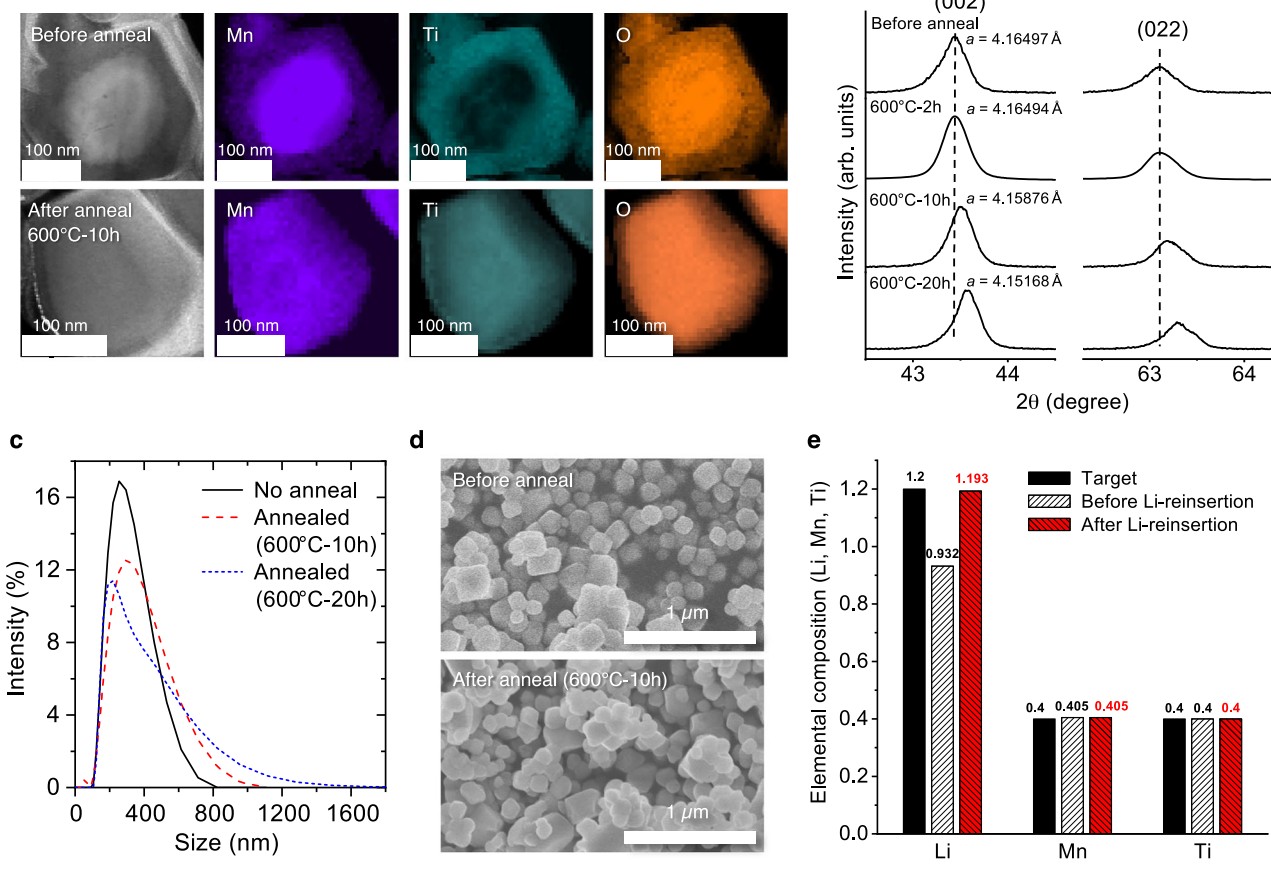

**Fig. 2 | The necessity of annealing and Li-reinsertion after washing of NM-LMTO in decoupling the crystallinity and the particle size. a** STEM HAADF images and STEM-EELS mapping based on Mn $L_{3,2}$, Ti $L_{3,2}$, and O $K$ signal from the cross-section of an NM-LMTO particle (initial calcination at 830 °C for 7 min) before and after annealing at 600 °C for 10 h, with a precursor-to-salt ratio of 1:1.5 by weight. **b** The (002) and (022) XRD peaks of NM-LMTO (initial calcination at 800 °C for 5 min) before and after annealing at 600 °C for 2, 10, and 20 h, with a precursor-to-salt ratio of 1:1.5 by weight. **c** Particle size distribution of NM-LMTO particles (initial calcination at 800 °C for 5 min) before and after annealing at 600 °C for 10 and 20 h, as measured by a particle size analyzer (Zetasizer). **d** SEM images of the NM-LMTO particles (initial calcination at 800 °C for 5 min) before and after annealing at 600 °C for 10 h. **e** Elemental composition (Li, Mn, Ti) of NM-LMTO before and after Li-reinsertion, determined by ICP-OES. Source data for (**b**, **c**, **e**) are provided as a Source Data file.

the Mn-DRX while introducing $H^+$ to the sub-200 nm Mn-DRX particles with exposed particle surfaces.

For example, Fig. 2e illustrates the Li deficiency in NM-LMTO after washing with DIW, as detected by Inductively Coupled Plasma Optical Emission Spectroscopy (ICP-OES). The expected Li:Mn:Ti molar ratio for LMTO ($Li_{1.2}Mn_{0.4}Ti_{0.4}O_2$) is 1.2:0.4:0.4. However, NM-LMTO washed with DIW shows a ratio of 0.932:0.405:0.4, indicating a Li deficiency in the crystal structure. This Li extraction is accompanied by an increase in the pH of the washing solution to ~11.5, due to the consumption of $H^+$ ions in the DIW during the $Li^+/H^+$ exchange, which results in excess $OH^-$ ions in the solution. A titration test of the DIW filtrate after washing confirms that this $Li^+/H^+$ exchange produces LiOH and $Li_2CO_3$ in the solution, the latter being a reaction product of LiOH and dissolved $CO_2$ in DIW (Fig. S10).

It is known that such $Li^+/H^+$ exchange can be inhibited when using a concentrated LiOH solution with a high pH for washing, as the low activity of $H^+$ (high activity of $OH^-$) and high activity of $Li^+$ in the concentrated LiOH solution shift the $Li^+$(cathode) + $H^+$(aq) → $H^+$(cathode) + $Li^+$(aq) reaction to the left[37,38]. However, preparing a large volume of concentrated LiOH solution for thorough CsBr washing requires a significant amount of LiOH to achieve the desired pH that can limit the $Li^+/H^+$ exchange. Therefore, we opted to use the DIW to wash the LMTO-CsBr mixture, allowing the $Li^+/H^+$ exchange to occur initially.

Subsequently, we treated the $Li^+/H^+$ exchanged NM-LMTO with a concentrated LiOH solution, reversing the $Li^+$(cathode) + $H^+$(aq) → $H^+$(cathode) + $Li^+$(aq) reaction to replenish the lost Li from the NM-LMTO, *i.e.*, Li-reinsertion. To be more specific, we immersed and stirred the DIW-washed NM-LMTO particles in a LiOH solution with a pH of ~11.9 for 6 h under an argon flow. The effect of this Li-reinsertion step is evidenced by the recovery of the Li:Mn:Ti ratio in NM-LMTO from 0.932:0.405:0.4 to 1.193:0.405:0.4 after exposing the DIW-washed NM-LMTO to the concentrated LiOH solution (Fig. 2e). In the SI, we also present the ICP-OES results on the NM-LMTO washed with LiOH solutions of various pH values (Fig. S11). These results demonstrate the inhibition of $Li^+/H^+$ exchange through a more traditional washing method, as opposed to the Li-reinsertion method.

## Effects of synthesis parameters on the performance of NM-LMTO

To elucidate the impact of the aforementioned synthesis parameters on NM-LMTO, we present the cycling performance of NM-LMTO in Li||NM-LMTO cells using electrodes composed of 70 wt% NM-LMTO, 20 wt% carbon black (CB), and 10 wt% polyvinylidene fluoride (PVDF) in Fig. 3. Note that the Mn-DRX's performance improves with increased intensity or duration of carbon mixing up to a point where excessive mixing may induce undesirable side reactions[5,39]. This enhancement is

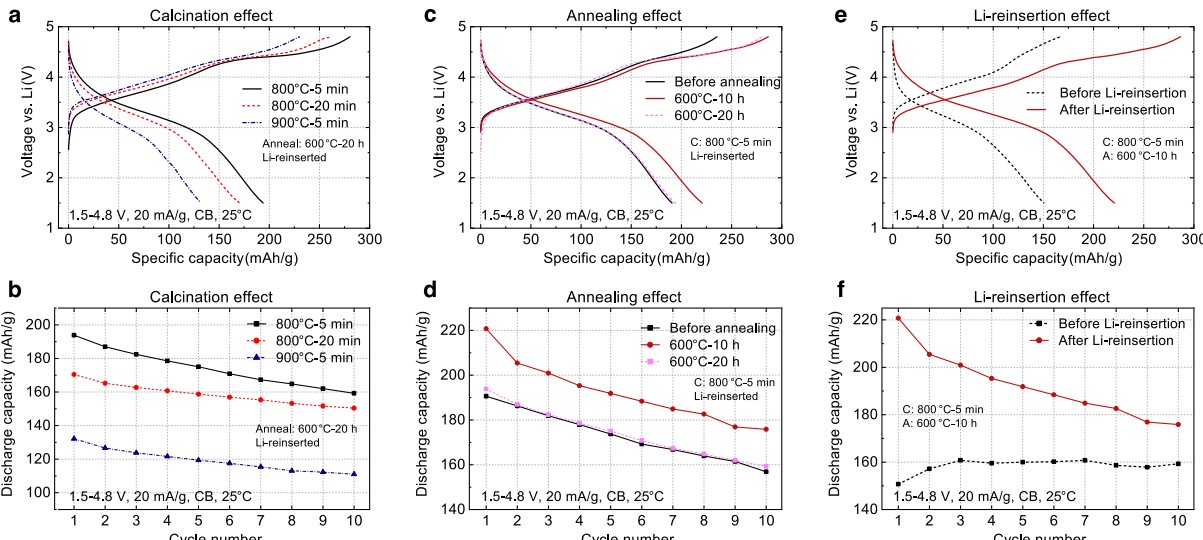

**Fig. 3 | Impact of synthesis parameters on the performance of NM-LMTO in the 70(LMTO):20(CB):10(PVDF) electrode in Li∥NM-LMTO cells. a** Voltage profiles and (**b**) capacity retention of NM-LMTO with varying initial calcination temperatures and times (1.5–4.8 V, 20 mA/g). These samples underwent the same annealing (600 °C for 20 h) and DIW washing followed by the Li-reinsertion step. **c** Voltage profiles and (**d**) capacity retention of NM-LMTO with different second annealing times (0, 10, and 20 h) at 600 °C (1.5–4.8 V, 20 mA/g). These samples underwent the same calcination step (800 °C for 5 min) and DIW washing followed by the Li-reinsertion step. **e** Voltage profiles and (**f**) capacity retention of NM-LMTO before and after Li-reinsertion post DIW washing. The NM-LMTO cycled here underwent calcination at 800 °C for 5 min followed by the annealing at 600 °C for 10 h with a precursor-to-salt weight ratio of 1:1.5. Source data for **a**–**f** are provided as a Source Data file.

attributed to the tighter contact between carbon and Mn-DRX particles (transitioning from CB mixing to mechanical carbon coating), alongside a reduction in Mn-DRX's particle size facilitated by the ball milling process. However, mechanical carbon coating via strong and extended ball milling can damage and pulverize the NM-LMTO particles, undermining the objectives of this study. Therefore, despite the potential cost of sub-optimized capacity and retention of our electrode, we adopted gentle two-hour CB mixing with NM-LMTO and CB for the results in Fig. 3.

Figure 3a, b illustrate the voltage profiles and capacity retention of NM-LMTO in Li∥NM-LMTO cells with varying primary particle sizes achieved by adjusting the initial calcination step (precursors-to-salt weight ratio = 1:1.5): 900 °C for 5 min (average size ~429 nm, standard deviation ~250 nm), 800 °C for 20 min (average size ~266 nm, standard deviation ~36 nm), and 800 °C for 5 min (average size ~118 nm, standard deviation ~35 nm), while keeping the second-step annealing (600 °C for 20 h), washing, and Li-reinsertion steps constant. The SEM images corresponding to these NM-LMTO samples are presented in Fig. 1e. The specific capacity (mAh/g) and cycling rate (mA/g) are reported based on the mass of the active material (LMTO) in the composite electrode. The initial discharge capacity of the 900 °C-5 min NM-LMTO, cycled between 1.5–4.8 V at 20 mA/g and 25 °C, is approximately 133 mAh/g. In comparison, the 800 °C-20 min and 800 °C-5 min variants exhibit initial discharge capacities of roughly 170 mAh/g and 194 mAh/g, respectively, demonstrating an increased capacity with decreasing particle size. This trend is expected as the slow intrinsic Li diffusivity ($10^{-16}$ to $10^{-14}$ cm²/s) in LMTO (or other Mn-DRXs) requires shorter Li travel distances to achieve higher capacity at a given rate of cycling. On the other hand, the capacity retention in this test appears better with the 900 °C-5 min and 800 °C-20 min NM-LMTO than the 800 °C-5 min variants (Fig. 3b). This behavior is most likely because larger LMTO particles make CB percolation and LMTO/CB contact in the electrode easier to maintain during cycling, which tends to degrade easily in the CB electrode due to LMTO's volume change[10].

Figure 3c, d illustrate the influence of the second annealing on the voltage profiles and capacity retention of NM-LMTO in Li∥NM-LMTO

cells with a sub-200 nm primary particle size (1.5–4.8 V, 20 mA/g, 25 °C), prepared by calcinating at 800 °C for 5 min with the precursors-to-salt weight ratio of 1:1.5, followed by different annealing conditions: with and without annealing at 600 °C for 10 and 20 h. These samples underwent identical washing and Li-reinsertion steps. With small particle sizes, all NM-LMTO samples deliver reasonably high initial discharge capacities above 190 mAh/g. However, in the absence of the second annealing step, the 800 °C-5 min NM-LMTO exhibits a shorter first charging voltage plateau at around 4.4 V, a typical feature of LMTO reported in the literature associated with oxygen oxidation[36,40]. Our speculation focuses on the Ti-rich shell (specifically Ti⁴⁺-rich, representing a $d^0$ transition metal devoid of valence band electrons) of the non-annealed NM-LMTO, as shown in Fig. 2a. This shell, characterized by lower electrical conductivity compared to when Mn³⁺ and Ti⁴⁺ are thoroughly mixed, likely diminishes the charging capacity and restricts the extent of oxygen oxidation. Regarding initial discharge capacity, the 800 °C-5 min NM-LMTO annealed at 600 °C for 10 h achieves the highest value of ~220 mAh/g among the three samples, showing a clear O-oxidation plateau and larger first charging capacity than the non-annealed sample. Longer annealing for 20 h decreases the capacity to ~194 mAh/g, likely due to increased LMTO particle agglomeration (as demonstrated in Fig. 2c, d), which imposes a longer Li diffusion length during cycling.

Figure 3e, f demonstrate the effect of the Li-reinsertion step after DIW washing on the performance of NM-LMTO in Li∥NM-LMTO cells by comparing voltage profiles and capacity retention before and after Li-reinsertion for sub-200 nm NM-LMTO (800 °C-5 min calcination; 600 °C-10 h annealing; 1.5–4.8 V, 20 mA/g, 25 °C). Without the Li-reinsertion step after DIW washing (resulting in Li⁺/H⁺ exchange), the sub-200 nm NM-LMTO exhibits an initial discharge capacity of only ~152 mAh/g with significant overpotential during initial charging. Conversely, after Li-reinsertion (reversing the Li⁺/H⁺ exchange), the sub-200 nm NM-LMTO restores its voltage profiles and capacity, reaching ~220 mAh/g in the first discharge to 1.5 V with a defined first charging oxygen oxidation plateau expected for LMTO. These experiments underscore the importance of these washing and Li-reinsertion steps (to limit Li⁺/H⁺ exchange) for NM-LMTO

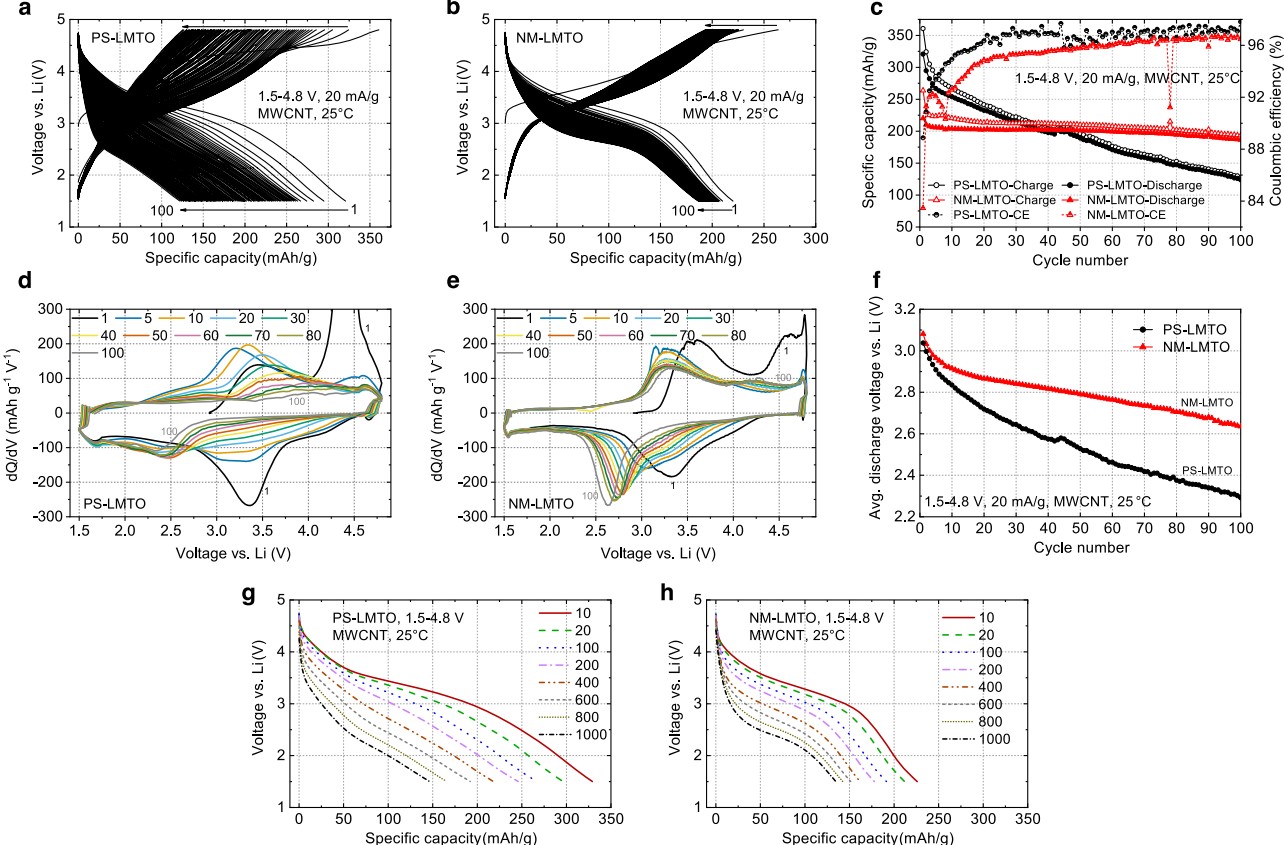

**Fig. 4 | Performance of PS-LMTO vs. NM-LMTO in the 70(LMTO):20(MWCNT):10(PVDF) electrode in Li‖PS/NM-LMTO cells.**
**a**, **b** Voltage profiles of (**a**) PS-LMTO and (**b**) NM-LMTO when cycled between 1.5 and 4.8 V at 20 mA/g and 25 °C, and (**c**) their capacity retention and coulombic efficiency (%). **d**, **e** The dQ/dV plots over various cycles of (**d**) PS-LMTO and (**e**) NM-LMTO from the above tests, with legends indicating the corresponding cycle numbers, and (**f**) their average discharge voltage retention. **g**, **h** The discharge voltage profiles of (**g**) PS-LMTO and **h** NM-LMTO when they are charged at 20 mA/g to 4.8 V and discharged at different rates of 10, 20, 100, 200, 400, 600, 800, and 1000 mA/g to 1.5 V at 25 °C. The NM-LMTO in this figure was prepared with an initial molten-salt calcination at 800 °C for 5 min followed by the annealing at 600 °C for 20 h with a precursor-to-salt weight ratio of 1:1.5. Source data for (**a**–**h**) are provided as a Source Data file.

performance. In the SI, we also explore the effect of preventing Li⁺/H⁺ exchange on the voltage profile of NM-LMTO using a more traditional LiOH solution washing process, instead of Li reinsertion after DIW washing, with varying pH levels (Fig. S11). Although further dedicated study on the detrimental effects of Li⁺/H⁺ exchange is warranted, we speculate that degraded performance following this exchange may be attributed to residual H⁺ ions in the LMTO structure hindering Li diffusion by occupying tetrahedral sites crucial for Li diffusion in a rock-salt-type structure[7].

## The performance of PS-LMTO vs. NM-LMTO in the MWCNT electrode

To demonstrate the benefits of directly producing sub-200 nm Mn-DRX single particles, we compared the performance of NM-LMTO (calcined at 800 °C for 5 min, annealed at 600 °C for 20 h, Li-rein-serted, with a 1:1.5 precursor-to-salt weight ratio) and PS-LMTO in 70(LMTO):20(MWCNT, multi-walled carbon nanotube):10(PVDF) electrodes in Li‖NM/PS-LMTO cells. The MWCNT and LMTO were mixed using a shaker mill for 1 h. Due to MWCNT's higher surface-to-volume ratio and improved electrical conductivity compared to CB, mixing with MWCNT enables stable performance of NM-LMTO even without mechanical carbon coating, thereby preserving NM-LMTO's particle size and morphology[10]. In the SI, we present additional performance data for PS-LMTO and NM-LMTO in 90:5(MWCNT):5 electrodes (MWCNT mixed with LMTO at 300 rpm for 30 min using a planetary ball mill instead of a shaker mill) and 85:2.5(SWCNT, single-walled carbon nanotubes):7.5(MWCNT):5 electrodes (prepared via direct carbon slurry mixing) (Fig. S12). We also show the performance of S-LMTO in 70:20(MWCNT):10 and 70:20(CB):10 electrodes (Fig. S13).

Figure 4a–c compare the voltage profiles, capacity retention, and coulombic efficiency (CE) of PS-LMTO and NM-LMTO cycled in the 70:20(MWCNT):10 electrodes in Li‖NM/PS-LMTO cells between 1.5–4.8 V at 20 mA/g and 25 °C. PS-LMTO initially achieves a discharge capacity of around 321 mAh/g but quickly decays to about 124 mAh/g after 100 cycles (38.6% capacity retention, average CE of 96.4%), corresponding to a capacity loss of 1.97 mAh/g per cycle. In contrast, NM-LMTO exhibits an initial discharge capacity of approximately 220 mAh/g, decreasing to about 187 mAh/g after 100 cycles (85% capacity retention, average CE of 95.2%), resulting in an average capacity loss of 0.33 mAh/g per cycle.

This demonstrates that NM-LMTO outperforms PS-LMTO in terms of capacity retention, despite having a slightly lower average CE. The lower CE for NM-LMTO could be attributed to either: (i) residual moisture or LiOH from washing and Li-reinsertion processes, which are not fully removed during vacuum drying and may promote electrolyte decomposition, or (ii) a more uniform MWCNT distribution in the NM-LMTO electrode compared to the PS-LMTO electrode (as will be discussed in detail in the next section). This uniform distribution can enhance the MWCNT/electrolyte contact area, potentially leading to greater electrolyte decomposition at high voltage. While both factors are plausible, the increase in NM-LMTO's average CE from ~95%

(averaged over 60 cycles) to ~98% (over 60 cycles) with an increase in NM-LMTO content from 70% (70:20:10 electrode) to 90% (90:5:5 electrode) suggests that residual moisture or LiOH is unlikely to be the primary cause of the lower CE observed in NM-LMTO (Figs. 4b, S12c). If these residues were the main issue, the CE would likely decrease with increased active material content, as this would also increase the residue content. Instead, the results point to the difference in MWCNT distribution between NM- and PS-LMTO electrodes as the key factor affecting CE. Although PS-LMTO exhibits a slightly higher CE, its poorer capacity retention compared to NM-LMTO indicates that electrolyte decomposition is not the primary cause of capacity fading in the cycled cells.

In the SI, we also present the performance of NM-LMTO synthesized at 800 °C for 5 min of calcination and 600 °C for 10 h of annealing (instead of 600 °C for 20 h of annealing), which shows similarly good performance (Fig. S14). Additionally, in Li‖NM/PS-LMTO cell tests, NM-LMTO exhibits improved capacity retention over PS-LMTO in other electrode formulations, achieving 85.3% retention over 60 cycles in the 90:5(MWCNT):5 electrode and 90.9% retention over 50 cycles in the direct carbon-slurry mixed 85:2.5(SWCNT):7.5(MWCNT):5 electrode, versus 46.8% and 54.3%, respectively, for PS-LMTO under the same formulations (Fig. S12). The Graphite‖NM/PS-LMTO full-cell performance is also shown in the SI for both NM- and PS-LMTO using the 90:5:5 electrode, highlighting the improved cycling stability of NM-LMTO (Fig. S15, Supplementary Note 1).

The reversible capacity and capacity retention of DRX materials are strongly influenced by the conditions of particle pulverization after solid-state synthesis and carbon mixing, which affect particle size, morphology, and contact with carbon. Larger capacity utilization typically accelerates capacity loss in DRX materials. Given that PS-LMTO, with smaller particle sizes, achieves higher initial capacities than NM-LMTO, it would be inaccurate to attribute NM-LMTO's improved capacity retention solely to its particle morphology and crystallinity. Indeed, reducing capacity utilization by lowering the charge cut-off voltage improves the capacity retention of PS-LMTO (Fig. S16). However, LMTO produced through solid-state synthesis and subsequent particle pulverization has consistently shown limited capacity retention in the literature, as observed in our experiment[36,40]. Furthermore, while S-LMTO (with MWCNT mixed via shaker milling) achieve capacities similar to NM-LMTO in the early cycles, its capacity retention remains worse (Fig. S13).

With improved capacity retention, NM-LMTO demonstrates better voltage retention than PS-LMTO. Figure 4d, e show the dQ/dV plots of PS- and NM-LMTO from the 1.5–4.8 V cycling test. For PS-LMTO, the discharge dQ/dV peak at approximately 3.3 V in the first cycle is quickly lost upon cycling, and a new discharge dQ/dV peak at around 2.4 V appears after 100 cycles. Additionally, the initial charging dQ/dV peaks at approximately 3.4 V and 4.4 V are completely lost upon cycling, showing no clear oxidation peaks after 100 cycles. In contrast, NM-LMTO's discharge dQ/dV peaks shift more slowly from around 3.3 V to 2.7 V over 100 cycles, and its charging dQ/dV peaks remain more clearly defined upon cycling. This indicates more reversible redox processes in NM-LMTO than in PS-LMTO. As a result, NM-LMTO's average discharge voltage loss is much slower than that of PS-LMTO, changing from approximately 3.09 V to 2.61 V for NM-LMTO (4.8 mV loss per cycle on average), compared to a change from around 3.05 V to 2.3 V for PS-LMTO (7.5 mV loss per cycle on average) (Fig. 4f). In the SI, we show that the dQ/dV evolution of S-LMTO is similar to that of NM-LMTO (Fig. S13e), suggesting that the differences observed in PS-LMTO (Pulverized S-LMTO) are not due to intrinsically different redox processes. Instead, the pulverized particle morphology in PS-LMTO likely accelerates degradation of higher-voltage redox processes (*e.g.*, $Mn^{3+}/Mn^{4+}$, O-redox) to lower voltage processes (*e.g.*, $Mn^{2+}/Mn^{3+}$)

following greater oxygen loss, which is commonly observed in O-redox-active Mn-DRX materials[4,5,25].

We also compared the rate capability of PS- and NM-LMTO. Figure 4g, h show the discharge voltage profiles of PS- and NM-LMTO in the 70:20(MWCNT):10 electrodes when they are charged at 20 mA/g and discharged at different rates of 10, 20, 100, 200, 400, 600, 800, and 1000 mA/g between 1.5 and 4.8 V at 25 °C. PS-LMTO delivers higher discharge capacities than NM-LMTO at all rates. This is expected given the pulverized polycrystalline particles of PS-LMTO with sub-50 nm grains (Fig. 1d), which lead to shorter Li travel distances during cycling, compared to the sub-200 nm NM-LMTO single-crystal particles (Fig. 1e). However, the irregularly-shaped secondary particles formed by PS-LMTO (Figs. 1d, 5a) make Li transport less ideal despite the small primary particle size. Even so, electrolyte penetration through loosely packed secondary particles and Li diffusion along grain boundaries contribute to the high capacity of PS-LMTO, similar to the Li transport mechanism in polycrystalline NMC particles[41].

However, the capacity reduction with the increasing rate from 10 mA/g to 1000 mA/g is substantially smaller for NM-LMTO (~91 mAh/g decrease) than for PS-LMTO (~180 mAh/g decrease), resulting in only a ~10 mAh/g difference at the 1000 mA/g rate. This reduced capacity difference at high rates is likely attributed to the limitations in PS-LMTO. While the sub-50 nm nano-grains in PS-LMTO enhance Li transport, the secondary particle structure leaves many primary particles (grains) in the core of the secondary particles with limited contact with conductive carbon. This increases charge-transfer and contact resistances, which become increasingly critical for achieving high capacity at higher rates[10].

## Degradation of NM-LMTO vs. PS-LMTO upon cycling

We characterized the 70(NM/PS-LMTO):20(MWCNT):10(PVDF) electrodes before and after cycling to understand the effect of LMTO's particle morphology on the electrode microstructure and degradation.

Figure 5a, b show cross-sectional SEM images of the PS- and NM-LMTO electrodes before cycling and after 40 cycles. In these SEM images, LMTO particles appear in white (bright gray), MWCNT-PVDF in dark gray, and pores in black. The most notable difference between the two electrodes is the distribution of LMTO particles within the electrode matrix. The PS-LMTO electrode exhibits a non-uniform distribution of active material, with some pulverized PS-LMTO nanoparticles agglomerating into large secondary particles ($d > 1$ μm), along with debris-like nanoparticles showing irregular shapes.

In contrast, the NM-LMTO electrode exhibits a much more uniform distribution of sub-200 nm LMTO single particles. Such uniform Mn-DRX distribution in the electrode film has not been reported previously. Electrode uniformity plays a significant role in stabilizing capacity retention, as high uniformity enables more homogeneous redox reactions for each particle and creates uniform stress-strain relationships during cycling-induced volume changes[42–44]. This uniformity is likely due to NM-LMTO particles already being small single crystals with clearly defined morphology and reduced agglomeration compared to PS-LMTO, facilitating their mixing with carbon and binder. In contrast, PS-LMTO's particle pulverization after solid-state synthesis generates significantly agglomerated secondary particles composed of debris-like nanoparticles, requiring more intense mixing with carbon to break up the agglomerated particles. Even when carbon mixing breaks up the agglomerated PS-LMTO particles, the likelihood of achieving uniform size and morphology is low.

SEM images of the electrodes after 40 cycles suggest that the NM-LMTO electrode experiences more homogeneous and reduced degradation upon cycling compared to the PS-LMTO electrode. Both electrodes exhibit increased porosity after 40 cycles, indicated by the increased black spots in the SEM images. For the PS-LMTO electrode, the increase in porosity is observed both within the agglomerated

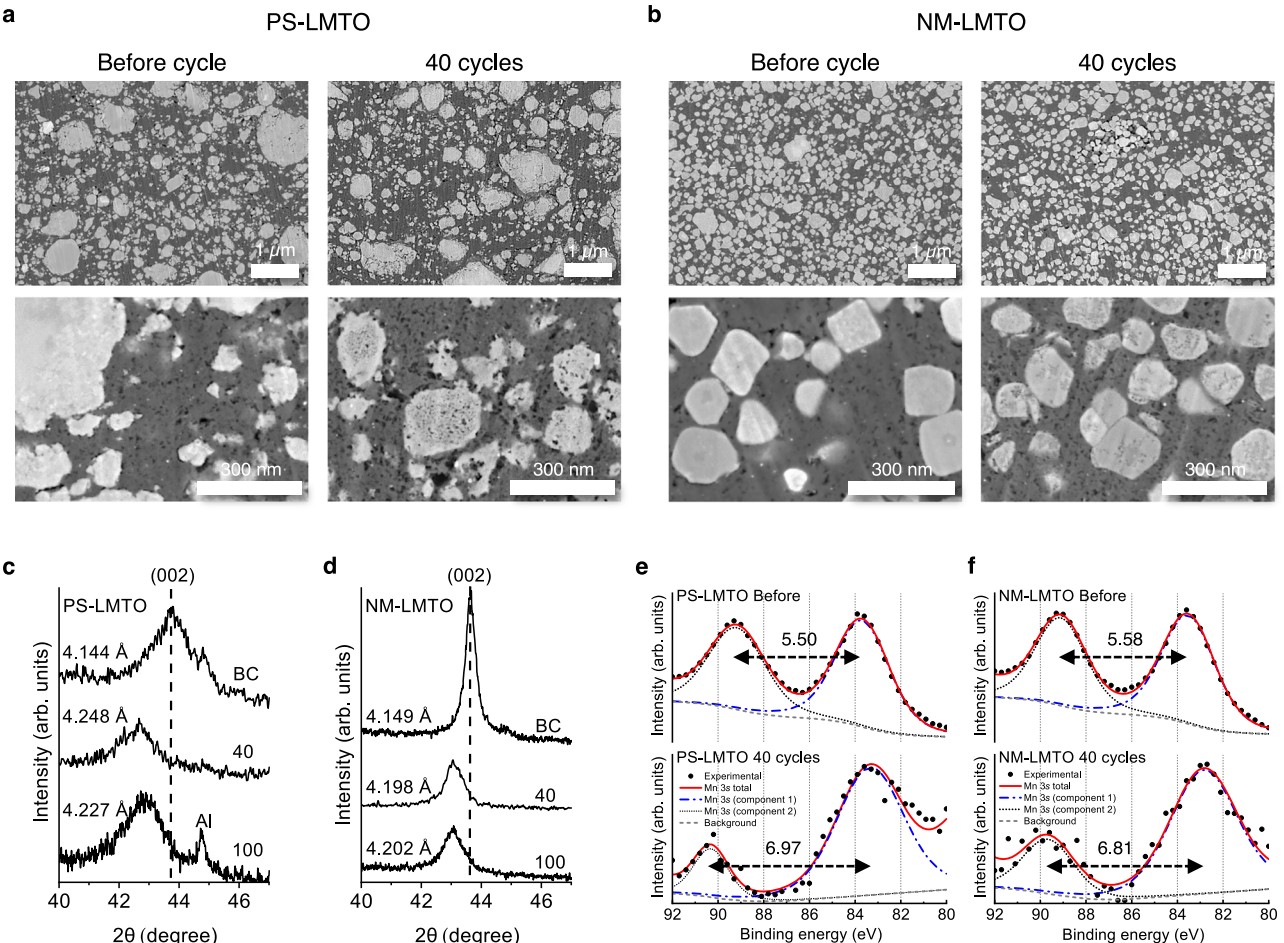

**Fig. 5 | Characterization of PS-LMTO and NM-LMTO. a, b** Cross-sectional SEM images of (**a**) PS-LMTO and (**b**) NM-LMTO in the 70(LMTO):20(MWCNT):10(PVDF) electrodes before cycling and after 40 cycles in Li||PS/NM-LMTO cells between 1.5 and 4.8 V at 20 mA/g and 25 °C. **c, d** (002) XRD peak of the (**c**) PS-LMTO and (**d**) NM-LMTO before cycling (BC) and after 40 and 100 cycles between 1.5 and 4.8 V at 20 mA/g and 25 °C. **e, f** Mn 3 *s* XPS results on the (**e**) PS-LMTO and (**f**) NM-LMTO electrode before cycling and after 40 cycles between 1.5–4.8 V at 20 mA/g and 25 °C. The Mn 3 *s* peak exhibits two multiplet-split components due to exchange coupling between the non-ionized 3 *s* electron and 3 *d* valence electrons. Source data for (**c**–**f**) are provided as a Source Data file.

particles, likely due to rearrangement of grains in the secondary particles, and in the carbon-binder matrix, caused by mechanical stress and strain from the active material's volume change or electrolyte uptake.

Previously, we demonstrated that pore generation in the Mn-DRX electrode is largely due to the volume change of the active material during cycling, which develops electrode cracks and failures over time[10]. However, the NM-LMTO electrode exhibits a smaller overall porosity increase with a more uniform distribution compared to the PS-LMTO electrode. We hypothesize that this improvement stems from the homogeneous distribution of electrode components in the NM-LMTO electrode, leading to a more even stress and strain development in the electrode matrix and more uniform electrolyte absorption during cycling.

Notably, intraparticle pore generation is significantly reduced in NM-LMTO particles compared to PS-LMTO particles. This difference is likely due to the monolithic nature of NM-LMTO particles, which restricts pore formation mechanisms. For instance, pores can form inside the single-particle Li-excess cathode materials via O-vacancy introduction, migration and segregation[45,46]. In contrast, polycrystalline secondary particles, such as PS-LMTO, can experience additional porosity increase due to grain boundary delamination within the particles.

Despite prior concerns about particle cracking in Mn-DRXs[24,47], NM-LMTO particles do not exhibit obvious intragranular cracks after 40 cycles, though the presence of nanocracks cannot be entirely ruled out. The absence of notable intraparticle cracking is likely due to NM-LMTO's smooth particle surface and small particle size. According to fracture mechanics, the fracture of ceramic materials intensifies with the presence of surface flaws, their length, and the level of external or internal stresses. Smooth particle surfaces imply fewer surface flaws, while small particle sizes limit the length of these flaws and reduce the Li-content gradient within the particles during cycling. This reduction in the Li-content gradient mitigates uneven strain development inside a particle. These features of NM-LMTO particles collectively increase the critical stress required for particle fracture, preventing cycling-induced intragranular particle cracking.

XRD analysis of the cycled electrodes indicates reduced irreversible degradation for NM-LMTO compared to PS-LMTO after cycling (Figs. 5c, d, S17, S18). Before cycling, the (002) peak positions are nearly identical, reflecting similar lattice parameters for PS-LMTO (4.144 Å) and NM-LMTO (4.149 Å). However, the NM-LMTO peak is much sharper, indicating higher crystallinity. After 40 and 100 cycles, the irreversible shift of the (002) peak to a lower angle is more significant for PS-LMTO than NM-LMTO. This behavior reflects greater irreversible volume expansion for PS-LMTO after cycling, as indicated by the change in lattice parameter from 4.144 Å (Before cycle) to 4.248 Å (40 cycles) and 4.227 Å (100 cycles) for PS-LMTO, compared to the change from 4.149 Å (Before cycle) to 4.198 Å (40 cycles) and 4.202 Å (100 cycles) for NM-LMTO. This lattice parameter increase

after cycling is most likely due to more significant oxygen loss from PS-LMTO than NM-LMTO, as oxygen loss leads to more reduced transition metal species with larger ionic radii at the discharged state. In this context, although atomic occupancy refinement may lack accuracy, our XRD refinement reveals that the O occupancy is lower for PS-LMTO than for NM-LMTO after cycling, suggesting greater oxygen loss (Figs. S17, S18). We note that even for NM-LMTO, there is peak broadening after 40 and 100 cycles, although to a lesser degree than in PS-LMTO. Given the absence of notable intragranular cracking in NM-LMTO, this peak broadening is most likely due to crystallinity loss of LMTO upon cycling, where irreversible rearrangements of ions take place.

X-ray photoelectron spectroscopy (XPS) analysis on the PS- and NM-LMTO electrodes, shown in Fig. 5e, f, confirms that the reduced structural damages upon cycling for NM-LMTO accompany reduced oxygen loss. The degree of separation of two peaks in Mn $3s$ XPS spectra is known to reflect the Mn oxidation state, with larger separations indicating more reduced states. As a reference, the peak separation values for MnO ($Mn^{2+}$), $Mn_2O_3$ ($Mn^{3+}$), and $MnO_2$ ($Mn^{4+}$) are reported to be within 5.5–6.1 eV, 5.2–5.6 eV, and 4.1–4.7 eV, respectively[48–52]. For the PS-LMTO electrode, the peak separation value increases from 5.50 eV before cycling to 6.97 eV after 40 cycles. In contrast, for NM-LMTO, the value changes from 5.58 eV to 6.81 eV after 40 cycles. The smaller peak separation for the NM-LMTO electrode compared to PS-LMTO after 40 cycles implies less oxygen loss, leading to less surface reduction. Soft X-ray absorption spectroscopy results, presented in the SI, further confirm greater Mn reduction from PS-LMTO than NM-LMTO at both the bulk and surface levels (Fig. S19, Supplementary Note 2). In the SI, we also demonstrate that the cathode electrolyte interphase (CEI) film develops less during cycling for the NM-LMTO electrode based on the O $1s$ XPS spectra (Fig. S20). Moreover, after 40 cycles, we observe ~35% less Mn deposited on the Li-metal anode chip from the NM-LMTO cell than the PS-LMTO cell, indicating reduced metal dissolution from NM-LMTO (Fig. S21). This reduced Mn deposition at the anode can help mitigate dendritic Li growth at the Li metal anode and reduce side reactions in other anode materials[53]. Furthermore, the enhanced stability of NM-LMTO, along with reduced Mn deposition (beneficial for stable Li deposition), slows impedance growth in the Li‖NM-LMTO cell during cycling, leading to more stable capacity retention compared to the Li‖PS-LMTO cell (Figs. S22–S24, Supplementary Tables 1–3, Supplementary Note 3).

## Future prospects for Mn-DRX synthesis
Since the majority of Mn-DRXs are $Mn^{3+}$ compounds and require an inert or reducing atmosphere (argon gas, nitrogen gas) for synthesis, $Mn^{3+}$ or $Mn^{4+}$ precursors (e.g., $Mn_2O_3$, $MnO_2$) are necessary, which typically have poor solubility in water[5]. This constraint limits hydrochemical precursor processing for Mn-DRXs, which otherwise offers an easy method to control particle morphology, similar to the Ni-Mn-Co hydroxide coprecipitation for NMC compounds[29].

In this regard, molten-salt methods offer opportunities, as unlike water, molten salts can dissolve $Mn^{3+}$ precursors and others with high-valent transition-metals, promoting solvent-mediated nucleation and growth of Mn-DRXs[24–26,31]. This opens additional parameters to control the morphology and size of the produced particles compared to solid-state synthesis or mechanochemistry, as demonstrated by our NM method, a modified molten-salt method.

The two-step heating protocol in our NM method, involving brief calcination above the melting point of the salt flux followed by extended annealing, could limit particle growth while forming highly crystalline Mn-DRX single particles with reduced agglomeration. We believe this two-step molten-salt method will be effective in limiting or controlling particle growth for a variety of Mn-DRXs. However, the initial calcination temperature required for DRX synthesis (preferably lower for achieving smaller particles) varies depending on the DRX composition, as the NM synthesis does not alter the relative thermodynamic stability of a given DRX phase compared to competing phases at a specific temperature. Consequently, when a higher temperature is needed to stabilize the DRX phase fully over competing phases, the required initial calcination temperature must also increase to ensure a pure phase during NM synthesis. This, in turn, makes the synthesis of small particles more challenging.

For example, while the particle sizes of NM-LMTO-70 and NM-LMNO are considerably smaller than previously reported (Figs. 1f, g, and S7), their higher calcination temperatures, required to produce pure phases, resulted in larger particles compared to NM-LMTO and NM-LNTO, which needed lower calcination temperatures. This does not rule out the possibility of synthesizing sub-200 nm particles for these compounds, as our optimization efforts for NM-LMTO-70 and NM-LMNO were less extensive. However, this underscores the need to further study the thermodynamic stability of DRX phases with different compositions to identify the lowest calcination and annealing temperatures that avoid impure phase formation during calcination and decomposition during annealing.

Additionally, further attention is warranted as other molten salts (or mixtures thereof) or different sets of Mn-DRX precursors might facilitate the formation of size-controlled Mn-DRX single particles even with a simpler one-step calcination. These alternative salt fluxes, with an optimized precursor-to-salt ratio, may reduce the diffusion rates of Mn-DRX components dissolved in the molten salt while maintaining a sufficiently high supersaturation level of the reactants. This can promote rapid nucleation while slowing down particle growth and Ostwald ripening[31]. Additionally, some molten salts with reduced solubility for Mn-DRX precursors may promote heterogeneous nucleation of Mn-DRX particles over homogeneous nucleation to reduce the average particle size, assuming the particle growth and Ostwald ripening are limited. Finally, different Mn-DRX precursor sets may provide a greater driving force for Mn-DRX formation[54], allowing further reduction of molten-salt calcination temperature to limit particle growth.

Our results demonstrated that inappropriate water washing of the molten-salt flux damages the Mn-DRX particles due to $Li^+/H^+$ exchange. In fact, the greatest hurdle to synthesize directly cyclable LMTO single particles using the NM method was not the synthesis of small particles but this washing step, which consistently limited the capacity from the NM-LMTO particles in our early trials when we were not fully aware of this $Li^+/H^+$ exchange problem (Figs. 3e, f). Upon realizing this problem, to reverse this reaction, we exposed the DIW-washed Mn-DRX particles to a concentrated LiOH solution, followed by filtering and drying the Li-reinserted powder, recovering the expected performance of LMTO. The $Li^+/H^+$ exchange phenomenon is not new for cathode materials[37,38]. However, previous studies using molten-salt synthesis of Mn-DRX washed their particles with bare DIW without performing Li-reinsertion or washing with a LiOH solution[24–26]. We believe this was feasible in previous studies because the single-crystal particles produced had much larger sizes (5–20 µm), resulting in limited surface area exposure and thus minor $Li^+/H^+$ exchange[24–26]. However, our demonstration of the $Li^+/H^+$ exchange and Li-reinsertion for sub-200 nm LMTO single particles illustrates the sensitivity of these materials to water. This finding is useful for designing hydro-chemical syntheses of Mn-DRX, if feasible, or for discussing the sensitivity of Mn-DRX to moisture in the air.

An intriguing idea is to explore the use of polar aprotic solvents, like dimethyl sulfoxide (DMSO), instead of water for washing molten salt flux. While these solvents typically exhibit lower dielectric constants and, therefore, are less effective at dissolving molten salt fluxes compared to water, their lack of acidic protons could mitigate the detrimental $Li^+/H^+$ exchange. This, in turn, could eliminate the need for the Li-reinsertion step or LiOH washing. Avoiding these steps could prevent the incorporation of residual LiOH in NM-DRX compounds,

which negatively impacts cycling performance, while also simplifying the NM synthesis process.

While the initial capacity achieved by NM-LMTO is not as high as that of PS-LMTO, its significantly improved capacity and voltage retention suggest the critical importance of controlling the microstructure and crystallinity of Mn-DRX for stable performance. Particle pulverization, commonly used to achieve high capacity in Mn-DRXs, fails to control particle size/morphology and damages crystallinity[5]. Additionally, a non-uniform electrode with pulverized Mn-DRX particles can lead to inhomogeneous redox processes and strain-stress development, accelerating electrode degradation. In this regard, the demonstration of an electrode made with uniformly distributed, highly crystalline NM-LMTO single particles offers promise for developing high-performance Mn-DRX electrodes through particle microstructure engineering, alongside Mn-DRX composition engineering.

It is important to note that ultimately, the intrinsically low Li diffusivity and electronic conductivity in Mn-DRXs must be resolved for NM-LMTO and any Mn-DRXs to achieve >250 mAh/g capacity in reasonably small particles without particle pulverization. While we achieved sub-200 nm LMTO single particles using the NM method, the capacity of ~210 mAh/g needs to be improved for NM-LMTO to deliver more attractive specific energy and capacity. This shortfall indicates the need for additional particle size reduction or the improvement of intrinsic Li diffusivity or electronic conductivity through composition engineering. While the NM method could further reduce particle size, improving the intrinsic kinetic properties of Mn-DRX should take precedence. Nanoparticles produced with the NM method would ultimately face similar challenges in secondary particle processing, such as coating, or lose energy density at the electrode level if the particle size becomes too small. If our NM synthesis were applied to Mn-DRXs with higher intrinsic kinetics, it could produce highly crystalline and well-dispersed single Mn-DRX particles with larger sizes, rather than the smaller sizes preferably demonstrated in this work. This would enhance both the capacity and energy density of the materials and electrodes, while also facilitating secondary particle processing if needed.

Besides, optimizing particle size is essential for achieving high capacities, maximizing specific energy and energy density, and minimizing capacity fading in DRX electrodes. Further research is needed to investigate how DRX composition affects operating voltage, lithium diffusivity, and electrical conductivity, as these factors play a key role in determining the optimal particle size[4,5]. This understanding will also aid in identifying the ideal particle size prior to extensive testing, thereby streamlining the development of advanced DRX materials.

Additionally, efforts should focus on developing methodologies to avoid mechanical carbon mixing via ball milling at any stage of DRX electrode preparation. As previously discussed, the performance of DRX electrodes heavily depends on the intimacy of the DRX/carbon contact, with ball milling often being the most straightforward method to achieve this. In our study, PS- and NM-LMTO electrodes demonstrated better performance when carbon was mixed via ball milling compared to direct slurry mixing (Figs. 4a, b, S12). However, ball milling introduces risks, such as potential damage to well-formed particles, while also increasing electrode production costs. Identifying alternative approaches that maintain or enhance DRX/carbon contact without relying on ball milling will be essential for improving both the performance and economic viability of DRX electrode technologies.

Finally, while this work focused on producing size-controlled DRX particles, further studies on particle surface properties are needed to better understand the reactivity of surfaces formed through molten-salt synthesis compared to those created by solid-state synthesis followed by particle pulverization. Surfaces generated from solvents typically exhibit the lowest free energies, suggesting that the surface of NM-DRX particles may be more stable than the exposed surfaces of pulverized particles. However, it remains unclear how these differences in surface structure influence the stability of DRX materials at the cathode/electrolyte interface. Such variations could result in differing degrees of metal dissolution or electrolyte decomposition[8,21]. Consequently, these distinctions may necessitate tailored approaches to engineering DRX materials at both the electrode and cell levels, critical for the practical application of DRX materials in Li-ion batteries.

## Methods

### Synthesis

The nucleation-promoting and growth-limiting molten-salt synthesis (NM) method was employed to prepare various lithium manganese titanium oxides, including $Li_{1.2}Mn_{0.4}Ti_{0.4}O_2$ (LMTO), $Li_{1.1}Mn_{0.7}Ti_{0.2}O_2$ (LMTO-70), $Li_{1.2}Mn_{0.6}Nb_{0.2}O_2$ (LMNO), and $Li_{1.2}Ni_{0.2}Ti_{0.6}O_2$ (LNTO).

For LMTO, a stoichiometric amount of $Mn_2O_3$ (Sigma-Aldrich, 99%), $TiO_2$ (Sigma-Aldrich, 99.7%), and $Li_2CO_3$ (Thermo-Scientific, 99%) with an additional 5 wt% excess of $Li_2CO_3$ were utilized as precursors. The molten salt flux included CsBr (Thermo-Scientific, 99%), CsCl (Sigma-Aldrich, 99.9%), CsI (Sigma-Aldrich, 99.9%), KBr (Thermo-Scientific, 99%), KCl (Sigma-Aldrich, 99-100.5%), or KI (Sigma-Aldrich, 99%), with a precursor-to-salt weight ratio of 1:1.5. A total of 7.5 g of salts and 5 g of precursors were dispersed in 15 mL of acetone (Sigma-Aldrich, ≥99.5%) and mixed using a Fritsch Planetary Micro Mill (PULVERISETTE 7) at 500 rpm for 6 h in a 45 mL stainless steel jar, with twenty 10 mm stainless steel balls serving as the grinding medium. The resulting mixture was collected and dried overnight in a vacuum oven at 80 °C.

For LMTO-70, $Mn_2O_3$ (Nanografi Nano Technology, 28 nm, 99.4%), $TiO_2$ (Sigma-Aldrich, 99.7%), and $LiOH•H_2O$ (Sigma-Aldrich, ≥98%) with an additional 5 wt% excess of $LiOH•H_2O$ were used, combined with CsBr as the molten salt flux. The precursor-to-salt ratio was maintained at 1:1.5, and 15 mL of deionized water (DIW) was used as the dispersing medium. The mixture was measured within the same jar and grinding media as described for LMTO and milled at 500 rpm for 6 h and then dried overnight at 80 °C.

For LMNO, $Mn_2O_3$ (Sigma-Aldrich, 99%), $Nb_2O_5$ (Thermo-Scientific, 99.5%), and $LiOH•H_2O$ (Sigma-Aldrich, ≥98%) with an additional 5 wt% excess of $LiOH•H_2O$ were combined. Similarly, LNTO utilized $NiCO_3$ (Alfa Aesar, 98%), $TiO_2$ (Sigma-Aldrich, 99.7%), and $LiOH•H_2O$ with an additional 5 wt% excess of $LiOH•H_2O$. Both mixtures were dispersed in 15 mL of acetone and placed within the same jar and grinding media as described for LMTO. Both LMNO and LNTO were milled at 500 rpm. LMNO was milled for 12 h while LNTO was milled for 6 h.

2 g of the dried precursor-salt mixture were placed in a crucible and calcined in a rapid heating furnace (MTI OTF-1200X-4-RTP) at a ramping rate of 1 °C/s. The calcination conditions varied as follows: 800 °C for 5 or 20 min and 900 °C for 5 or 20 min for LMTO, 900 °C for 1 min for LMTO-70, 950 °C for 3 min for LMNO, and 750 °C for 5 min for LNTO. Calcination was carried out under an argon atmosphere for all samples, except LNTO, which was calcined in air. After calcination, the samples were furnace-cooled, yielding materials encapsulated in solidified CsBr.

In the annealing step, the samples were treated under an argon atmosphere in a separate furnace (MTI OTF-1200X-S). The temperature was increased at a rate of 5 °C/min to 600 °C, which is below the melting point of CsBr, and maintained for specific durations based on the material: 2, 10, or 20 h for LMTO, and 5 h for LMTO-70, LMNO, and LNTO.

For CsBr removal, the samples were sonicated in 50 mL of DIW under argon flow for 15 min, followed by washing with 1000 mL of DIW under argon flow for 30 min. The resulting powder was then collected through vacuum filtration under argon.

A Li-reinsertion step was conducted by immersing 1.5 g of washed NM-LMTO, NM-LMTO-70, NM-LMNO, or NM-LNTO powders in a solution of 12.5 g of LiOH (Sigma-Aldrich, ≥98%) dissolved in 100 mL

DIW. The solution was stirred under argon flow for 6 h. The resulting powder was vacuum-filtered under argon and dried in a vacuum oven.

To synthesize LMTO through solid-state synthesis (S-LMTO), stoichiometric amounts of $Mn_2O_3$ (Sigma-Aldrich, 99%), $TiO_2$ (Sigma-Aldrich, 99.7%), and $Li_2CO_3$ (Thermo-Scientific, 99%) with an additional 10 wt% excess of $Li_2CO_3$ were used as precursors. These precursors were mixed using a PULVERISETTE 7. 5 g of the mixture were placed in a 45 mL stainless-steel jar with twenty 5 mm diameter and ten 10 mm diameter stainless-steel balls. The ball mill was operated at 400 rpm for 6 h. Following mixing, the powders were collected in an argon-filled glovebox. Two grams of the mixed powder were then calcined in a furnace (MTI OTF-1200X-S) at 1000 °C in an argon flow for 2 h, with a ramping rate of 5 °C/min, to produce S-LMTO. To prepare pulverized LMTO (PS-LMTO), the S-LMTO powder was further milled using the PULVERISETTE 7 at 500 rpm for 5 h. For this, 2 grams of S-LMTO were loaded into a 45 mL stainless-steel jar containing five 10 mm diameter and ten 5 mm diameter stainless-steel balls. The obtained PS-LMTO was collected and stored in an argon-filled glovebox (Vigor, $H_2O$ < 0.5 ppm & $O_2$ < 1 ppm) for further use.

## Electrochemistry

In the production of the 70:20:10 composite film, NM-, PS-, or S-LMTO (280 mg) were blended with either multi-walled carbon nanotubes (MWCNT, Nano Solution, TMC-230-05) or Super C65 carbon black (MSE) (80 mg) in a mixing vessel within an argon atmosphere. A 5 mL stainless-steel vial was used as the grinding vessel, containing twenty 3 mm diameter stainless-steel balls as grinding media. The vessel was sealed to preserve the argon atmosphere, then mechanically mixed using a SPEX 8000 M Mixer/Mill for 1 h for the mixture containing MWCNT and 2 h for the others based on CB. After mixing, the powder mixture was collected inside an argon-filled glovebox. 180 mg of the mixture, containing 140 mg of active material and 40 mg of carbon black or MWCNT, and 400 mg of a 5 wt% polyvinylidene fluoride (PVDF, MTI, HSV900) solution in $N$-methyl-2-pyrrolidone (NMP, Sigma-Aldrich) were added to a Thinky high-density polyethylene (HDPE) mixing container. An additional 400 µL of NMP was then added, and the resulting slurry was homogenized using a Thinky mixer (Thinky, AR-100) for 20 min. The electrode film, consisting of 70 wt% active material, 20 wt% carbon black/MWCNT, and 10 wt% PVDF binder, was fabricated by casting the slurry onto an Al foil (MSE Supplies, 99.5%, 15 µm) using a doctor blade set. The single-sided, slurry-coated Al foil was then dried in a vacuum oven at 80 °C for 12 h. After drying, the slurry-coated Al foil was calendered with an electrode roll press (STC-DG100, SHENZHEN TICO TECHNOLOGY CO., LTD.), punched out into individual discs using a cutting die with a 13 mm diameter, and finally dried again at 80 °C in a vacuum oven overnight, and transferred into an Ar-filled glovebox for coin cell assembly. The final electrode thickness, excluding the Al foil, was approximately 30–35 µm as determined by SEM measurements.

To prepare the electrode with a composition of 90 wt% active material (PS/NM-LMTO), 5 wt% multi-walled carbon nanotubes (MWCNT), and 5 wt% PVDF, the following method was employed: Initially, 400 mg of PS/NM-LMTO and 22.222 mg of MWCNT were measured inside an argon-filled glovebox into a 45 mL stainless-steel jar containing five 10 mm diameter and ten 5 mm diameter stainless-steel balls. The jar was sealed inside the glovebox, then mixed at 300 rpm for 30 min using a PULVERISETTE 7. Subsequently, 190 mg of this mixture was combined with 10 mg of PVDF (added as 200 mg of a 5 wt% PVDF in NMP solution), and additional 400 µL of NMP, and the entire batch was mixed in a Thinky mixer for 20 min. The resulting slurry was cast onto an Al foil using a doctor blade to form the single-sided coated electrode film. The electrode film was then dried, calendered, and punched as outlined for the 70:20:10 electrode film, then stored within an argon-filled glovebox for coin cell assembly.

The composite films for NM- and PS-LMTO were also prepared using a direct slurry mixing method. The composite electrode produced with this method consisted of 85 wt% NM/PS-LMTO as the active material, 7.5 wt% of MWCNT (Nano Solution, TMC-230-05), 2.5 wt% of single-walled carbon nanotubes (SWCNT, TUBALL, 1% BATT solution), and 5 wt% of PVDF (included in the TUBALL 1% BATT solution). The NM- or PS-LMTO powders were first passed through a 45-µm sieve to remove any clumps. In a Thinky mixer, 500 mg of the SWCNT/PVDF/NMP solution (providing 5 mg of SWCNT and 10 mg of PVDF), 15 mg of MWCNT, and additional 400 µL of NMP were mixed for 20 min. Subsequently, 170 mg of the active material was added, and the mixture was further mixed for 40 min. The sieving, measuring, and mixing of these materials proceeded under an argon environment. The slurry was then cast onto aluminum foil using a doctor blade to form the single-sided coated electrode film. The electrode film was dried, calendered, and punched following the procedure used for the 70:20:10 electrode film, then stored in an argon-filled glovebox for coin cell assembly. We maintained the areal loading of all NM/PS-LMTO electrodes to be approximately 4 mg of LMTO active-material powder per $cm^2$.

The Li‖NM/PS-LMTO coin cell assembly was prepared using CR2032 casings made from 316 grade stainless steel using the composite film as the positive electrode, and a single lithium metal chip (MSE Supplies, 15.6 mm diameter, 0.25 mm thick, 99.9%) as the negative electrode. The lithium metal chips did not undergo any additional alterations and were stored under an argon atmosphere for 3 months at approximately 20 °C prior to usage. Celgard 2400 polypropylene separator (25 µm thick, 41% porosity) was punched into 19 mm disks and dried overnight at 50 °C in a vacuum oven before transferring into an argon-filled glovebox. The separator was carefully placed between the single-sided coated positive electrode disc and the negative electrode to prevent direct contact. Additionally, 150 µL of the electrolyte (MU Ionic Solutions Corporation), consisting of a 1 M solution of $LiPF_6$ dissolved in an ethyl carbonate and dimethyl carbonate (EC/DMC, 1:1 v/v) mixture was evenly distributed within the cell using an Eppendorf® micropipette equipped with polypropylene tips. All these components were assembled and enclosed in an argon-filled glovebox (Vigor, $H_2O$ < 0.5 ppm & $O_2$ < 1 ppm) to minimize exposure to ambient conditions. Galvanostatic charge/discharge and rate-capability tests were performed employing a Landt CT3002A battery testing system. The coin cells were positioned within a temperature-controlled battery test chamber (Landt LBI-300HT), precisely set at 25 °C, to ensure consistent testing conditions and accurate performance assessment. For reproducibility, we clarify that at least five cells were tested for each electrochemical experiment. The data presented in the manuscript correspond to the median-performing cell in terms of capacity and retention, selected to minimize bias and reflect typical behavior. Repeated datasets were not included in the SI, as cell-to-cell variation was minimal and the median cell was representative of the overall trend. The specific capacity (mAh/g) and cycling rate (mA/g) were reported based on the mass of the active material (LMTO) in the composite electrode in Li‖NM/PS-LMTO cell tests.

The Graphite‖NM/PS-LMTO coin cells were assembled using PS-LMTO and NM-LMTO composite electrodes, prepared with the 90:5(MWCNT):5 composition as detailed earlier, and paired with graphite as the negative electrode. The graphite electrode was fabricated via direct slurry mixing, consisting of 93.2 wt% graphite (Alfa Aesar, AA40798TC), 2.5 wt% Super C65 Nano Carbon black (MSE), and 4.3 wt% PVDF (MTI HSV900), dispersed in NMP. The slurry was cast onto Cu foil (99.99%, 25 µm thick) using a doctor blade set with a controlled thickness to achieve an appropriate areal loading for Graphite‖NM/PS-LMTO cell configuration. The drying and pressing of the graphite electrodes were executed following the protocols established in our earlier preparation techniques. Graphite electrode discs were uniformly punched to a diameter of 12 mm, suitable for full coin cell

assembly. These were then paired with NM- and PS-LMTO electrode discs, each having a diameter of 13 mm. The negative (N) and positive electrode (P) capacities were balanced to achieve an N/P ratio of approximately 1.2. This was based on the first discharge specific capacity of ~350 mAh/g for graphite, the first charge capacity of ~270 mAh/g for NM-LMTO, and ~380 mAh/g for PS-LMTO, each measured at a constant specific current of 20 mA/g in Li||NM/PS-LMTO cell configurations. The areal loading of the PS/NM-LMTO was approximately 4 mg$_{LMTO}$/cm² (based on the mass of the LMTO active material), and the graphite electrode loading was accordingly adjusted to maintain N/P ≈ 1.2 in both NM-LMTO and PS-LMTO full-cells. The cell configuration, casing materials, and fabrication process of Graphite|| NM/PS-LMTO cells was identical to that used for the Li||NM/PS-LMTO cell assembly.

In the three-electrode coin cell configuration, a 4 mm diameter copper mesh (EL-Cell, 35 mesh, 0.04 mm thick) was lithium-plated and employed as the reference electrode[55]. The cell assembly involved positioning the copper mesh between the positive electrode and negative electrode, with two separators placed on either side (one adjacent to the positive electrode and the other to the negative electrode) ensuring that the copper mesh was enclosed between them. Electrical contact with the reference electrode was established via an insulated copper wire, which was routed through the side opening between the gasket and the edge of the positive cap before crimping. Lithium plating onto the copper mesh was performed by using it as the working electrode and lithium metal as the counter electrode. The plating was achieved by discharging the cell at a constant current of 5 μA for one hour.

Electrochemical impedance spectroscopy (EIS) measurements were performed using an SP-200 Biologic workstation on various cell configurations: Li||PS/NM-LMTO cells, PS/NM-LMTO||PS/NM-LMTO, Li||Li cells, and three-electrode cells. These measurements were taken under open-circuit conditions both before cycling and after designated cycle counts. An amplitude of 5 mV was applied across a frequency spectrum, which varied depending on the cell type: from 200 kHz to 5 mHz for data presented in Fig. S22 (half-cell EIS), from 200 kHz to 0.1 Hz in Fig. S23 (symmetric cell EIS), and from 200 kHz to 10 mHz for the three-electrode EIS (Fig. S24). For all the EIS measurements, the cells were rested at open-circuit voltage for 6 h. A 5 mV perturbation amplitude was applied, and data were collected at 6 points per decade over the selected frequency range. The EIS data is reported using parametric Nyquist plots where symbols represent measured data and solid lines shows fitted data. Fitting was performed using EC-Lab software and plotted using Origin graphing software.

## Characterization

To evaluate the crystallinity and phase purity of the samples, X-ray diffraction (XRD) patterns were collected using a Malvern PANalytical Empyrean X-ray diffractometer with a copper source, spanning a 2θ range from 10° to 90°. For XRD analysis of the electrodes before and after cycling, the coin cells were carefully disassembled in a controlled inert atmosphere inside an argon-filled glovebox. The electrodes were then briefly immersed in dimethyl carbonate (DMC) to remove any residual electrolyte species. To conduct XRD on the electrode film without exposure to air, an airtight PEEK domed sample holder with a zero-background silicon plate was employed. For enhanced data analysis, Rietveld refinement was performed using the PANalytical X'pert HighScore Plus software.

The cross-sections of NM-LMTO particles, both before and after argon annealing were prepared using a Hitachi Ethos NX5000 focused ion beam scanning electron microscope (FIB-SEM). The powder samples were placed onto double-sided carbon tape. Before FIB processing, platinum (Pt) was deposited onto the surfaces via electron beam-assisted deposition (EBD) to protect the samples and fill gaps between individual particles, thus preserving the integrity of the lamella.

HAADF-STEM images and EELS spectrum images of the cross-sectional samples were acquired using a Thermo-Scientific Talos F200X STEM operated at 200 keV, equipped with a Gatan Enfinium ER Model 977 EEL spectrometer with a collection angle of 25.11 mrad. The DUAL-EELS acquisition mode was employed to simultaneously capture both the low-loss and core-loss regions, with an energy dispersion of 0.25 eV/channel. Both core-loss and low-loss signals were aligned and calibrated using the zero-loss peak. The particle size of NM- and PS-LMTO was determined using a Hitachi SU-8000 scanning electron microscopy (SEM) and a Malvern Zetasizer Nano ZS instrument.

Elemental analysis for various compounds was conducted using inductively coupled plasma optical emission spectrometry (ICP-OES) with the Thermo Scientific iCAP 6000 series, which provides a detection limit below 1 ppm and an analytical error of less than 5%. The Mn concentration in the lithium chip of the cycled NM-LMTO and PS-LMTO electrodes was measured using the same ICP-OES instrument. Standard solutions for ICP were prepared by diluting Sigma-Aldrich stock solutions: Li (998 mg/L ± 4 mg/L), Mn (1003 mg/L ± 5 mg/L), Ti (1000 mg/L ± 2 mg/L), Ni (1000 mg/L ± 2 mg/L), and Nb (1000 mg/L). For the DRX powders, sample preparation involved digestion with a mixture of $H_2O_2$ and $HNO_3$ in a 3:4 wt% ratio. For the detection of Mn on the lithium chip, the coin cell was disassembled in the glovebox, and the lithium chip was then dissolved in a 2 wt% $HNO_3$ solution.

To quantify dissolved lithium species resulting from $Li^+/H^+$ exchange, a 10 mL aliquot of the washing solution, collected after washing the NM-LMTO powder (previously calcined and rinsed in 1000 mL DIW to remove CsBr), was titrated with 0.01 M HCl. The pH was continuously measured using a benchtop pH meter (Orion Star A211) equipped with a Thermo Scientific Orion 9107BN Triode 3-in-1 pH probe with automatic temperature compensation.

The cross-sections of the electrode films of NM-LMTO and PS-LMTO, both before and after 40 cycles, were prepared using a Hitachi IM4000Plus Ar ion milling system at 6 keV. Subsequently, SEM images of the electrode films were captured using a Hitachi SU9000 SEM/ STEM at 1.2 kV. Energy dispersive X-ray spectroscopy (EDS mapping) was conducted using the Oxford Instruments Extreme detector.

X-ray Photoelectron Spectroscopy (XPS) of the positive electrode films before and after cycling was carried out using a Thermo-Scientific instrument with Al $K\alpha$ radiation as the excitation source. To prevent air exposure during transfer from the glovebox to the XPS machine, an airtight holder was employed after washing the electrode film briefly in DMC and allowing it to dry overnight within the argon-filled glovebox. Spectra were acquired with a spot size of 400 μm and a constant pass energy. The binding energy scale was corrected for charge effects using the C 1$s$ peak at 284.5 eV of carbon nanotubes (CNTs) to account for hydrocarbon contamination. Data processing for the XPS results was performed using the Thermo Scientific Avantage Data System.

Soft X-ray Absorption Spectroscopy (soft XAS) was conducted in both partial fluorescence yield (PFY) and total electron yield (TEY) modes at beamline 13-2 of the Stanford Synchrotron Radiation Lightsource (SSRL), SLAC National Accelerator Laboratory. Samples were mounted on a holder using double-sided carbon tape, placed in an airtight container inside an argon-filled glovebox, and transferred to the measurement chamber under ultra-high vacuum (~5 × 10$^{-9}$ Torr) via a load-lock system to prevent air exposure. The incident beam was monochromatized using a spherical grating monochromator (600 lines/mm), achieving an energy resolution better than 0.1 eV and a beam flux of approximately 1e11 photons/s at both the O $K$-edge and Mn $L$-edge. Samples were positioned at a 30° incidence angle relative to the sample surface. XAS spectra were acquired by sweeping the incident beam energy across the O $K$-edge and Mn $L$-edge while simultaneously collecting PFY and TEY signals. The PFY signal was recorded using a transition-edge sensor (TES) spectrometer, positioned perpendicular to the incident beam, which measured the energy and intensity of scattered X-rays. The intensity of these

scattered X-rays was normalized to the incident beam intensity ($I_0$), determined via the drain current of a gold mesh placed upstream of the chamber, and integrated over the O $K$-edge and Mn $L$-edge emission regions to obtain PFY XAS spectra. To minimize elastic scattering, the beam was linearly polarized in the horizontal direction, and the incident photon energy was calibrated post-measurement. Additional details on the TES spectrometer can be found in the reference paper by Lee et al.[56]. The TEY signal was measured as the drain current generated by the sample as a function of beam energy using a current amplifier and was normalized to $I_0$. Each XAS spectrum required approximately 20 min per sample for the O $K$-edge, comprising five repeated 4 min scans, and 16 min per sample for the Mn $L$-edge, consisting of four repeated 4 min scans. This approach enabled real-time monitoring of sample degradation and incident photon energy drift, with no evidence of sample damage observed. The energy shift for Mn $L$-edge and O $K$-edge spectra were 3.3 eV and 1.9 eV, respectively[57].

## Data availability
The data that support the findings of this study are available from the corresponding author upon request. Source data are provided with this paper.

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

## Acknowledgements

J.L. acknowledges funding from the Wildcat Discovery Technologies, Inc. and the NSERC Discovery Grant (NSERC RGPIN-2020-04463). H.A. acknowledges funding from the McGill Engineering Doctoral Award (MEDA) and Fonds de recherche du Québec (FRQNT) scholarships. M.W. acknowledges funding from MEDA. R.G. acknowledges support from the NSERC Discovery Grant (NSERC RGPIN-2024-04977) and the Facility for Electron Microscopy Research (FEMR) at McGill University. D.-H.S. acknowledges support from the National Research Foundation of Korea (NRF), funded by the Ministry of Science and ICT (RS-2023-00282389). G.P.D. acknowledges funding from the NSERC Strategic Project Grant (NSERC STPGP 521217-18). Use of the Stanford Synchrotron Radiation Lightsource, SLAC National Accelerator Laboratory, is supported by the U.S. Department of Energy, Office of Science, Office of Basic Energy Sciences under Contract No. DE-AC02-76SF00515. S.-J.L. acknowledges the Quantum Sensors Group at NIST-Boulder for their contributions to the development and support of the TES spectrometer at SSRL.

## Author contributions

J.L. planned the project and supervised all aspects of the research. H.A. performed all the synthesis with support from M.W., completed EIS measurements with assistance from N.M., conducted electrochemical testing with support from M.W., and carried out XRD, XPS, ICP, and Zetasizer characterization. N.D., P.T.L., N.B., and R.G. characterized and analyzed the SEM and STEM data. R.F., S.-J.L., and G.L. performed the XAS characterization and analysis. D.-H.S., R.G., G.P.D., and J.L. served as supervisors of this project, discussing all aspects of the research. J.L. and H.A. wrote the manuscript, and all authors contributed to revising it.

## Competing interests

The authors declare no competing interests.
