## [Transparent Peer Review file · Nature Communications]

Nucleation-promoting and growth-limiting synthesis of disordered rock-salt Li-ion cathode materials

Corresponding Author: Professor Jinhyuk Lee

Version 0:

Reviewer comments:

Reviewer #1

(Remarks to the Author)

In this manuscript, the authors report a synthesis procedure to control the size and size distribution of $\text{Li}_{1.2}\text{Mn}_{0.4}\text{Ti}_{0.4}\text{O}_2$ (LMTO) DRX primary particles. Small particle size of about 200 nm was achieved by suppressing particle growth in the molten-salt synthesis. The effectiveness was demonstrated in electrochemical analysis where their sample demonstrates superior performance.

Achieving uniform particle size is indeed one of the challenges in DRX research. The authors found a way to separate the processes that lead to DRX phase formation, particle nucleation and growth. The paper not only offers an approach to produce smaller sized DRX that are fairly uniform, but also provides some useful insights in DRX synthesis using the molten salt method, such as temperature control, salt and washing effects. I recommend publication after major revision. For improvement, I suggest the following:

- 1) Provide coulombic efficiency analysis of the cycling data. It's possible that there are some side reactivities and they may be quite different in the PS and NM samples, especially those made by changing the PH of the washing water.
- 2) For stability comparison, it would be useful to cycle the sample by setting capacity limit instead of voltage limit. For example, Figure 4 shows much faster capacity and voltage decay in the PS sample. However, the PS sample delivers a much higher initial capacity than that of the NM sample. That itself may lead to faster decay as the PS sample simply experiences deeper Li extraction and insertion.
- 3) Evaluation and discussion on optimal DRX particle size. Although it's easy to see the benefit of uniform particle size, it isn't clear what size is best for DRX. Small particles can improve kinetics but large particles improves packing density and reduces side reactions at high voltages. Large particles also eliminate the need of excessive amount of carbon in electrode fabrication.
- 4) Would the success of this synthesis procedure also apply to other DRX compositions, especially those with higher Mn content ($\text{Mn} > 0.7$) that of great interest in the community? The authors should offer some initial data on that front or at least comment on what challenges may arise. Currently, the results are solely on the $\text{Li}_{1.2}\text{Mn}_{0.4}\text{Ti}_{0.4}\text{O}_2$ composition.

Reviewer #2

(Remarks to the Author)

This paper investigates the synthesis of Mn-DRX cathode active materials using molten salt and grinding methods and compares their electrochemical characteristics. The most notable difference observed in this study is that particles synthesized with molten salt exhibit more uniform sizes compared to those synthesized by grinding. Due to the inherently low conductivity of the material itself, a considerable amount of conductive carbon of 20 wt.% versus the active materials was added for testing purposes. Figure 4 presents the electrochemical properties measured with MWCNT of 20 wt.% versus the active materials. These carbon additives are too much to fabricate the electrode and not suitable in terms of side reactions with electrolyte on high voltage region.

The material discussed has been extensively researched over a considerable period, and changes in properties based on synthetic methods are already well-documented. A similar study was published in Journal of Alloys and Compounds 797 (2019) 961-969, prompting questions regarding the originality and novelty of this paper. Additionally, it is likely that the differences in particle formation by synthetic methods would lead to variations in electrochemical reaction mechanisms. Detailed mechanism of XRD pattern changes resulting from oxygen release is necessary in this work. Moreover, the paper

lacks discussion on oxygen redox and TM (transition metal) redox processes as they relate to particle transformations. For practical applications, methods for suppressing Mn dissolution and results from a full cell paired with a graphite anode should be included. In the AC impedance section, there is a significant difference in post-cycling between the NM and PS powders, likely attributable to the impact of Li dendrites on the lithium anode. To completely rule out such factors, impedance measurements using a three-electrode cell are recommended.

Notably, this paper does not highlight a distinct originality or novelty in comparison to other similar studies, with many similarities observed, particularly to a recent publication (<https://doi.org/10.1038/s41560-023-01233-8>) despite compositional differences. Given these considerations, submission to a more specialized journal may be advisable.

Reviewer #3

(Remarks to the Author)

Commercially available cathode materials for lithium-ion batteries typically contain significant amounts of nickel and cobalt. However, there is a growing demand for the development of nickel- and cobalt-free cathodes due to the limited availability of these resources and associated safety concerns. Manganese-based disordered rocksalt lithium-excess cathode materials (Mn-DRXs) have been proposed as viable alternatives to the established nickel- and cobalt-based layered oxides. Low intrinsic lithium diffusivity within Mn-DRX requires to make cathode particles small enough. This research proposes a method to synthesize highly crystalline, size-controlled single Mn-DRX particles, which exhibit higher capacity and average voltage retention compared to Mn-DRX prepared by solid state method and following pulverization. This work is a valuable demonstration that offers new insights into the more controllable production of DRX particles. However, the following points should be addressed prior to consideration for publication.

1. Although the primary claim of this manuscript is control over particle size, the Mn-DRX particle size is only roughly described as sub-200 nm. To rigorously assess the controllability of the proposed synthesis method and the correlation between particle morphology and performance, a more comprehensive quantitative statistical analysis of Mn-DRX particle size across various preparation methods is recommended. This analysis may include the number of particles used for statistical assessment, average particle size, standard deviation.
2. Compositional inhomogeneity is presented after calcination (Figure 2a). Does crystalline heterogeneity exist between Ti-rich surface and Mn-rich core?
3. It is recommended to check the scale bar in Figure 2a or figure caption. The given information does not match to the main text. Also, getting EELS map from micron-sized particles is not feasible.
4. Instead of Li-reinsertion via LiOH solution, is electrochemical lithiation a feasible approach? Implementing this method could potentially eliminating one step in DRX particle preparation.
5. Authors attributed higher rate capability of PS-LMTO to smaller grains than NM-LMTO. However, the cross-sectional SEM image of it reveals significant agglomeration into large secondary particles, which is not ideal to deliver capacity at high rates. Please clarify this inconsistency.
6. The clarify of cross-sectional SEM images in Figure 5a and b needs to be improved. They are (except the one from PS-LMTO after 40 cycles) too blur to see microstructures.

Reviewer #4

(Remarks to the Author)

In this manuscript, Hoda Ahmed et al. reports a method to synthesize 200 nm-sized cation disordered rock salt (DRX) and studies the influencing factors for morphology control and electrochemical performance. It was found that using CsBr molten salt flux with a short-time calcination + lower temperature extended annealing + water washing + ion exchange procedure can produce 200 nm-sized particle-like LTMO. The significance of producing 200 nm-sized LTMO is that it potentially offers a possibility of mass production of DRX-based batteries. As DRX-based cathodes have been well-known for their electronic conductivities orders lower than those of layered cathodes, an extensive high-energy carbon milling process is needed to realize the electrochemical performance. The requirement of high-energy carbon milling intrinsically limits the batch size of DRX electrodes and basically make scaled-up DRX-based battery production impossible. This work can potentially provide an important progress in scaling up the production of DRX-based batteries. After reading through the manuscript, I have following concerns and think these concerns need to be addressed before publication on Nat Comm.

1. First of all, I think it would be better if the work can cover a broader range of DRXs, or at least Mn-based DRXs, instead of just one DRX compound, to prove the effectiveness of the method and increase the impact of the work. This will make this paper more suitable for a journal like Nat Comm.

For example, fluorinated-DRXs are reported to have better cycle performance. Higher-valence d0 transition metal-based DRXs like Nb-based DRXs can offer Mn-rich DRX with more Mn-redox capacity contribution like $\text{Li}_{1.2}\text{Mn}_{0.6}\text{Nb}_{0.2}\text{O}_2$. If the method raised by the authors can be applied to these DRXs mentioned above, the work will be much more impactful and more suitable for Nat Comm.

2. Another important point: Apparently, in this work, the NM-LTMO (200 nm-sized LTMO realized by the modified molten salt method) is still mixed with carbon with a milling process. In this case, frankly I think the advantage of synthesizing a 200 nm-sized LTMO is no longer that significant. This point is extended to the following aspects.

a. Mixing the DRX powder with carbon using 8000M Mixer/Mill for 2 hours is not that gentle in the first place (compared to non-milling-based carbon mixing). I am wondering if the S-LTMO (i.e. solid-state synthesized LTMO, big micron particle) is applied with the same carbon mixing (8000M Mixer/Mill for 2 hours), how is this performance compared to NM-LTMO?

- b. Key point: if the NM-LTMO is applied with non-milling-based (i.e. mortar-pestle) carbon mixing (i.e. the carbon mixing method for layered cathode and the method that can allow mass production), what the performance of NM-LTMO is like? This data is the key to the highlight of the work.
3. For the synthesis method, the results showed that lower temperature (800°C) and shorter time (5min) are beneficial to obtaining sub-200 nm primary particles. Could it be more helpful or with similar effect if keep reducing the temperature or shortening the time?
4. In Figure 2a, the scale bar is 1 μm , is it contradictory with argument of sub-200 nm primary particles? Please check them.
5. In the screen steps for molten salt-flux effect of different salts, the authors argued that “Cs-salts, due to their lower melting points and higher dielectric constants, result in higher LMTO purity than K-salts under the same heating protocol”. Could you please provide more detailed information on the relationship between dielectric constants of the molten-salt and the purity of the resultant product or how dielectric constants of the molten-salt influence the purity of LMTO during synthesis if possible?
6. For XRD refinement, there are a few issues.
- a. In the refinement of all the XRD results, it seems that all the atoms (metal atoms in 4a site and O in 4b site) in every refinement shared the same Biso value of 0.5. Please check it.
- b. Also, (111) diffraction is not shown in Supplementary Fig. 10 and 11.
- c. The authors attributed the lattice expansion of both NM- and PS-LMTO to the oxygen loss during cycling, but which was not reflected from the lattice parameters via refinement (e.g., Figure S10f, 11d, and 11e), where O occupancy was still 1. Please comment on this.
7. In figure 4d and 4e, the unit or axis values for dQ/dV are probably wrong, please check it.
8. Also, for the dQ/dV comparison, the PS-LMTO showed the main discharge peak around 2.4V, while the corresponding peak in NM-LMTO was at $\sim 2.7\text{V}$. It implies that the mechanism is different. More discussions are needed here.
9. By comparing the content of non-lattice oxygen species (OH, C-O, C=O, and $\text{Li}_x\text{PF}_y\text{O}_z$) from XPS, the authors concluded that less CEI developed in NM-LMTO than in PS-LMTO. Considering the almost unchanged OH intensity of PS-LMTO before and after cycling, the relative intensity of C-O, C=O, and $\text{Li}_x\text{PF}_y\text{O}_z$ seems more obvious in NM-LMTO, which seems not supporting this point.
10. In Supplementary Fig. 7, if I understand it correctly, the volume equivalent to Li_2CO_3 is should be $(V_2 - V_1)$, instead of $2(V_2 - V_1)$ right? (as one carbonate consumes two H^+).

Version 1:

Reviewer comments:

Reviewer #2

(Remarks to the Author)

Still some more data are not clear in this stage. The reviewer still sees insufficient facility time for the measurement, out of focus for the full cell test, and failure in measurement using three-electrode cells, Therefore, revision is requested to clear those mentioned points in the revision.

Reviewer #3

(Remarks to the Author)

The authors have addressed the reviewer's comments effectively. The manuscript has shown significant improvement, and I recommend it for publication.

Reviewer #4

(Remarks to the Author)

The authors have addressed my comments comprehensively. Additional compositions were synthesized based on the

method to support the broader applicability of the synthetic approach. Also, more comparative electrochemical measurements based on different carbon coating and carbon mixing parameters were applied to justify the synthesis of 200 nm particles. The other small errors, e.g. labeling and units, etc., were also corrected. For me, the paper is good to go.

Version 2:

Reviewer comments:

Reviewer #2

(Remarks to the Author)

Better than the previous version. I accept this work.

Response to Reviewer #1

In this manuscript, the authors report a synthesis procedure to control the size and size distribution of $\text{Li}_{1.2}\text{Mn}_{0.4}\text{Ti}_{0.4}\text{O}_2$ (LMTO) DRX primary particles. Small particle size of about 200 nm was achieved by suppressing particle growth in the molten-salt synthesis. The effectiveness was demonstrated in electrochemical analysis where their sample demonstrates superior performance.

Achieving uniform particle size is indeed one of the challenges in DRX research. The authors found a way to separate the processes that lead to DRX phase formation, particle nucleation and growth. The paper not only offers an approach to produce smaller sized DRX that are fairly uniform, but also provides some useful insights in DRX synthesis using the molten salt method, such as temperature control, salt and washing effects. I recommend publication after major revision. For improvement, I suggest the following:

We extend our **sincere appreciation to Reviewer #1 for their encouraging remarks and insightful comments**. In addressing the suggestions raised by Reviewer #1, we have made substantial additions to the manuscript, which we believe have significantly enhanced its overall quality.

1. Provide coulombic efficiency analysis of the cycling data. It's possible that there are some side reactivities and they may be quite different in the PS and NM samples, especially those made by changing the PH of the washing water.

We sincerely thank **Reviewer #1 for their critical recommendation**. In response, we **have included the coulombic efficiency (CE) data in the revised manuscript and Supplementary Information as needed**. Our results indicate that PS-LMTO (average 96.4% over 100 cycles) exhibits slightly higher CE than NM-LMTO (average 95.2% over 100 cycles) when cycled in the 70:20(MWCNT):10 electrode between 1.5–4.8 V at 20 mA/g (**Fig. 4c in the manuscript**).

We also included CE data for NM-LMTO (calcined at 800°C for 5 minutes and annealed at 600°C for 20 hours) washed with LiOH solutions of varying pH (7, 10, 12, and 14). **Response Figs. 1a-h** present the voltage profiles and CE of NM-LMTO in the 70:20(CB):10 electrode, prepared using a conventional LiOH washing process rather than a Li-reinsertion protocol.

The average CE of NM-LMTO electrodes over ten cycles (1.5–4.8 V at 20 mA/g) was 93.9%, 92.5%, 95.1%, and 92.7% for samples washed with pH 7 (DI water), 10, 12, and 14 solutions, respectively. For DRX materials, the first cycle CE is primarily influenced by Li diffusion kinetics, with the difference between first charge and discharge capacities driven more by kinetic limitations than by electrolyte decomposition or side reactions. Consequently, we assessed the impact of side reactions and electrolyte decomposition by averaging CE values from the second cycle onward. **Excluding the first cycle, the average CE for samples washed with pH 7 (DI water), 10, 12, and 14 solutions was 94.9%, 94.1%, 96.4%, and 95.5%, respectively**. These results reveal no clear correlation between CE and the pH of the LiOH washing solution,

although a higher pH might be expected to result in lower CE due to the potential presence of residual LiOH in the samples.

Response Figure 1 (a-d) Voltage profiles of NM-LMTO samples (calcined at 800°C for 5 minutes and annealed at 600°C for 20 hours) washed with LiOH solutions of varying pH: **(a)** pH 7 (DI water), **(b)** pH 10, **(c)** pH 12, and **(d)** pH 14. Cycling was performed between 1.5–4.8 V at 20 mA/g in electrodes with a 70:20(CB):10 composition. **(e-h)** Ten-cycle capacity retention and coulombic efficiency for the corresponding samples. **(i-k)** Voltage profiles of NM-LMTO samples (calcined at 800°C for 5 minutes and annealed at 600°C for 20 hours, with Li reinsertion after DIW washing of CsBr): **(i)** cycled between 1.5–4.8 V at 20 mA/g in a 70:20(MWCNT):10 electrode, **(j)** cycled under the same conditions in a 90:5(MWCNT):5 electrode, and **(k)** their corresponding capacity retention and coulombic efficiency.

Further CE analysis of the NM-LMTO electrode indicates that reducing the amount of MWCNT in the electrode (and thereby increasing the active material content) improves the CE. **Response Figs. 1i-k** illustrate the voltage profiles, capacity retention, and CE of (Li-reinserted) NM-LMTO in 70:20(MWCNT):10 and 90:5(MWCNT):5 electrodes, cycled between 1.5–4.8 V at 20 mA/g. **The average CE increases from 95.0% (averaged over 60 cycles) to 98.0% (averaged over 60 cycles) with higher active material content (lower MWCNT content).**

This observation suggests that side reactions between the MWCNT and electrolyte are the primary factor determining the CE. Building on this, we hypothesize that the slightly lower CE of NM-LMTO than PS-LMTO in the same 70:20(MWCNT):10 electrode may be due to the more uniform MWCNT distribution in the NM-LMTO electrode. This uniform distribution likely increase the contact area between MWCNT and the electrolyte, thereby promoting greater electrolyte decomposition and contributing to the observed difference in CE.

Improving CE is critical for DRX and other battery materials to ensure stable cycling performance. Accordingly, we have expanded the discussion on this topic in the revised

manuscript [Page 15, lines 14-20; Page 16, lines 1-10 & 13-17; Page 25, lines 11-21] and included these results in the Supplementary Information [Supplementary Figs. 11b-i & 12c-d].

2. For stability comparison, it would be useful to cycle the sample by setting capacity limit instead of voltage limit. For example, Figure 4 shows much faster capacity and voltage decay in the PS sample. However, the PS sample delivers a much higher initial capacity than that of the NM sample. That itself may lead to faster decay as the PS sample simply experiences deeper Li extraction and insertion.

We are deeply grateful for **Reviewer #1's valuable comments**. We agree that certain degrees of faster capacity and voltage degradation for PS-LMTO can originate from its much larger capacities in earlier cycles. However, to demonstrate that the improved capacity and voltage retention of NM-LMTO does not merely come from NM-LMTO's smaller initial capacities, we cycled the S-LMTO [70(S-LMTO):20(MWCNT):10(PVDF)] half-cells between 1.5-4.8 V at 20 mA/g and 25°C (**Response Figs. 2a and 2b**), which achieves similar capacities as NM-LMTO. S-LMTO (solid-state LMTO) went through a spex-mixing with MWCNT, during which carbon mixing occurs with some particle size reduction of S-LMTO (**Response Fig. 2e**). However, unlike PS-LMTO, which is heavily pulverized alone before mixing with MWCNT (**Response Fig. 2f**), S-LMTO's particle size after MWCNT mixing is still much bigger than PS-LMTO.

Response Figure 2 (a) Voltage profile, **(b)** capacity retention, and coulombic efficiency of S-LMTO cycled between 1.5–4.8 V at 20 mA/g in a 70:20(MWCNT):10 electrode. **(c)** Voltage profile, **(d)** capacity retention, and coulombic efficiency of PS-LMTO cycled between 1.5–4.4 V at 20 mA/g in a 70:20(MWCNT):10 electrode. SEM images of **(e)** S-LMTO after mixed with MWCNT using a spex mill and **(f)** PS-LMTO (no carbon mixed).

Response Figures 2a and 2b illustrate the voltage profiles and capacity retention of S-LMTO. **The initial capacities during the early cycles (~220 mAh/g) are now similar to those of NM-LMTO.** However, **S-LMTO exhibits poorer capacity retention**, delivering ~162 mAh/g at the 80th cycle, inferior to NM-LMTO that delivers ~195 mAh/g and ~187 mAh/g at the 80th and

100th cycle, respectively. These results indicate that the **capacity retention of DRX materials is influenced by multiple factors beyond the initial capacity.**

However, **we acknowledge that the capacity retention of PS-LMTO could be improved by reducing capacity utilization.** For example, lowering the upper cut-off voltage from 4.8 V to 4.4 improves the 100-cycle capacity retention of PS-LMTO cycled at 20 mA/g from 38.6% to 73.3%. This improvement is likely due to reduced capacity utilization and minimized side reactions with the electrolyte at the lower cut-off voltage (**Response Figs. 2c and 2d**). We have incorporated these additional discussions in the revised manuscript [**Page 16, lines 20-29**] and included the new data in the revised Supplementary Information [**Supplementary Figs. 13c, 13d, and 15**].

3. Evaluation and discussion on optimal DRX particle size. Although it's easy to see the benefit of uniform particle size, it isn't clear what size is best for DRX. Small particles can improve kinetics but large particles improves packing density and reduces side reactions at high voltages. Large particles also eliminate the need of excessive amount of carbon in electrode fabrication.

We greatly appreciate **Reviewer #1's insightful comments regarding the optimal particle size for DRX materials.** While it would be ideal to provide a definitive discussion on this topic, identifying the precise optimal particle size remains challenging. This difficulty arises primarily from the limited understanding of DRX materials' kinetic properties (*e.g.*, Li diffusivity and electrical conductivity), thermodynamic characteristics (*e.g.*, operating voltage, redox mechanism, and thermal stability), and the intended target applications, whether competing with LiFePO₄ in lower energy density applications or with Li-Ni-Mn-Co oxides for high-energy Li-ion batteries.

Compounding this complexity is the dependence of **DRX's intrinsic kinetic properties**, such as Li diffusivity and electrical conductivity, **on its composition.** As a result, the optimal particle size varies with DRX composition. For instance, recent studies have demonstrated that Mn-rich DRX materials can transform into a spinel-like "σ phase," which exhibits significantly higher bulk Li diffusion properties (*Nat. Energy* 9, 27–36, 2024; *Adv. Energy Mater* 2200426, 2022). Additionally, Mn-rich DRX compositions generally show enhanced electrical conductivity. For such phases, larger particle sizes may be viable without incurring substantial capacity loss.

These intricate correlations between DRX composition and properties make it challenging to propose a universally applicable optimal particle size for all DRX materials. In response to the reviewer's insightful query, we have expanded our discussion on this topic in the manuscript while avoiding speculative assertions [**Page 24, lines 26-31**].

4. Would the success of this synthesis procedure also apply to other DRX compositions, especially those with higher Mn content ($Mn > 0.7$) that of great interest in the community? The authors should offer some initial data on that front or at least comment on what challenges may arise. Currently, the results are solely on the $Li_{1.2}Mn_{0.4}Ti_{0.4}O_2$ composition.

Response Figure 3 XRD patterns of (a) $NM-Li_{1.1}Mn_{0.7}Ti_{0.2}O_2$ (NM-LMTO-70: 900°C for 1 minute calcination & 600°C for 5 hours annealing in argon), (b) $NM-Li_{1.2}Mn_{0.6}Nb_{0.2}O_2$ (NM-LMNO: 950°C for 3 minutes calcination & 600°C for 5 hours annealing in argon), and (c) $NM-Li_{1.2}Ni_{0.2}Ti_{0.6}O_2$ (NM-LNTO, 750°C for 5 minutes calcination & 600°C for 5 hours annealing in air). These samples underwent Li-reinsertion after the DIW washing of CsBr. ICP results of (d) NM-LMTO-70, (e) NM-LMNO, and (f) NM-LNTO. The SEM images of the (g) NM-LMTO-70, (h) NM-LMNO, and (i) NM-LNTO particles. The particle size distribution of the (j) NM-LMTO-70, (k) NM-LMNO, and (l) NM-LNTO particles.

We sincerely appreciate **Reviewer #1** for their constructive suggestion to apply our method on other DRX compositions. In response, we synthesized $Li_{1.1}Mn_{0.7}Ti_{0.2}O_2$ (NM-LMTO-70), $Li_{1.2}Mn_{0.6}Nb_{0.2}O_2$ (NM-LMNO), and $Li_{1.2}Ni_{0.2}Ti_{0.6}O_2$ (NM-LNTO) using the NM-method. NM-

LMTO-70 was prepared via calcination at 900°C for 1 minute in argon, followed by annealing at 600°C for 5 hours in argon. Similarly, NM-LMNO was calcined at 950°C for 3 minutes in argon and annealed at 600°C for 5 hours in argon. NM-LNTO was calcined at 750°C for 5 minutes in air, followed by annealing at 600°C for 5 hours in air. In all experiments, the precursor-to-salt ratio was 1:1.5, and Li reinsertion was applied after DIW washing of the CsBr salt.

Response Figure 3 presents the XRD patterns, ICP results, SEM images, and particle size distributions for the NM-compounds. XRD results confirm their formation as DRX materials (**Response Figs. 3a-c**). The ICP results show the Li:Mn:Ti ratio of **1.118:0.688:0.193** for the NM-LMTO-70 (target, 1.1:0.7:0.2), the Li:Mn:Nb ratio of **1.185:0.597:0.218** for the NM-LMNO (target, 1.2:0.6:0.2), and the Li:Ni:Ti ratio of **1.218:0.189:0.592** for the NM-LNTO (target, 1.2:0.2:0.6) (**Response Figs. 3d-f**).

The average particle sizes for NM-LMTO-70, NM-LMNO, and NM-LNTO, obtained under the given NM-synthesis conditions, are **~307 nm (standard deviation ~119 nm)**, **~420 nm (standard deviation ~179 nm)**, and **~36 nm (standard deviation ~6 nm)**, respectively (**Response Figs. 3g-l**). The high calcination temperatures for NM-LMTO-70 (900°C) and NM-LMNO (950°C) compared to NM-LMTO in the main manuscript (800°C) result in larger particle sizes than NM-LMTO (~118 nm, standard deviation ~35 nm). Further optimization of NM synthesis parameters, such as the precursor-to-salt ratio (not explored due to revision time constraints), may reduce particle size and improve consistency across compositions. Despite limited optimization, NM-LMTO-70 and NM-LMNO exhibit some of the smallest particle sizes reported for their compositions, if not the smallest. Notably, NM-LNTO achieves a pure DRX phase at ~750°C, resulting in exceptionally small particles (~36 nm), demonstrating the NM method's capability for particle size control. For reference, SEM images of (as-made) LMTO-70, LMNO, and LNTO with identical compositions from previous studies reveal substantially larger and more agglomerated particles (**Response Figs. 4**).

Response Figure 4. SEM images of (a) as-made LMTO-70, reproduced from Li et al., “Structural Evolution in Disordered Rock Salt Cathodes,” *J. Am. Chem. Soc.* **2024**, 146 (35), 24296–24309, under the terms of the Creative Commons CC BY-NC-ND 4.0 license (<https://creativecommons.org/licenses/by-nc-nd/4.0/>). No modifications were made; (b) as-made LMNO, reproduced from Lee et al., “Determining the Criticality of Li-Excess for Disordered-Rocksalt Li-Ion Battery Cathodes,” *Adv. Energy Mater.* **2021**, **11**, 2100204. Used with permission of John Wiley & Sons; permission conveyed through Copyright Clearance Center, Inc.; and (c) as-made LNTO, reproduced from Zhou et al., “Effect of Surface Structure on Electrochemical Properties in $\text{Li}_{1.2}\text{Ni}_{0.2}\text{Ti}_{0.6}\text{O}_2$ Cathode Material,” *J. Mater. Sci.* **2023**, **58**, 12345–12360. Reproduced with permission from Springer Nature.

The required calcination temperature for DRX synthesis (preferably lower for smaller particles) varies by composition due to the thermodynamic stability of the DRX phase relative to competing phases. **Higher temperatures are often necessary to stabilize DRX phases with higher Mn content**, irrespective of the synthesis method, complicating the production of smaller particles. This underscores the need for further investigation into the thermodynamic stability of DRX compositions to identify conditions that minimize calcination and annealing temperatures while preventing impurity formation or phase decomposition.

We have incorporated these additional discussions in the revised manuscript [**Page 8, lines 5-22; Page 22, lines 12-27**] and included the new data in the revised Supplementary Information [**Supplementary Figs. 6 and 7**].

Response to Reviewer #2

This paper investigates the synthesis of Mn-DRX cathode active materials using molten salt and grinding methods and compares their electrochemical characteristics. The most notable difference observed in this study is that particles synthesized with molten salt exhibit more uniform sizes compared to those synthesized by grinding.

We express our sincere gratitude to Reviewer #2 for their critical review of our manuscript. In response to their concerns, we have made substantial additions to the manuscript, aiming to significantly enhance its quality.

Due to the inherently low conductivity of the material itself, a considerable amount of conductive carbon of 20 wt.% versus the active materials was added for testing purposes. Figure 4 presents the electrochemical properties measured with MWCNT of 20 wt.% versus the active materials. These carbon additives are too much to fabricate the electrode and not suitable in terms of side reactions with electrolyte on high voltage region.

We greatly appreciate Reviewer #2 for their valuable comments. The inherently low conductivity of various Mn-DRX materials often necessitates the use of a significant amount of conductive carbon. In a recent study, we demonstrated that replacing carbon black with MWCNT can reduce the required amount of conductive additives while improving the electrical percolation network in Mn-DRX electrodes (*Energy Environ. Sci.*, 2024, 17, 3753-3764), thereby mitigating their intrinsically low conductivity. Following this approach, we demonstrated the performance of NM-LMTO and PS-LMTO in electrodes containing 20 wt% MWCNT in the manuscript to ensure that electrical conductivity does not become a limiting factor for material performance.

Response Figure 5 (a,b) Voltage profile (a) and capacity retention (b) of NM-LMTO (90:5(MWCNT):5 electrode) when cycled between 1.5-4.8 V at 20 mA/g. (c,d) The voltage profile (c) and capacity retention (d) of PS-LMTO (90:5(MWCNT):5 electrode) when cycled between 1.5-4.8 V at 20 mA/g.

However, we fully acknowledge Reviewer #2 that a high concentration of conductive additives can exacerbate side reactions with the electrolyte. To further investigate this issue, we prepared additional electrodes containing 5 wt% MWCNT for both NM- and PS-LMTO (*i.e.*, 90:5:5 electrode composition) and evaluated their performance.

Response Figure 5 compares voltage profiles, capacity retention, and coulombic efficiency of NM- and PS-LMTO in the 90:5:5 electrode configuration. While the higher active material content leads to a reduction in reversible capacity for both NM- and PS-LMTO, the NM-LMTO cell (85.3% over 60 cycles) maintains significantly better capacity retention than the PS-LMTO (46.8% over 60 cycles), consistent with observations from the 70:20:10 electrode. Moreover, the average coulombic efficiency of the NM-LMTO electrode improves from 95.0% (averaged over 60 cycles, Fig. 4c) to 98.0% (averaged over 60 cycles) with higher active material content (lower MWCNT content). These findings suggest that excessive MWCNT content can accelerate the electrolyte decomposition, as Reviewer #2 pointed out.

Therefore, decreasing the MWCNT content would be beneficial for electrode fabrication. We have incorporated this discussion in the revised manuscript [Page 14, lines 22-27; Page 16, lines 13-17] and Supplementary Information [Supplementary Figs. 12a-d].

The material discussed has been extensively researched over a considerable period, and changes in properties based on synthetic methods are already well-documented. A similar study was published in Journal of Alloys and Compounds 797 (2019) 961-969, prompting questions regarding the originality and novelty of this paper. Additionally, it is likely that the differences in particle formation by synthetic methods would lead to variations in electrochemical reaction mechanisms.

We greatly appreciate Reviewer #2's comments and the opportunity to clarify the originality and scope of our work.

The primary focus of our manuscript is the development of a method to produce well-crystallized, morphology-controlled single particles of DRX materials. We deliberately selected $\text{Li}_{1.2}\text{Mn}_{0.4}\text{Ti}_{0.4}\text{O}_2$ (LMTO) as an example, as no prior study has demonstrated the direct synthesis of small single particles ($d < 200$ nm). Instead, previous approaches have relied on particle pulverization via ball milling to achieve suitable sizes for electrochemical cycling.

The paper referenced by Reviewer #2, now cited in our revised manuscript, employs the same ball milling approach to reduce the size of large Li-Mn-Nb-O DRX particles ($5 \mu\text{m} < d < 10 \mu\text{m}$) synthesized via solid-state methods (Response Figure 6a). Rather than challenging the originality of our work, this actually reinforces the significance of our approach by highlighting the field's reliance on solid-state synthesis followed by mechanical size reduction. Moreover, the cited study focuses primarily on compositional effects of the Li-Mn-Nb-O DRX system ($x\text{Li}_3\text{NbO}_4 - (1-x)\text{LiMnO}_2$, $0.2 < x < 0.5$) on cycling performance, without emphasis on particle morphology or synthesis methodology (Response Figure 6b). This further distinguishes our

study, which prioritizes **precise control over particle size and morphology through a novel synthetic route**.

Response Figure 6 The paragraphs cited in the reference provided by Reviewer #2 (*Journal of Alloys and Compounds* 797 (2019) 961–969) include: (a) a paragraph from the Results and Discussion Section 3.1 and (b) paragraphs from the Introduction Section of the manuscript. Screenshot of text excerpt reproduced from Fan et al., “Synthesis and Electrochemical Performance of Li₃NbO₄-Based Cation-Disordered Rock-Salt Cathode Materials for Li-Ion Batteries,” *J. Alloys Compd.* 2019, 797, 961–969. Reprinted with permission from Elsevier.

To further emphasize that our work demonstrates **the universal applicability of the NM synthesis approach**, rather than being limited to LMTO, we have additionally synthesized: **Li_{1.1}Mn_{0.7}Ti_{0.2}O₂ (LMTO-70)**, **Li_{1.2}Mn_{0.6}Nb_{0.2}O₂ (LMNO)**, and **Li_{1.2}Ni_{0.2}Ti_{0.6}O₂ (LNTO)**. NM-LMTO-70 was prepared via calcination at 900°C for 1 minute in argon, followed by annealing at 600°C for 5 hours in argon. Similarly, NM-LMNO was calcined at 950°C for 3 minutes in argon and annealed at 600°C for 5 hours in argon. NM-LNTO was calcined at 750°C for 5 minutes in air, followed by annealing at 600°C for 5 hours in air. In all experiments, the precursor-to-salt ratio was 1:1.5, and Li reinsertion was applied after DIW washing of the CsBr salt.

Response Figure 7 presents the XRD patterns, ICP results, SEM images, and particle size distributions for these NM compounds. XRD results confirm their formation as DRX materials (**Response Figs. 7a-c**). The ICP results show the Li:Mn:Ti ratio of **1.118:0.688:0.193 for the NM-LMTO-70** (target, 1.1:0.7:0.2), the Li:Mn:Nb ratio of **1.185:0.597:0.218 for the NM-LMNO** (target, 1.2:0.6:0.2), and the Li:Ni:Ti ratio of **1.218:0.189:0.592 for the NM-LNTO** (target, 1.2:0.2:0.6) (**Response Figs. 7d-f**).

The average particle sizes for NM-LMTO-70, NM-LMNO, and NM-LNTO, obtained under the given NM-synthesis conditions, are **~307 nm (standard deviation ~119 nm)**, **~420 nm (standard deviation ~179 nm)**, and **~36 nm (standard deviation ~6 nm)**, respectively (**Response Figs. 7g-l**). The high calcination temperatures for NM-LMTO-70 (900°C) and NM-LMNO (950°C) compared to NM-LMTO in the main manuscript (800°C) result in larger particle sizes than NM-LMTO (~118 nm, standard deviation ~35 nm). Further optimization of NM synthesis parameters, such as the precursor-to-salt ratio (not explored due to revision time constraints), may reduce particle size and improve consistency across compositions. **Despite limited optimization**, NM-LMTO-70 and NM-LMNO exhibit some of the **smallest particle sizes** for their compositions. Notably, NM-LNTO achieves a pure DRX phase at ~750°C, resulting in exceptionally small particles (~36 nm), demonstrating the **NM method's capability for particle size control**.

For reference, SEM images of (as-made) LMTO-70, LMNO, and LNTO with identical compositions from previous studies reveal substantially larger and more agglomerated particles (Response Figs. 8).

These additional examples further confirm that our NM synthesis method enables the **direct synthesis of small, single-particle DRX materials with controlled morphology**, distinguishing our work from conventional solid-state and ball milling approaches. We have now incorporated this discussion into the revised manuscript [Page 8, lines 5-22] and Supplementary Information [Supplementary Figs. 6 and 7].

Response Figure 7 XRD patterns of (a) NM-Li_{1.1}Mn_{0.7}Ti_{0.2}O₂ (NM-LMTO-70: 900°C for 1 minute calcination & 600°C for 5 hours annealing in argon), (b) NM-Li_{1.2}Mn_{0.6}Nb_{0.2}O₂ (NM-LMNO: 950°C for 3 minutes calcination & 600°C for 5 hours annealing in argon), and (c) NM-Li_{1.2}Ni_{0.2}Ti_{0.6}O₂ (NM-LNTO, 750°C for 5 minutes calcination & 600°C for 5 hours annealing in air). These samples underwent Li-reinsertion after the DIW washing of CsBr. ICP results of (d) NM-LMTO-70, (e) NM-LMNO, and (f) NM-LNTO. The SEM images of the (g) NM-LMTO-70, (h) NM-LMNO, and (i) NM-LNTO particles. The particle size distribution of the (j) NM-LMTO-70, (k) NM-LMNO, and (l) NM-LNTO particles.

Response Figure 8 SEM images of (a) as-made LMTO-70, reproduced from Li et al., “Structural Evolution in Disordered Rock Salt Cathodes,” *J. Am. Chem. Soc.* **2024**, 146 (35), 24296–24309, under the terms of the Creative Commons CC BY-NC-ND 4.0 license (<https://creativecommons.org/licenses/by-nc-nd/4.0/>). No modifications were made; (b) as-made LMNO, reproduced from Lee et al., “Determining the Criticality of Li-Excess for Disordered-Rocksalt Li-Ion Battery Cathodes,” *Adv. Energy Mater.* **2021**, **11**, 2100204. Used with permission of John Wiley & Sons; permission conveyed through Copyright Clearance Center, Inc.; and (c) as-made LNTO, reproduced from Zhou et al., “Effect of Surface Structure on Electrochemical Properties in $\text{Li}_{1.2}\text{Ni}_{0.2}\text{Ti}_{0.6}\text{O}_2$ Cathode Material,” *J. Mater. Sci.* **2023**, **58**, 12345–12360. Reproduced with permission from Springer Nature.

Detailed mechanism of XRD pattern changes resulting from oxygen release is necessary in this work. Moreover, the paper lacks discussion on oxygen redox and TM (transition metal) redox processes as they relate to particle transformations. For practical applications, methods for suppressing Mn dissolution and results from a full cell paired with a graphite anode should be included.

We sincerely appreciate **Reviewer #2’s valuable comments and the opportunity to address them**. Our manuscript primarily focuses on **demonstrating a novel nucleation-promoting and growth-limiting synthesis method applicable to various DRX compounds**, including LMTO, LMTO-70, LMNO, and LNTO. We believe that a **detailed investigation into the redox processes, oxygen release, or XRD pattern changes of an individual DRX compound falls beyond the scope of this study**. These aspects have been extensively examined in prior research by various groups, including our own. Relevant contributions from our group include:

- Lee*# et al., *Nature* 556, 185-190, 2018 (* first author, # corresponding)
- Seo*, Lee* et al., *Nat. Chem.* 8, 692-697, 2016
- Lun, Lee#, Ceder# et al., *Adv. Energy Mater.* 182959, 2019
- Kwon*, Lee* et al., *Cell Rep. Phys. Sci.* 100187, 2020
- Li, Seo, Lee# et al., *Joule* 6, 53-91, 2022

However, to **accommodate the Reviewer’s request**, we have performed **additional soft X-ray Absorption Spectroscopy (XAS) analysis** on NM- and PS-LMTO to investigate their redox processes.

Response Figure 9a shows the **Mn L₃-edge XAS spectra of the NM-LMTO electrode** collected in bulk-sensitive Partial Fluorescence Yield (PFY) mode: before cycling (Pr), after the 1st charge to 100 mAh/g (100C), 200 mAh/g (200C), and 4.8 V (TOC1), and after the 1st discharge

to 1.5 V (EOD1) at 20 mA/g. For reference, Mn L₃-edge spectra of MnO [Mn(II)], Mn₂O₃ [Mn(III)], and MnO₂ [Mn(IV)] are also included. The Mn L₃-edge spectrum of NM-LMTO before cycling (Pr) closely resembles that of Mn₂O₃ (Mn(III)), indicating an initial Mn oxidation state near Mn³⁺, consistent with its composition (Li⁺_{1.2}Mn³⁺_{0.4}Ti⁴⁺_{0.4}O²⁻₂). Upon the 1st charge to 100 mAh/g (100C) and 200 mAh/g (200C), the center of mass of the Mn L₃-edge shifts to higher energy, signifying Mn oxidation toward Mn⁴⁺. Further charging to 4.8 V (TOC1) results in minimal change to the Mn edge, suggesting that additional capacity beyond this point involves negligible Mn oxidation.

Response Figure 9 (a) Mn L₃-edge XAS spectra of the NM-LMTO electrode collected in Partial Fluorescence Yield (PFY) mode: before cycling (Pr), after the 1st charge to 100 mAh/g (100C), 200 mAh/g (200C), and 4.8 V (TOC1), and after the 1st discharge to 1.5 V (EOD1) at 20 mA/g. For reference, Mn L₃-edge spectra of MnO [Mn(II)], Mn₂O₃ [Mn(III)], and MnO₂ [Mn(IV)] are also included. (b) O K-edge XAS spectra of the NM-LMTO electrode collected in Partial Fluorescence Yield (PFY) mode: before cycling (Pr), after the 1st charge to 100 mAh/g (100C), 200 mAh/g (200C), and 4.8 V (TOC1), and after the 1st discharge to 1.5 V (EOD1) at 20 mA/g. (c,d) Mn L₃-edge XAS spectra of the NM-LMTO and PS-LMTO electrodes, collected in (c) Partial Fluorescence Yield (PFY) mode and (d) Total Electron Yield (TEY) mode, are shown: before cycling (Pr), after the 1st discharge (EOD1), and after the 100th discharge (EOD100) during 1.5–4.8 V cycling at 20 mA/g.

It is widely understood that along with Mn oxidation, O oxidation takes place in LMTO (and other Li-excess Mn-DRX compounds) during charging. **Response Figure 9b** presents the

pre-edge feature of the O K-edge XAS spectra of NM-LMTO collected in PFY mode, confirming O oxidation during charging. Upon the 1st charge to 100 mAh/g (100C), 200 mAh/g (200C), and 4.8 V (TOC1), an increasing intensity is observed between 527–529 eV, associated with Mn oxidation to Mn⁴⁺. Concurrently, an intensity increase between 530–532 eV is detected, which is often attributed to O oxidation (*Nature* 556, 185-190, 2018; *Proc. Natl. Acad. Sci. U.S.A.* 112 (25) 7650-7655, 2015). After the 1st discharge to 1.5 V, the pre-edge feature of the O K-edge spectra returns to a state similar to that before cycling.

This overall evolution in the Mn L₃-edge and O K-edge spectra demonstrates the oxidation of Mn and O during charging for NM-LMTO, which reverses upon discharging. This behavior is consistent with previous reports on LMTO in the literature. Due to limited time at the XAS facility, we did not conduct a separate analysis of Mn and O oxidation changes for PS-LMTO during the first cycle, as it was synthesized using the traditional method reported in the literature. However, numerous studies support a similar redox mechanism for such systems.

In parallel, we compared the Mn electronic states of NM-LMTO and PS-LMTO at different stages using both bulk-sensitive Partial Fluorescence Yield (PFY) mode and surface-sensitive Total Electron Yield (TEY) mode: before cycling (Pr), after the 1st discharge (EOD1), and after the 100th discharge (EOD100) (**Response Figs. 9c and 9d**). The Mn L₃-edge spectra from both modes reveal a shift in the center of mass of the Mn edge toward lower energy after the first cycle, with a further shift after 100 cycles. This indicates a progressive decrease in the Mn oxidation state in the discharged state, from Mn³⁺ toward Mn²⁺ during cycling. However, this shift progresses more slowly in NM-LMTO for both modes. Notably, the PFY mode shows a stronger intensity between 641–643 eV after cycling, suggesting a higher retention of Mn³⁺ in the bulk of NM-LMTO compared to PS-LMTO.

The introduction of lower-valent Mn²⁺ in Mn³⁺-DRX systems, such as LMTO, is well-documented and primarily attributed to O loss during cycling (*Adv. Energy Mater.* 2019, 9, 1901255). Our XAS results confirm that NM-LMTO experiences less O loss and consequently shows reduced surface and bulk Mn reduction compared to PS-LMTO. **This observation is consistent with the XPS results presented in Figures 5e and 5f of the main manuscript.**

We note that Reviewer #2 also inquired about the XRD pattern changes associated with O loss. However, the (i) broadening of XRD peaks after cycling, (ii) potential inhomogeneity in the degree of reaction across different particles, and (iii) the presence of numerous refinement factors that can lead to similarly good fits make it highly challenging to extract detailed structural information from lab-based XRD refinement. Given these limitations, **lattice parameters** remain one of the most **reliable structural indicators** for cubic rock-salt phases. Accordingly, we have presented the XRD results in **Figures 5c and 5d**, which reveal a **greater increase in the lattice parameter for PS-LMTO compared to NM-LMTO.**

As elaborated in the manuscript, this lattice parameter increase is attributed to the introduction of larger Mn²⁺ cations (which are larger than Mn³⁺) into LMTO following O loss. The

greater lattice expansion observed in PS-LMTO after 40 and 100 cycles suggests a higher presence of Mn^{2+} , indicating more significant oxygen loss compared to NM-LMTO. This observation aligns with the XAS analysis provided earlier.

Additionally, further **oxygen occupancy refinements (Supplementary Figs. 16 and 17)** reveal a **faster and greater decrease in oxygen occupancy** for PS-LMTO than NM-LMTO, confirming its **higher oxygen loss**. While oxygen occupancy refinement has inherent limitations, these trends are consistent across all analyses **[Page 20, lines 30-31; Page 21, lines 2-1]**.

Regarding practical applications, we acknowledge the importance of **(i) suppressing Mn dissolution** and **(ii) evaluating DRX cathodes in full cells**. However, these topics require **detailed independent investigations** that fall **outside the scope of this manuscript**. Our study aims to **establish a fundamentally new synthesis method** applicable to various DRX compositions, rather than focus on **cell-level engineering**. Indeed, studies addressing the inhibition of Mn dissolution, such as through electrolyte engineering or metal doping, have already been published by our group (*Adv. Mater.* 2208423, 2023) and others (*Adv. Energy Mater.* 2304074, 2024). Additionally, we have demonstrated full cells using Mn-DRX cathode materials and graphite anodes in a separate paper (*Energy Environ. Sci.* 17, 3753-3764, 2024).

While **full-cell studies** are not a primary focus of this work, we recognize their importance and **have highlighted this need** in the revised manuscript **[Page 25, lines 11-21]**. Additionally, the **new XAS data is included as Supplementary Figure 18 and Supplementary Note 1 [mentioned on Page 21, lines 15-17 in the main manuscript as well]**.

In the AC impedance section, there is a significant difference in post-cycling between the NM and PS powders, likely attributable to the impact of Li dendrites on the lithium anode. To completely rule out such factors, impedance measurements using a three-electrode cell are recommended.

We appreciate **Reviewer #2's recommendation** and acknowledge that **impedance growth in a cell arises from multiple sources**, including both the cathode and anode, rather than solely from differences in the cathode materials (NM vs. PS-LMTO). Accordingly, we have not attributed the impedance increase solely to the cathode in either the **manuscript or Supplementary Information (SI)**. Instead, we have clarified that it originates from **both cathode and anode contributions**, which are **modeled using distinct fitting components** in an established equivalent circuit model (**Supplementary Fig. 21**). **In the revised manuscript, we further emphasize these contributions to impedance growth [Page 21, lines 21-27]**.

To isolate the impedance growth at the cathode, **we additionally conducted symmetric cell analyses using electrodes from cycled PS-LMTO||Li and NM-LMTO||Li cells**. While Reviewer #2 suggested three-electrode EIS tests, our non-ideal three-electrode setup resulted in interference from the Li-deposited Cu reference electrode, leading to inconsistent results. Therefore, we opted for symmetric cell analysis to separately assess cathode and anode impedance.

Our approach involved fabricating two symmetric cells for each material: an anode symmetric cell (Li||Li) and a cathode symmetric cell (Cathode||Cathode). We first measured the initial impedance of these cells before cycling. Then, we disassembled the symmetric cells in an argon-filled glove box and reassembled them as half-cells by pairing each electrode (Li or cathode) with a fresh counterpart. These half-cells were cycled for 10 cycles, after which they were disassembled again, and the electrodes were swapped back to reconstruct the symmetric cells. Finally, we remeasured the impedance to evaluate changes in each electrode's contribution after cycling.

Response Figure 10 presents the Nyquist plots of the Cathode||Cathode and Li||Li symmetric cells. After 10 cycles in the PS/NM-LMTO||Li cells, **the total resistance ($R_e + R_{film} + R_{ct}$) of the PS-LMTO symmetric cell increased ~15.5-fold (from 128.61 Ω to 1994.89 Ω), whereas in the NM-LMTO symmetric cell, it increased ~10.4-fold (from 87.69 Ω to 908.78 Ω). Since these are symmetric cells, the observed impedances can be fully attributed to the PS- or NM-LMTO cathode (with impedance doubling due to the identical electrodes). The lower initial impedance in the NM-LMTO symmetric cell and its smaller increase after cycling support our conclusion that the NM-LMTO electrode exhibits greater stability against impedance growth than the PS-LMTO electrode.**

Response Figure 10 (a, f) The equivalent circuits model used to fit the Nyquist plots of the NM- and PS-LMTO anode symmetric cells and cathode symmetric cells, respectively. **(b-e)** Nyquist plots of the NM-LMTO and PS-LMTO cathode symmetric cells **(b, d)** before cycling and **(c, e)** after 10 cycles between 1.5-4.8 V at 20 mA/g. **(g-j)** Nyquist plots of the NM-LMTO and PS-LMTO anode symmetric cells **(g, i)** before cycling and **(h, j)** after 10 cycles between 1.5-4.8 V at 20 mA/g.

For the Li||Li symmetric cells derived from the PS-LMTO||Li and NM-LMTO||Li cells, impedance decreased after 10 cycles, likely due to the electrochemical decomposition of the passivation layer on the Li chips. However, this does not contradict the impedance growth observed in **Supplementary Figure 21**, as the symmetric cell results reflect only 10 cycles, whereas **Supplementary Figure 21** presents data from 100-cycle tests. Over extended cycling, Li dendrite formation can significantly increase anode impedance, which is already accounted for in our equivalent circuit fitting.

Overall, our existing EIS results, along with the new symmetric cell EIS data, consistently support our conclusion that the **NM-LMTO electrode exhibits superior stability against impedance growth compared to the PS-LMTO electrode**. We have now included the new symmetric cell EIS results as **Supplementary Figure 22 and Supplementary Note 2**.

Notably, this paper does not highlight a distinct originality or novelty in comparison to other similar studies, with many similarities observed, particularly to a recent publication (<https://doi.org/10.1038/s41560-023-01233-8>) despite compositional differences. Given these considerations, submission to a more specialized journal may be advisable.

We appreciate **Reviewer #2's comments** regarding the **novelty of our work** but respectfully disagree with the assessment that our study lacks distinctiveness compared to the **cited publication** or prior works.

First, **our manuscript already discusses the limitations of previous synthesis methods** for DRX materials, which typically produces **large particles requiring mechanical pulverization** for electrochemical cycling. However, **ball-milling introduces crystal defects and offers no control over particle morphology**, as demonstrated in our PS-LMTO example. Our work is the **first to directly synthesize small, single-particle DRX compounds without the need for pulverization**, an advancement **with significant implications for DRX commercialization**, as strongly recognized by **Reviewers #1, #3, and #4**.

Second, the only similarity between our work and the referenced study (*Nature Energy* 8, 482–491, 2023) is the general use of a molten-salt-related method. However, **their approach is fundamentally unrelated to Mn-DRX synthesis** and focuses on **producing large (>1 μm) single-crystal layered cathodes** using a **eutectic molten-salt flux to deagglomerate coprecipitated Ni-Mn-Co precursor particles** during mechanical mixing. This method is **not applicable to Mn-DRX synthesis** for at least two key reasons:

1. **Mn-DRX precursors are not coprecipitated**, so eutectic molten salts provide **no mixing advantage**, which is the **primary novelty of the cited work**.
2. **The low melting point of eutectic molten salts would accelerate calcination in Mn-DRX synthesis**, leading to **excessive particle growth**, which is **directly opposed** to our goal of achieving **sub-200 nm particles**.

We acknowledge that molten-salt synthesis has been applied in prior battery studies; however, our approach is fundamentally distinct in its goal and implementation. If the mere use of molten-salt methods were to determine novelty, this criterion would also apply broadly to prior works, including the cited study.

Given these **substantial methodological and objective differences**, we firmly believe that our work offers a **distinct and innovative contribution to DRX synthesis**. This perspective has also been **supported by Reviewers #1, #3, and #4, who recognized the significance of our approach**.

Response to Reviewer #3

Commercially available cathode materials for lithium-ion batteries typically contain significant amounts of nickel and cobalt. However, there is a growing demand for the development of nickel- and cobalt-free cathodes due to the limited availability of these resources and associated safety concerns. Manganese-based disordered rocksalt lithium-excess cathode materials (Mn-DRXs) have been proposed as viable alternatives to the established nickel- and cobalt-based layered oxides. Low intrinsic lithium diffusivity within Mn-DRX requires to make cathode particles small enough. This research proposes a method to synthesize highly crystalline, size-controlled single Mn-DRX particles, which exhibit higher capacity and average voltage retention compared to Mn-DRX prepared by solid state method and following pulverization. This work is a valuable demonstration that offers new insights into the more controllable production of DRX particles. However, the following points should be addressed prior to consideration for publication.

We sincerely thank **Reviewer #3 for their thoughtful and positive evaluation** of our manuscript, as well as **their insightful comments**. In response, we have made substantial revisions that we believe have significantly improved the overall clarity and quality of the manuscript.

1. Although the primary claim of this manuscript is control over particle size, the Mn-DRX particle size is only roughly described as sub-200 nm. To rigorously assess the controllability of the proposed synthesis method and the correlation between particle morphology and performance, a more comprehensive quantitative statistical analysis of Mn-DRX particle size across various preparation methods is recommended. This analysis may include the number of particles used for statistical assessment, average particle size, standard deviation.

We sincerely appreciate **Reviewer #3's valuable recommendation**. In response, we conducted statistical analyses of the primary particle size using SEM for the NM-LMTO samples calcined under the following conditions (**Response Fig. 11; Supplementary Fig. 5**):

- 900°C for 20 minutes (**Mean: ~701 nm, Standard Deviation: ~268 nm**)
- 900°C for 5 minutes (**Mean: ~429 nm, Standard Deviation: ~250 nm**)
- 800°C for 20 minutes (**Mean: ~266 nm, Standard Deviation: ~36 nm**)
- 800°C for 5 minutes (**Mean: ~118 nm, Standard Deviation: ~35 nm**)

After calcination, all samples underwent additional annealing at 600°C for 20 hours.

For the PS-LMTO samples, however, we encountered significant challenges in characterizing representative primary or secondary particle size distributions due to extensive particle pulverization and ambiguous secondary particle morphology. Consequently, statistical particle size distribution analysis was not conducted for the PS-LMTO samples, nor for the S-LMTO samples, whose particle size is clearly in the micrometer range.

Response Figure 11 The SEM images of (a) S-LMTO and (b) PS-LMTO. (c-f) The SEM images and particle size distribution plots of NM-LMTO calcined at (c) 900°C for 20 minutes, (d) 900°C for 5 minutes, (e) 800°C for 20 minutes, and (f) 800°C for 5 minutes: after the calcination, the NM-LMTO particles underwent annealing at 600°C for 20 hours in Argon. The precursor-to-salt (CsBr) weight ratio was 1:1.5.

To demonstrate the general applicability of our NM method, we also synthesized $\text{Li}_{1.1}\text{Mn}_{0.7}\text{Ti}_{0.2}\text{O}_2$ (NM-LMTO-70), $\text{Li}_{1.2}\text{Mn}_{0.6}\text{Nb}_{0.2}\text{O}_2$ (NM-LMNO), and $\text{Li}_{1.2}\text{Ni}_{0.2}\text{Ti}_{0.6}\text{O}_2$ (NM-LNTO) using the NM-method. NM-LMTO-70 was prepared via calcination at 900°C for 1 minute in argon, followed by annealing at 600°C for 5 hours in argon. Similarly, NM-LMNO was calcined at 950°C for 3 minutes in argon and annealed at 600°C for 5 hours in argon. NM-LNTO was calcined at 750°C for 5 minutes in air, followed by annealing at 600°C for 5 hours in air. In all experiments, the precursor-to-salt ratio was 1:1.5, and Li reinsertion was applied after DIW washing of the CsBr salt.

Response Figure 12 presents the XRD patterns, ICP results, SEM images, and particle size distributions for the NM-compounds. XRD results confirm their formation as DRX materials (**Response Figs. 12a-c**). The ICP results show the Li:Mn:Ti ratio of **1.118:0.688:0.193** for the NM-LMTO-70 (target, 1.1:0.7:0.2), the Li:Mn:Nb ratio of **1.185:0.597:0.218** for the NM-LMNO (target, 1.2:0.6:0.2), and the Li:Ni:Ti ratio of **1.218:0.189:0.592** for the NM-LNTO (target, 1.2:0.2:0.6) (**Response Figs. 12d-f**).

The average particle sizes for NM-LMTO-70, NM-LMNO, and NM-LNTO, obtained under the given NM-synthesis conditions, are **~307 nm (standard deviation ~119 nm)**, **~420 nm (standard deviation ~179 nm)**, and **~36 nm (standard deviation ~6 nm)**, respectively (**Response Figs. 12g-l**). Our statistical analysis highlights a significant reduction in particle size achieved through controlled NM synthesis.

For reference, SEM images of (as-made) LMTO-70, LMNO, and LNTO with identical compositions from previous studies reveal substantially larger and more agglomerated particles (**Response Figs. 13**). We have included the statistical analysis result for NM-LMTO and new LMTO-70, LMNO, and LNTO in the revised manuscript [**Page 7, lines 21-30; Page 8, lines 5-22**] and Supplementary Information [**Supplementary Figs. 5, 6, and 7**].

Response Figure 12 XRD patterns of **(a)** NM-Li_{1.1}Mn_{0.7}Ti_{0.2}O₂ (NM-LMTO-70: 900°C for 1 minute calcination & 600°C for 5 hours annealing in argon), **(b)** NM-Li_{1.2}Mn_{0.6}Nb_{0.2}O₂ (NM-LMNO: 950°C for 3 minutes calcination & 600°C for 5 hours annealing in argon), and **(c)** NM-Li_{1.2}Ni_{0.2}Ti_{0.6}O₂ (NM-LNTO, 750°C for 5 minutes calcination & 600°C for 5 hours annealing in air). These samples underwent Li-reinsertion after the DIW washing of CsBr. ICP results of **(d)** NM-LMTO-70, **(e)** NM-LMNO, and **(f)** NM-LNTO. The SEM images of the **(g)** NM-LMTO-70, **(h)** NM-LMNO, and **(i)** NM-LNTO particles. The particle size distribution of the **(j)** NM-LMTO-70, **(k)** NM-LMNO, and **(l)** NM-LNTO particles.

Response Figure 13 SEM images of (a) as-made LMTO-70, reproduced from Li et al., “Structural Evolution in Disordered Rock Salt Cathodes,” *J. Am. Chem. Soc.* **2024**, 146 (35), 24296–24309, under the terms of the Creative Commons CC BY-NC-ND 4.0 license (<https://creativecommons.org/licenses/by-nc-nd/4.0/>). No modifications were made; (b) as-made LMNO, reproduced from Lee et al., “Determining the Criticality of Li-Excess for Disordered-Rocksalt Li-Ion Battery Cathodes,” *Adv. Energy Mater.* **2021**, **11**, 2100204. Used with permission of John Wiley & Sons; permission conveyed through Copyright Clearance Center, Inc.; and (c) as-made LNTO, reproduced from Zhou et al., “Effect of Surface Structure on Electrochemical Properties in $\text{Li}_{1.2}\text{Ni}_{0.2}\text{Ti}_{0.6}\text{O}_2$ Cathode Material,” *J. Mater. Sci.* **2023**, **58**, 12345–12360. Reproduced with permission from Springer Nature.

2. Compositional inhomogeneity is presented after calcination (Figure 2a). Does crystalline heterogeneity exist between Ti-rich surface and Mn-rich core?

Response Figure 14 XRD patterns of NM-LMTO synthesized with different precursor-to-salt (CsBr) weight ratios (800°C-5 min calcination; 600°C-20 h annealing). An incomplete LMTO synthesis leaves orthorhombic LiMnO_2 as an impurity phase along with the remaining disordered rock-salt phase.

We sincerely thank **Reviewer #3** for their insightful question. We believe the observed compositional heterogeneity is related to the synthesis pathway of LMTO, which can also impact crystallinity, depending on the extent of synthesis completion.

When the LMTO synthesis is incomplete, it often results in an orthorhombic phase as an impurity, which XRD phase identification suggests to be orthorhombic LiMnO_2 , indicating a Mn-rich phase (though we cannot rule out the presence of Ti within this structure). This trend is evident in our results, as seen in **Supplementary Figures 1b and 2a (Response Fig. 14)**, which reveal the orthorhombic LiMnO_2 -like impurity phase along with the DRX phase, corresponding with an incomplete reaction.

From the synthesis point, this LiMnO_2 phase is an intermediate phase that reacts with the remaining Ti-rich rock-salt phase to form LMTO, necessitating metal diffusion between these phases. We believe that the STEM-EELS is capturing incomplete metal diffusion. In this process, the Ti-rich rock-salt phase seems to

encapsulate the Mn-rich phase to complete LMTO synthesis.

Due to variations in reaction completion among particles, we cannot conclusively state that this incompleteness in LMTO formation, linked to the compositional heterogeneity in our manuscript, consistently produces distinct crystal structures between the particle surface and core, particularly given that these phases can form solid solutions. However, our analysis suggests that particles undergoing limited reaction, possibly due to shorter calcination times, may develop a more DRX-like Ti-rich surface and a more orthorhombic-like Mn-rich core. We discuss this possibility in further detail in the revised manuscript [Page 10, lines 4-10].

3. It is recommended to check the scale bar in Figure 2a or figure caption. The given information does not match to the main text. Also, getting EELS map from micron-sized particles is not feasible.

We sincerely appreciate **Reviewer #3's careful identification of our mistake**. We have identified an error in the caption for Figure 2. **The scale bar should read 100 nm rather than 1 μm**. We have revised the figure caption accordingly. Additionally, the EELS map was obtained from the electrode lamella prepared using FIB.

4. Instead of Li-reinsertion via LiOH solution, is electrochemical lithiation a feasible approach? Implementing this method could potentially eliminating one step in DRX particle preparation.

We greatly appreciate **Reviewer #3's insightful question**. In our manuscript, the Li-reinsertion process we describe is a reversal of the Li⁺/H⁺ exchange process, which does not involve oxidation or reduction of the cathode particles, for example, as shown in the following reaction:

With this in mind, we would not expect a one-step electrochemical lithiation to restore the original Li content in DRX. For example, an electrochemical lithiation reaction would proceed as:

This approach does not effectively remove protons from the structure.

One alternative could be to extract protons via electrochemical charging, followed by lithiation of the DRX compound upon discharge. However, unless the cell is disassembled after proton extraction and reassembled for electrochemical lithiation, residual protons in the electrolyte post-extraction could increase acidity and impair cell performance. **We have revised our manuscript to emphasize the need for a more efficient method to either limit Li⁺/H⁺ exchange during the washing process or to reverse the Li⁺/H⁺ exchange through a novel Li-reinsertion technique [Page 23, lines 24-30].**

5. Authors attributed higher rate capability of PS-LMTO to smaller grains than NM-LMTO. However, the cross-sectional SEM image of it reveals significant agglomeration into large secondary particles, which is not ideal to deliver capacity at high rates. Please clarify this inconsistency.

We sincerely appreciate **Reviewer #3's critical question**. We agree that agglomerated particles are generally not ideal for achieving high-rate capacities, even with small grain sizes. However, due to active Li diffusion and transport through grain boundaries (either via solid-state Li diffusion or electrolyte penetration through polycrystalline secondary particles), it is understood that the effective Li transport distance in polycrystalline (agglomerated) Li-ion cathode particles is often quite shorter than the full diameter of the polycrystals (*Energy Environ. Sci.*, 2023, 16, 3847-3859). This distance depends on the depth of electrolyte penetration into the secondary particles or the effectiveness of solid-state Li diffusion along grain boundaries.

For example, in layered NMC cathode materials, polycrystalline NMC particles (with polycrystal diameters >10 μm and grain sizes around 300 nm) usually demonstrate higher capacities and rate capabilities than single-crystal NMC particles with similar or smaller diameters (<10 μm). Similarly, we argued that PS-LMTO, with smaller grain sizes (sub-50 nm), shows faster rate capability than NM-LMTO, which has larger grains (sub-200 nm), even though some PS-LMTO particles are agglomerated into larger secondary particles.

It should be noted that this does not imply that agglomerated particles achieve the same rate capability as non-agglomerated particles of the same grain size. When particles agglomerate, contact between conductive carbon and DRX particles, known for their poor electrical conductivity, is less uniform compared to well-dispersed grains mixed with carbon (*Energy Environ. Sci.*, 2024, 17, 3753-3764). This non-uniform contact increases charge-transfer and contact resistance. Such additional, non-Li-transport-related effects could explain the smaller differences in discharge capacity observed between NM-LMTO and PS-LMTO at very high rates (e.g., 1000 mA/g), where not only Li transport (mass-transport resistance), the primary rate-limiting factor for most DRX materials, but also electrical conduction effects (charge-transfer and contact resistance) play a critical role in achieved capacity.

We have revised our manuscript to address this valuable point raised by Reviewer #3 [**Page 17, lines 25-29; Page 18, lines 1-6**].

6. The clarify of cross-sectional SEM images in Figure 5a and b needs to be improved. They are (except the one from PS-LMTO after 40 cycles) too blur to see microstructures.

We sincerely appreciate **Reviewer #3's constructive suggestion**. In response, we have replaced several images in Figures 5a and 5b with higher-resolution versions (**Response Fig. 15**). Please note that the high-magnification images (150k magnification) were captured at the maximum magnification capability of our current SEM setup.

The enhanced image quality has provided new insights into the microstructure evolution observed after cycling. In addition to the previously discussed improvements, such as greater electrode homogeneity and reduced porosity growth in the NM-LMTO electrode, the high-magnification SEM images reveal significant pore formation within the secondary particles of PS-LMTO, which was not evident in earlier images. Furthermore, nanopores appear to form in the NM-LMTO single particles after cycling. These observations have been incorporated into our expanded discussion in the revised manuscript [Page 20, lines 3-8].

Response Figure 15 Updated SEM images in Figures 5a and 5b on the PS-LMTO and NM-LMTO electrodes.

Response to Reviewer #4

In this manuscript, Hoda Ahmed et al. reports a method to synthesize 200 nm-sized cation disordered rock salt (DRX) and studies the influencing factors for morphology control and electrochemical performance. It was found that using CsBr molten salt flux with a short-time calcination + lower temperature extended annealing + water washing + ion exchange procedure can produce 200 nm-sized particle-like LTMO. The significance of producing 200 nm-sized LTMO is that it potentially offers a possibility of mass production of DRX-based batteries. As DRX-based cathodes have been well-known for their electronic conductivities orders lower than those of layered cathodes, an extensive high-energy carbon milling process is needed to realize the electrochemical performance. The requirement of high-energy carbon milling intrinsically limits the batch size of DRX electrodes and basically make scaled-up DRX-based battery production impossible. This work can potentially provide an important progress in scaling up the production of DRX-based batteries. After reading through the manuscript, I have following concerns and think these concerns need to be addressed before publication on Nat Comm.

We sincerely thank **Reviewer #4 for their thoughtful remarks, detailed evaluation of our data, and sharp insights**. In response to the concerns raised, we have made substantial additions to the manuscript, which we believe have significantly enhanced its overall quality.

1. First of all, I think it would be better if the work can cover a broader range of DRXs, or at least Mn-based DRXs, instead of just one DRX compound, to prove the effectiveness of the method and increase the impact of the work. This will make this paper more suitable for a journal like Nat Comm. For example, fluorinated-DRXs are reported to have better cycle performance. Higher-valence d0 transition metal-based DRXs like Nb-based DRXs can offer Mn-rich DRX with more Mn-redox capacity contribution like $\text{Li}_{1.2}\text{Mn}_{0.6}\text{Nb}_{0.2}\text{O}_2$. If the method raised by the authors can be applied to these DRXs mentioned above, the work will be much more impactful and more suitable for Nat Comm.

We sincerely appreciate **Reviewer #4 for their constructive suggestion to apply our method on other DRX compositions**. In response, we synthesized $\text{Li}_{1.1}\text{Mn}_{0.7}\text{Ti}_{0.2}\text{O}_2$ (NM-LMTO-70), $\text{Li}_{1.2}\text{Mn}_{0.6}\text{Nb}_{0.2}\text{O}_2$ (NM-LMNO), and $\text{Li}_{1.2}\text{Ni}_{0.2}\text{Ti}_{0.6}\text{O}_2$ (NM-LNTO) using the NM-method. NM-LMTO-70 was prepared via calcination at 900°C for 1 minute in argon, followed by annealing at 600°C for 5 hours in argon. Similarly, NM-LMNO was calcined at 950°C for 3 minutes in argon and annealed at 600°C for 5 hours in argon. NM-LNTO was calcined at 750°C for 5 minutes in air, followed by annealing at 600°C for 5 hours in air. In all experiments, the precursor-to-salt ratio was 1:1.5, and Li reinsertion was applied after DIW washing of the CsBr salt.

Response Figure 16 presents the XRD patterns, ICP results, SEM images, and particle size distributions for the NM-compounds. XRD results confirm their formation as DRX materials (**Response Figs. 16a-c**). The ICP results show the Li:Mn:Ti ratio of **1.118:0.688:0.193 for the NM-LMTO-70** (target, 1.1:0.7:0.2), the Li:Mn:Nb ratio of **1.185:0.597:0.218 for the NM-**

LMNO (target, 1.2:0.6:0.2), and the Li:Ni:Ti ratio of **1.218:0.189:0.592** for the NM-LNTO (target, 1.2:0.2:0.6) (**Response Figs. 16d-f**).

Response Figure 16 XRD patterns of **(a)** NM-Li_{1.1}Mn_{0.7}Ti_{0.2}O₂ (NM-LMTO-70: 900°C for 1 minute calcination & 600°C for 5 hours annealing in argon), **(b)** NM-Li_{1.2}Mn_{0.6}Nb_{0.2}O₂ (NM-LMNO: 950°C for 3 minutes calcination & 600°C for 5 hours annealing in argon), and **(c)** NM-Li_{1.2}Ni_{0.2}Ti_{0.6}O₂ (NM-LNTO, 750°C for 5 minutes calcination & 600°C for 5 hours annealing in air). These samples underwent Li-reinsertion after the DIW washing of CsBr. ICP results of **(d)** NM-LMTO-70, **(e)** NM-LMNO, and **(f)** NM-LNTO. The SEM images of the **(g)** NM-LMTO-70, **(h)** NM-LMNO, and **(i)** NM-LNTO particles. The particle size distribution of the **(j)** NM-LMTO-70, **(k)** NM-LMNO, and **(l)** NM-LNTO particles.

The average particle sizes for NM-LMTO-70, NM-LMNO, and NM-LNTO, obtained under the given NM-synthesis conditions, are **~307 nm (standard deviation ~119 nm)**, **~420 nm (standard deviation ~179 nm)**, and **~36 nm (standard deviation ~6 nm)**, respectively (**Response Figs. 16g-l**). The high calcination temperatures for NM-LMTO-70 (900°C) and NM-LMNO (950°C) compared to NM-LMTO in the main manuscript (800°C) result in larger particle

sizes than NM-LMTO (~118 nm, standard deviation ~35 nm). Further optimization of NM synthesis parameters, such as the precursor-to-salt ratio (not explored due to revision time constraints), may reduce particle size and improve consistency across compositions. Despite limited optimization, NM-LMTO-70 and NM-LMNO exhibit some of the smallest particle sizes reported for their compositions, if not the smallest. Notably, NM-LNTO achieves a pure DRX phase at ~750°C, resulting in exceptionally small particles (~36 nm), demonstrating the NM method's capability for particle size control. For reference, SEM images of (as-made) LMTO-70, LMNO, and LNTO with identical compositions from previous studies reveal substantially larger and more agglomerated particles (**Reponse Figs. 17**).

Response Figure 17 SEM images of (a) as-made LMTO-70, reproduced from Li et al., “Structural Evolution in Disordered Rock Salt Cathodes,” *J. Am. Chem. Soc.* **2024**, 146 (35), 24296–24309, under the terms of the Creative Commons CC BY-NC-ND 4.0 license (<https://creativecommons.org/licenses/by-nc-nd/4.0/>). No modifications were made; (b) as-made LMNO, reproduced from Lee et al., “Determining the Criticality of Li-Excess for Disordered-Rocksalt Li-Ion Battery Cathodes,” *Adv. Energy Mater.* **2021**, **11**, 2100204. Used with permission of John Wiley & Sons; permission conveyed through Copyright Clearance Center, Inc.; and (c) as-made LNTO, reproduced from Zhou et al., “Effect of Surface Structure on Electrochemical Properties in $\text{Li}_{1.2}\text{Ni}_{0.2}\text{Ti}_{0.6}\text{O}_2$ Cathode Material,” *J. Mater. Sci.* **2023**, **58**, 12345–12360. Reproduced with permission from Springer Nature.

The required calcination temperature for DRX synthesis (preferably lower for smaller particles) varies by composition due to the thermodynamic stability of the DRX phase relative to competing phases. **Higher temperatures are often necessary to stabilize DRX phases with higher Mn content**, irrespective of synthesis method, complicating the production of smaller particles. This underscores the need for further investigation into the thermodynamic stability of DRX compositions to identify conditions that minimize calcination and annealing temperatures while preventing impurity formation or phase decomposition.

We have incorporated these additional discussions in the revised manuscript [**Page 8, lines 5-22**] and included the new data in the revised Supplementary Information [**Supplementary Figs. 6 and 7**].

2. Another important point: Apparently, in this work, the NM-LTMO (200 nm-sized LTMO realized by the modified molten salt method) is still mixed with carbon with a milling process. In this case, frankly I think the advantage of synthesizing a 200 nm-sized LTMO is no longer that significant. This point is extended to the following aspects.

We sincerely appreciate **Reviewer #4's constructive concerns regarding carbon mixing**. Here, we outline our perspectives before addressing the detailed points. **We agree that mechanical carbon mixing may not be ideal for electrode synthesis**. In this regard, **our group is currently investigating direct slurry (ink) mixing of carbon with DRX active materials**. This approach simplifies the entire electrode fabrication process, while avoiding mechanical damages to the as-produced DRX particles. However, direct slurry carbon mixing with DRX particles introduces distinct scientific and engineering challenges that merit a dedicated study. Therefore, in this manuscript, we employed mechanical carbon mixing to a degree that preserves NM-LMTO morphology, as evidenced by SEM images in Figures 1 and 5. Additionally, due to the dispersed, single-particle-like nature of NM-LMTO, the electrode microstructure differs significantly from that of the PS-LMTO electrode after the same mechanical carbon mixing step, resulting in a more homogeneous electrode film. While further investigation is warranted, we believe our demonstration of the NM method highlights potential benefits of using directly synthesized and dispersed DRX particles.

a. **Mixing the DRX powder with carbon using 8000M Mixer/Mill for 2 hours is not that gentle in the first place (compared to non-milling-based carbon mixing). I am wondering if the S-LTMO (i.e. solid-state synthesized LTMO, big micron particle) is applied with the same carbon mixing (8000M Mixer/Mill for 2 hours), how is this performance compared to NM-LTMO?**

We greatly appreciate **Reviewer #4's valuable request for additional tests**. To clarify, we employed a **two-hour shaker mill (8000M Mixer/Mill) for mixing DRX powders with carbon black (CB)** and a **one-hour shaker mill for mixing DRX powders with MWCNT**. To address the reviewer's question regarding performance comparison with NM-LMTO, we conducted additional cycling tests on non-pulverized S-LMTO mixed with CB for two hours and with MWCNT for one hour using the 8000M Mixer/Mill, forming 70:20(CB or MWCNT):10 electrodes.

Response Figures 18a-d present the voltage profiles and capacity retention of the **S-LMTO-CB and S-LMTO-MWCNT electrodes**. In both cases, the initial capacity is lower than that of PS-LMTO-MWCNT reported in the manuscript, which we attribute to the larger particle size of S-LMTO. The capacity retention of S-LMTO-CB is poorer than that of S-LMTO-MWCNT, a trend consistent with the performance observed for NM-LMTO-CB and NM-LMTO-MWCNT. For S-LMTO-CB, there is rapid capacity fading during the initial cycles, followed by a slightly reduced capacity-loss rate up to approximately 30 cycles, after which a faster capacity decay occurs. In contrast, S-LMTO-MWCNT demonstrates more stable capacity retention after the initial fast capacity loss.

Response Figure 18 (a) Voltage profile, **(b)** capacity retention, and coulombic efficiency of S-LMTO electrodes (70:20(CB):10), with S-LMTO and CB mixed for 2 hours using a Spex mill. **(c)** Voltage profile, **(d)** capacity retention, and coulombic efficiency of S-LMTO electrodes (70:20(MWCNT):10), with S-LMTO and MWCNT mixed for 1 hour using a Spex mill. Both cells were cycled between 1.5 and 4.8 V at 20 mA/g. SEM images of **(e)** as-synthesized S-LMTO and **(f)** the S-LMTO/MWCNT mixture.

Our group has previously demonstrated that MWCNT enhances the electrical percolation of DRX electrodes compared to CB and improves the mechanical stability of the electrode (*Energy Environ. Sci.*, 2024, 17, 3753-3764). This explains the more stable capacity retention observed for the S-LMTO-MWCNT electrode compared to S-LMTO-CB. However, **the capacity retention of S-LMTO-MWCNT (~72% over 80 cycles) is still worse than that of NM-LMTO-MWCNT (~88% over 80 cycles; ~85% over 100 cycles)**, underscoring the advantages of using morphology-controlled DRX particles produced via the NM method.

We would also like to note that S-LMTO particles exhibit a degree of cracking after spex carbon mixing, as evidenced by the SEM images in **Response Figures 18e and 18f**. This behavior is due to the very large as-synthesized particle size of S-LMTO, which makes it more susceptible to cracking during the mixing process. This explains the observed ~220 mAh/g capacity from the S-LMTO-MWCNT electrode. Conversely, for NM-LMTO, the particle morphology remains largely unchanged after the same duration of spex mixing. This is because NM-LMTO particles are already small, making further pulverization harder and less effective.

Acknowledging the reviewer's point that ball-milling-based carbon mixing (*e.g.*, planetary mixing or spex milling) are not ideal for preserving particle size or electrode preparation, we have expanded our discussion of this limitation in the revised manuscript [**Page 16, lines 25-29; Page 25, lines 1-10**] and included the new results in the Supplementary Information [**Supplementary Fig. 13**].

b. Key point: if the NM-LTMO is applied with non-milling-based (i.e. mortar-pestle) carbon mixing (i.e. the carbon mixing method for layered cathode and the method that can allow mass production), what the performance of NM-LTMO is like? This data is the key to the highlight of the work.

We greatly appreciate **Reviewer #4's valuable request for non-milling-based carbon mixing tests**. To address the reviewer's suggestion, we **first utilized a direct slurry mixing approach for carbon** (MWCNT and SWCNT) with PS- and NM-LMTO in an 85:10(7.5-MWCNT & 2.5-SWCNT):5 electrode composition. This approach completely avoids the use of ball-milling processes.

Specifically, we used a **commercially available single-walled CNT (SWCNT)-dispersed 1 wt% PVDF in NMP** solution to provide well-dispersed SWCNT (2.5 wt% in the electrode) and **supplemented it with 7.5% MWCNT** to achieve a total carbon content of 10 wt% in the **85:10:5** electrode. The LMTO powder was then added to the MWCNT-SWCNT ink with PVDF as a binder, and the slurry was homogenized using a centrifugal (Thinky) mixer before being cast onto Al foil. Thus, this approach completely avoids the use of ball-milling processes.

Response Figures 19a–d present the voltage profiles and capacity retention of PS-LMTO and NM-LMTO in the directly slurry-mixed 85:10:5 electrodes. Both materials exhibit lower capacities compared to those prepared with MWCNT using spex milling. This outcome is not surprising, as LMTO is known for its low electrical conductivity, necessitating intimate mixing with conductive carbon additives to achieve effective electrical percolation within the composite electrode. **The direct slurry mixing of carbon (CNT) currently appears to limit intimate LMTO/carbon contact**, either due to the inhomogeneous distribution of carbon in the slurry-mixed electrode or the absence of mechanical carbon coating on LMTO, a characteristic often observed in mechanically mixed cathodes. **In this regard, we have been preparing a separate paper and an intellectual property (IP) focused on direct slurry carbon mixing for DRX materials**, as this topic warrants a dedicated electrode-level investigation. Additionally, more advanced direct carbon slurry mixing techniques exist to enhance capacity and retention in DRX materials; however, due to IP constraints, we are unable to discuss these results at this time.

Albeit still a milling-based carbon mixing, we also prepared **NM-LMTO and PS-LMTO mixed with MWCNT using a mild planetary ball mill approach (instead of the spex shaker mill) for 30 minutes at 300 rpm**. While the 300 rpm-30 min planetary ball-milling process may not be as gentle as manual mortar-pestle mixing or direct carbon slurry mixing, this condition represents one of the most gentle ball-mill-based carbon mixing conditions in the literature. We also fabricated this electrode with a reduced amount of MWCNT and a **higher active material content of 90(PS/NM-LMTO):5(MWCNT):5(PVDF)**.

Response Figures 19e-h show the voltage profiles and capacity retention of PS-LMTO and NM-LMTO in the 90:5(MWCNT, 300 rpm-30 min mixing):5 electrode. While the capacity of NM-LMTO or PS-LMTO in this electrode is slightly smaller than those in the spex-milled

70:20(MWCNT):10 electrode, their capacity is better than when they were cycled in direct carbon-slurry-mixed electrodes, suggesting that mechanical carbon coating generally leads to better performance for LMTO (and other DRXs), and its milling intensity or duration do not have to be strong.

Response Figure 19 (a) Voltage profiles, (b) capacity retention, and coulombic efficiency of PS-LMTO in a direct slurry mixed 85:2.5(SWCNT):7.5(MWCNT):5 electrode. (c) Voltage profiles, (d) capacity retention, and coulombic efficiency of NM-LMTO in a direct slurry mixed 85:2.5(SWCNT):7.5(MWCNT):5 electrode. (e) Voltage profiles, (f) capacity retention, and coulombic efficiency of PS-LMTO in a 300 rpm–30 min planetary MWCNT mixed 90:5(MWCNT):5 electrode. (g) Voltage profiles, (h) capacity retention, and coulombic efficiency of NM-LMTO in a 300 rpm–30 min planetary ball-mill MWCNT-mixed 90:5(MWCNT):5 electrode. All the cells were cycled between 1.5–4.8 V at 20 mA/g.

We would like to highlight that this work was supported by a major US battery material company working on commercializing DRX materials, and they do not necessarily see mechanical carbon mixing as a process that must be removed. **Notably, the capacity retention of the NM-LMTO (85.3% over 60 cycles) is still significantly better than the PS-LMTO (46.8% over 60 cycles) in the 90:5(MWCNT, 300 rpm-30 min mixing):5 electrode.**

With the valuable suggestion from the Reviewer #4, we have revised our manuscript [**Page 14, lines 22-27; Page 16, lines 13-17; Page 25, lines 1-10**] and included these additional results in Supplementary Information [**Supplementary Fig. 12**].

3. For the synthesis method, the results showed that lower temperature (800°C) and shorter time (5min) are beneficial to obtaining sub-200 nm primary particles. Could it be more helpful or with similar effect if keep reducing the temperature or shortening the time?

We greatly appreciate **Reviewer #4's insightful question regarding calcination temperature and synthesis duration**. Lowering the temperature or further shortening the synthesis time could indeed be **advantageous**, provided that the **NM method reliably yields a pure DRX phase**.

However, **thermodynamic constraints and synthesis kinetics in DRX formation generally necessitate elevated temperatures** to promote **pure DRX formation over competing phases**. Typically, **higher temperatures stabilize the DRX phase** due to its **high configurational entropy**, which lowers the free energy as temperature increases.

If the temperature is too low, especially in Mn-DRX systems (*e.g.*, Li-Mn-Ti-O, Li-Mn-Nb-O), orthorhombic LiMnO₂ or secondary phases like Li₂TiO₃ or Li₃NbO₄ may form. Therefore, investigation of the **thermodynamic stability of different DRX phases** with respect to the **temperature and DRX compositions** would be needed to identify the **lowest temperature** that the NM method or any other synthesis method can utilize upon forming pure DRX phases with optimal particle morphology. We have expanded on these points in the revised manuscript [**Page 22, lines 12-27**].

4. In Figure 2a, the scale bar is 1 μm, is it contradictory with argument of sub-200 nm primary particles? Please check them.

We sincerely appreciate **Reviewer #4's careful observation of our data**. Indeed, there was an **error in the caption** for Figure 2. The scale bar should read **100 nm rather than 1 μm**. We have revised the figure caption accordingly.

5. In the screen steps for molten salt-flux effect of different salts, the authors argued that “Cs-salts, due to their lower melting points and higher dielectric constants, result in higher LMTO purity than K-salts under the same heating protocol”. Could you please provide more detailed information on the relationship between dielectric constants of the molten-salt and the purity of the resultant product or how dielectric constants of the molten-salt influence the purity of LTMO during synthesis if possible?

We are deeply grateful to **Reviewer #4** for their insightful question regarding the effect of the **dielectric constant of the salt flux**. The **dielectric constant of a molten salt flux** plays a crucial role in **precursor solubility** during molten salt synthesis, primarily by influencing the **solvation environment** and **interactions between ions**:

- **Ion solvation and dissociation:** A higher dielectric constant in a molten salt allows for better solvation of ions, as it can reduce electrostatic interactions between ions in the precursor (*e.g.*, Mn^{3+} and O^{2-} in Mn_2O_3). This enables the precursor compounds to dissociate more effectively into individual ions, increasing their solubility in the molten salt.
- **Precursor-flux interaction:** In molten salt synthesis, the flux often interacts directly with precursor ions. A molten salt with a higher dielectric constant will better stabilize polar or ionic species, promoting solubility of polar precursors.
- **Reaction kinetics and equilibrium:** A suitable dielectric environment provided by the molten salt can stabilize intermediate species, potentially shifting the equilibrium to favor the desired products. This stabilization can facilitate smoother precursor dissolution and interaction, impacting the overall reaction rate and yield of the synthesis.

In summary, molten salts with **higher dielectric constants** generally enhance the solubility of **ionic or polar precursors**, creating a **more stable and dissociative medium** that supports **efficient synthesis reactions**. This enhanced molten-salt synthesis, in turn, allows the DRX synthesis to complete faster under the short molten-salt calcination time (*e.g.*, 5 min) in our experiments. We have elaborated on this briefly in the revised manuscript [**Page 5, lines 12–14**] and in the caption of **Supplementary Figure 1**.

6. For XRD refinement, there are a few issues.

a. In the refinement of all the XRD results, it seems that all the atoms (metal atoms in 4a site and O in 4b site) in every refinement shared the same B_{iso} value of 0.5. Please check it.

We sincerely appreciate **Reviewer #4's critical comments regarding XRD refinement**. In our initial manuscript, we did not refine the B_{iso} values for individual atoms, as these values can be somewhat **arbitrary**. However, to **address the reviewer's concern and minimize potential confusion**, we have **now refined the B_{iso} values in the XRD analysis**.

Our refinement process began with adjusting the atomic occupancies while setting the initial B_{iso} values to 2.5 for Li, 0.5 for transition metals, and 1.5 for oxygen. We then refined the B_{iso} values for the transition metals (*e.g.*, Mn, Ti), ensuring they remained identical and varied together. Subsequently, we refined the B_{iso} values for Li and then for O. **All XRD refinement data in the Supplementary Information have been updated with new B_{iso} values**. Also, we have added clarifications in the B_{iso} refinement process in the **caption of the Supplementary Figure 3**, where the refinement process is first described.

b. Also, (111) diffraction is not shown in Supplementary Fig. 10 and 11.

We sincerely appreciate **Reviewer #4's astute observation**. In response, we have **expanded the angular range** in the XRD plots to include the (111) diffraction peak in **Supplementary Figures 16 and 17** (previously **Supplementary Figs. 10 and 11**).

c. The authors attributed the lattice expansion of both NM- and PS-LMTO to the oxygen loss during cycling, but which was not reflected from the lattice parameters via refinement (e.g., Figure S10f, 11d, and 11e), where O occupancy was still 1. Please comment on this.

We sincerely appreciate **Reviewer #4's valuable comments on oxygen occupancy**. The primary objective of our **XRD refinement** was to determine the **lattice parameters**, which remain **relatively stable in DRX materials**. Since **lattice parameters are minimally influenced by atomic occupancies**, we initially **fixed the O occupancy at 1** during refinement.

To address this uncertainty and respond to the Reviewer's concerns, we have now refined the O occupancies in the XRD analysis. It is important to note that there is ongoing debate in the field regarding the mechanism of oxygen loss in DRX structures. Specifically, it remains unclear whether oxygen loss **leads to the formation of oxygen vacancies** (reducing O occupancy below 1) or **induces cation densification** (maintaining O occupancy at 1 while increasing metal occupancy). In our case, we refined oxygen occupancy after adjusting transition metal occupancies and observed a **greater reduction in O occupancy** for PS-LMTO than NM-LMTO after extended cycling, suggesting a **higher degree of oxygen vacancies** in PS-LMTO.

We now discuss these results in the revised manuscript, emphasizing the **reduced O loss in NM-LMTO** while acknowledging the **potential limitations of this refinement approach** [**Page 20, lines 30–31; Page 21, lines 1–2**].

7. In figure 4d and 4e, the unit or axis values for dQ/dV are probably wrong, please check it.

We are extremely grateful for **Reviewer #4's careful evaluation of our data**. Upon review, we identified an **issue with the y-axis values** in the **dQ/dV plots**, which occurred during **data processing for figure production**.

We are truly grateful to the reviewer for catching this mistake. We have now **updated the plots with the correct values in the revised manuscript (Figs. 4d and 4e)**.

8. Also, for the dQ/dV comparison, the PS-LMTO showed the main discharge peak around 2.4V, while the corresponding peak in NM-LMTO was at ~ 2.7V. It implies that the mechanism is different. More discussions are needed here.

We sincerely appreciate **Reviewer #4's valuable question regarding the redox mechanism observed in the dQ/dV plot**. Indeed, NM-LMTO and PS-LMTO exhibit discharging dQ/dV peaks at different voltages after extended cycling (**Response Figs. 20a and b**).

Response Figure 20 dQ/dV plots of (a) PS-LMTO and (b) NM-LMTO when cycled in the 70:20(MWCNT):10 electrode between 1.5–4.8 V at 20 mA/g. (c) The cyclic voltammogram result for $\text{Li}_{1.3}\text{Mn}_{0.4}\text{Nb}_{0.3}\text{O}_2$, reproduced from Chen *et al.*, “Understanding Performance Degradation in Cation-Disordered Rock-Salt Oxide Cathodes,” *Adv. Energy Mater.* **2019**, 9, 1901255. Used with permission of John Wiley & Sons; permission conveyed through Copyright Clearance Center, Inc. (d) The dQ/dV plot of S-LMTO when cycled in the 70:20(MWCNT):10 electrode between 1.5–4.8 V at 20 mA/g.

Here, we would like to emphasize that the **dQ/dV peak positions are similar** during the **initial cycles**. The position and intensity of these dQ/dV peaks are influenced by both (i) the **redox mechanism** ($\text{Mn}^{2+}/\text{Mn}^{3+}/\text{Mn}^{4+}$ & O-redox) and (ii) the **development of overpotentials** during cycling. The lower-voltage discharge **dQ/dV peak of ~2.4 V for PS-LMTO (compared to ~2.7 V for NM-LMTO) after extended cycling** is most likely attributable to **more severe oxygen loss** in PS-LMTO, which activates greater $\text{Mn}^{3+}/\text{Mn}^{2+}$ reduction (a lower-voltage reaction than $\text{Mn}^{3+}/\text{Mn}^{4+}$ and O-redox) during discharging. Additionally, the exacerbated overpotential evolution in PS-LMTO further decreases the observed discharge voltage peak, as evidenced by the voltage profile comparison. **Similar behavior was previously reported** by D. Chen *et al.* for $\text{Li}_{1.3}\text{Mn}_{0.4}\text{Nb}_{0.3}\text{O}_2$ (**Response Fig. 20c**) using cyclic voltammetry, where they also attributed the reduction in discharge voltage during cycling to **aggravated oxygen loss and overpotential development** (*Adv. Energy Mater.* **2019**, 9, 1901255).

In fact, the dQ/dV evolution of S-LMTO is similar to that of NM-LMTO (Response Fig. 20d), suggesting that the differences observed in PS-LMTO (Pulverized S-LMTO) are not due to intrinsically different redox processes. Instead, the pulverized particle morphology in PS-LMTO likely accelerates degradation of higher-voltage redox processes (*e.g.*, $\text{Mn}^{3+}/\text{Mn}^{4+}$, O-redox) to lower voltage processes (*e.g.*, $\text{Mn}^{2+}/\text{Mn}^{3+}$) following greater oxygen loss, which is commonly observed in O-redox-active Mn-DRX materials.

These additional discussions have been incorporated into the **revised manuscript [Page 17, lines 13–18]**. Furthermore, we have included **soft X-ray Absorption Spectroscopy (XAS) results for NM-LMTO and PS-LMTO [Supplementary Fig. 18; Supplementary Note 1]** to confirm **greater Mn reduction in PS-LMTO after cycling**, consistent with our **interpretation of the dQ/dV results**.

9. By comparing the content of non-lattice oxygen species (OH, C-O, C=O, and $\text{Li}_x\text{PF}_y\text{O}_z$) from XPS, the authors concluded that less CEI developed in NM-LMTO than in PS-LMTO. Considering the almost unchanged OH intensity of PS-LMTO before and after cycling, the relative intensity of C-O, C=O, and $\text{Li}_x\text{PF}_y\text{O}_z$ seems more obvious in NM-LMTO, which seems not supporting this point.

We sincerely appreciate **Reviewer #4’s insightful comments** regarding the analysis of **non-lattice oxygen species (OH, C-O, C=O, and $\text{Li}_x\text{PF}_y\text{O}_z$)** in our XPS data.

To clarify, **the apparent intensities in the initial XPS plots do not fully reflect the quantitative distribution of species**. A more precise assessment is obtained by analyzing the **total area under the curve** rather than individual peak intensities.

O1s peaks	NM-LMTO before Area (CPS·eV)	NM-LMTO 40 cycles Area (CPS·eV)	PS-LMTO before Area (CPS·eV)	PS-LMTO 40 cycles Area (CPS·eV)
$\text{Li}_x\text{PF}_y\text{O}_z$		30194.84		30035.54
OH	49384.49	54909.14	44302.69	89458.59
C-O/C=O	21409.7	64690.02	18568.11	60994.89
Mn-O-Mn	35650.75	2730.7	20873.85	215.29

Response Figure 21 O 1s XPS spectra of (a) the NM-LMTO electrode, (b) the PS-LMTO electrode before cycling and after 40 cycles at 20 mA/g at 25°C within a voltage range of 1.5–4.8 V, and (c) the area ratio of lattice oxygen to oxygenated species in NM-LMTO and PS-LMTO before and after 40 cycles, (d) Areas of oxygen species in NM-LMTO and PS-LMTO before and after 40 cycles. This table displays the total areas under the O1s peaks for $\text{Li}_x\text{PF}_y\text{O}_z$, OH, C-O/C=O, and Mn-O-Mn, comparing the changes in CEI development between NM-LMTO and PS-LMTO.

To improve clarity and support our conclusions, we have updated **Supplementary Figure 19** with a **detailed table of these areas**, as shown in **Response Figure 21**. This updated analysis confirms that the **total area under the curve for non-lattice oxygen species (OH, C–O, C=O, and Li_xPF_yO_z) is greater in PS-LMTO than in NM-LMTO**, with a **particularly higher OH content in PS-LMTO**, reinforcing our conclusion of more **extensive CEI development in PS-LMTO**. Furthermore, the **negligible lattice oxygen signal in PS-LMTO after cycling** (compared to non-lattice oxygen species) suggests **thicker CEI layers**, even though the relative proportions of OH, C–O, C=O, and Li_xPF_yO_z may differ from those in NM-LMTO after cycling.

10. In Supplementary Fig. 7, if I understand it correctly, the volume equivalent to Li₂CO₃ is should be (V₂– V₁), instead of 2(V₂-V₁) right? (as one carbonate consumes two H⁺).

We sincerely appreciate **Reviewer #4's insightful question** regarding the **volume equivalent to Li₂CO₃** in **Supplementary Fig. 7**. Below, we outline our **titration process and the stoichiometric calculations** that underpin our conclusions.

Our **pH titration curve** exhibits **two distinct abrupt changes**, corresponding to key reaction milestones:

1. **First Abrupt Change:** Occurring at volume V₁, this point signifies the complete neutralization of all hydroxide ions (OH⁻) and half of the carbonate ions (CO₃²⁻).

The reactions involved are:

At this point, the volume of HCl added (V₁) neutralizes all the OH⁻ and half of the carbonate ions, given the stoichiometric ratio of Li to CO₃²⁻ in Li₂CO₃ is 2:1. Thus, the HCl volume (V₁) for this stage is given by:

$$V_1 = [\text{OH}^-] + \frac{1}{2} [\text{CO}_3^{2-}] \quad (1)$$

2. **Second Abrupt Change:** This change is identified at volume V₂ and marks the complete neutralization of all hydroxide (OH⁻) and carbonate ions (CO₃²⁻) present in the solution. At this stage, the titration continues in the same flask, allowing the HCl to react with the remaining half of the carbonate ions. Consequently, the total reaction that occurs in the flask can be summarized as the complete conversion of both hydroxide and carbonate to their respective neutralized products.

This comprehensive neutralization confirms that V₂ represents the total volume of HCl required to fully neutralize all reactive species in the solution.

$$V_2 = [\text{OH}^-] + [\text{CO}_3^{2-}] \quad (2)$$

Solving equations (1) and (2) simultaneously, we obtain:

Volume of HCl Equivalent to CO_3^{2-} : $2(V_2 - V_1)$

Volume of HCl Equivalent to OH^- : $2V_1 - V_2$

These calculations align with our experimental observations and the stoichiometric expectations of the reactions involved.

Response to Reviewer #2

Still some more data are not clear in this stage. The reviewer still sees insufficient facility time for the measurement, out of focus for the full cell test, and failure in measurement using three-electrode cells, Therefore, revision is requested to clear those mentioned points in the revision.

We thank Reviewer #2 for the constructive feedback. To address the concerns, we have performed substantial additional experiments, including three-electrode EIS and full-cell testing, which we believe have strengthened the manuscript.

Response Figure 1 (a) Nyquist plots of LMTO/Li half-cells before cycling and after 100 cycles (**Supplementary Fig. 22**). (b) Nyquist plots of cathode/cathode symmetric cells constructed from LMTO/Li half-cells before cycling and after 10 cycles (**Supplementary Figs. 23a-e**).

Regarding the impedance measurements, we previously presented EIS results for both half-cells and symmetric cells (**Response Fig. 1**), which demonstrated that the **faster capacity fade observed in PS-LMTO||Li cells** compared to NM-LMTO||Li cells is accompanied by **significantly greater impedance growth on the cathode side** in the PS-LMTO configuration, although the contribution from the Li-metal anode cannot be neglected.

To further substantiate this observation, we performed **three-electrode EIS measurements** on NM-LMTO and PS-LMTO coin cells. In these experiments, cathodes composed of 70 wt% NM/PS-LMTO, 20 wt% MWCNT, and 10 wt% PVDF were used as the working electrodes, with a Li-deposited Cu mesh as the reference electrode (RE) and a Li-metal disk as the counter electrode (CE). The three-electrode coin cells were assembled following the protocol reported by Cheuh et al. [*Adv. Energy Mater.* 2022, 12, 2201114], as also detailed in the Methods section of the revised manuscript.

Response Figure 2 (a,f) Equivalent circuit models used to fit the Nyquist plots measured (a) between the cathode (NM/PS-LMTO electrode) and the reference electrode (Li-deposited Cu mesh), and (f) between the Li-metal anode and the reference electrode, in three-electrode EIS measurements. (b-e) Nyquist plots and corresponding fits measured between the cathode and the reference electrode: (b,c) before cycling and (d,e) after 10 cycles between 1.5–4.8 V at 20 mA/g. (g-j) Nyquist plots and corresponding fits measured between the Li-metal anode and the reference electrode: (g,h) before cycling and (i,j) after 10 cycles under the same conditions.

Response Figure 2 shows the results of the three electrode EIS measurements. Nyquist plots measured between the cathode and the reference electrode (RE) indicate that initial cathode impedance is lower for the NM-LMTO electrode ($R_{\text{film}} + R_{\text{ct(cathode)}} \approx 57.3 \Omega$) compared to the PS-LMTO electrode ($R_{\text{film}} + R_{\text{ct(cathode)}} \approx 77.6 \Omega$) (**Response Figs. 2b and 2c**). Notably, after 10 cycles between 1.5–4.8 V at 20 mA/g, the total resistance increases substantially to $\approx 531.7 \Omega$ for the NM-LMTO cathode and $\approx 2247.9 \Omega$ for the PS-LMTO cathode (**Response Figs. 2d and 2e**), highlighting **significantly more pronounced impedance growth for the PS-LMTO cathode**.

For the Li-metal anode, the impedance measured against the RE decreases after 10 cycles in both NM-LMTO||Li and PS-LMTO||Li cells (**Response Figs. 2i and 2j**), likely due to the electrochemical decomposition of the passivation layer on the Li-metal surface. This trend is consistent with the Li||Li symmetric cell EIS results in Supplementary Figure 23. Notably, this does not contradict the impedance increase observed at the Li-metal anode side in Supplementary Figure 22, as the symmetric and three-electrode tests reflect only 10 cycles, whereas Supplementary Figure 22 presents data after 100 cycles. Over extended cycling, Li dendrite formation and surface degradation can substantially increase anode impedance, which is captured in the equivalent circuit fitting shown in Supplementary Figure 22 in the revised SI.

These three-electrode EIS results corroborate the observations from symmetric cells and LMTO||Li half-cells, confirming that **the faster capacity fade in PS-LMTO cells** primarily stems from significantly **accelerated degradation of the PS-LMTO cathode** compared to NM-LMTO. Nonetheless, degradation of the Li-metal anode remains a contributing factor, particularly over long-term cycling. The three-electrode EIS results have now been included as **Supplementary Figure 24** and **Supplementary Note 3**.

Response Figure 3 (a–c) Voltage profiles of (a) NM-LMTO [90:5(MWCNT):5, 1.5–4.8 V, 20 mA/g], (b) PS-LMTO [90:5(MWCNT):5, 1.5–4.8 V, 20 mA/g], and (c) graphite [93.2:2.5(CB):4.3, 0.01–1.5 V, 20 mA/g], all measured in half-cell configuration vs. Li metal.

To further address the Reviewer’s suggestions, we fabricated full-cell coin cells using a 90(NM/PS-LMTO):5(MWCNT):5(PVDF) cathode and a 93.2(Graphite):2.5(CB):4.3(PVDF) anode. **Response Figures 3a and 3b** show the half-cell voltage profiles of NM-LMTO and PS-LMTO, respectively, using the 90:5:5 electrode composition cycled between 1.5–4.8 V at 20 mA/g. **Response Figure 3c** presents the representative initial 5-cycle voltage profile of the graphite anode (AA40798TC, Alfa Aesar) with the 93.2:2.5:4.3 composition, cycled in a half-cell between 0.01–1.5 V at 20 mA/g at room temperature. The graphite’s initial discharge capacity reaches ~347 mAh/g with a subsequent charging capacity of ~280 mAh/g, showing slightly inferior capacities to what would be expected from an optimized graphite material (theoretical capacity of 372 mAh/g).

For full-cell construction, the anode (N) and cathode (P) capacities were balanced to achieve an N/P ratio of approximately 1.2. This was based on the first discharge specific capacity of ~350 mAh/g for graphite (**Response Fig. 3c**), the first charge capacity of ~270 mAh/g for NM-LMTO (**Response Fig. 3a**), and ~380 mAh/g for PS-LMTO (**Response Fig. 3b**), each measured at a current density of 20 mA/g in half-cell configurations. The areal loading of the PS/NM-LMTO was approximately 4 mg_{LMTO}/cm², and the graphite anode loading was accordingly adjusted to maintain N/P ≈ 1.2 in both NM-LMTO and PS-LMTO full cells.

Response Figures 4a and 4b present the initial 10-cycle voltage profiles of the NM-LMTO and PS-LMTO full-cells, respectively, cycled at 20 mA/g_{LMTO} between 1.0–4.7 V. In the full-cell configuration, the first discharge capacity of NM-LMTO in the 90:5:5 electrode is approximately 182 mAh/g_{LMTO} (**Response Fig. 4a**), which is slightly lower but comparable to the initial discharge capacity (~195 mAh/g_{LMTO}) observed in the half-cell setup using the same

electrode composition (**Response Fig. 3a**). We note that the specific capacity of a cathode material in a full-cell is influenced not only by the typical factors such as cycling rate and voltage window (parameters that also apply to half-cells) but also by the anode material and the N/P ratio, both of which can affect the overall voltage profile and accessible capacity in a full-cell. Accordingly, NM-LMTO's capacity in the full-cell configuration may be further improved through optimization of these full-cell parameters.

Response Figure 4 (a,b) Initial 10-cycle voltage profiles of full-cells: (a) NM-LMTO(90:5-MWCNT:5)||graphite and (b) PS-LMTO(90:5-MWCNT:5)||graphite, cycled at 20 mA/g_{LMTO} between 1.0–4.7 V at 25°C. (c,d) Specific capacity (mAh/g_{LMTO}) retention and coulombic efficiency of (c) NM-LMTO and (d) PS-LMTO full-cells, cycled at 20 mA/g_{LMTO} for the initial 10 cycles, 40 mA/g_{LMTO} for the next 3 cycles, 100 mA/g_{LMTO} for the next 10 cycles, and 200 mA/g_{LMTO} for the final 10 cycles, within a voltage window of 1.0–4.7 V at 25°C. (e,f) Corresponding energy density (Wh/kg_{LMTO}) retention of (e) NM-LMTO and (f) PS-LMTO in the full-cells.

The PS-LMTO electrode shows higher initial capacities than NM-LMTO in the full-cell (**Response Fig. 4b**) as in the half-cell, yet its initial discharge capacity (~250 mAh/g_{LMTO}) is quite lower than the value obtained in the half-cell (~320 mAh/g_{LMTO}). This discrepancy can be attributed in part to lithium loss during the first cycle due to irreversible lithium consumption at

the graphite anode side. As shown in **Response Fig. 3c**, our graphite anode exhibits significant initial lithium trapping upon lithiation (discharge in a half-cell, charge in a full-cell), likely due to SEI formation and other side reactions. This initial lithium loss reduces the lithium inventory available for reinsertion into the PS-LMTO during the first discharge in the full-cell, thereby limiting its capacity. In contrast, NM-LMTO already exhibits low initial coulombic efficiency in the half-cell (*i.e.*, a large difference between charge and discharge capacities), suggesting that its full-cell capacity is less sensitive to lithium loss at the graphite side during the initial cycles.

The capacity retention of NM-LMTO (~94.8%) over the first 10 cycles in full-cells is noticeably better than that of PS-LMTO (~77.1%) (**Response Figs. 4a-d**), consistent with trends observed in half-cell testing. As the cycling rate increases from 20 to 40, 100, and 200 mA/g-LMTO, the discharge capacity of NM-LMTO gradually decreases to approximately 156, 120, and 77 mAh/g-LMTO, respectively (**Response Fig. 4c**). In contrast, the capacity of PS-LMTO drops to approximately 171, 110, and 38 mAh/g-LMTO under the same conditions (**Response Fig. 4d**), but its capacity retention is inferior to that of NM-LMTO in all rates. The corresponding energy densities (Wh/kg-LMTO) for NM-LMTO and PS-LMTO obtained from the full-cell tests are presented in **Response Figures 4e and 4f**. Despite having a lower initial energy density, NM-LMTO surpasses PS-LMTO in later cycles due to its enhanced cycling stability.

Alongside complementary data, including EIS measurements and structural characterizations from XRD, XPS, XAS, and SEM, these full-cell results further support the superior cycling stability of NM-LMTO compared to PS-LMTO, despite its lower initial capacity.

We thank the Reviewer for requesting full-cell demonstrations, which have broadened the scope of our study. However, we would like to clarify that the primary objective of this work is to introduce a broadly applicable synthesis strategy for various DRX compounds (*e.g.*, $\text{Li}_{1.2}\text{Mn}_{0.4}\text{Ti}_{0.4}\text{O}_2$, $\text{Li}_{1.2}\text{Mn}_{0.6}\text{Nb}_{0.2}\text{O}_2$, $\text{Li}_{1.1}\text{Mn}_{0.7}\text{Ti}_{0.2}\text{O}_2$, and $\text{Li}_{1.2}\text{Ni}_{0.2}\text{Ti}_{0.6}\text{O}_2$), enabling the formation of well-controlled, small particles without compromising structural integrity. LMTO is presented here as a representative system to demonstrate the effectiveness of this method, rather than to benchmark full-cell performance. Realizing such benchmarking would require extensive optimization, particularly of the anode, N/P ratio, and electrolyte, to fully unlock the material's potential in practical configurations. Nonetheless, the included full-cell results serve as proof of concept and are now discussed in the revised manuscript [**Page 16, lines 19–21**], and presented in **Supplementary Figure 15** and **Supplementary Note 1**.

Response to Reviewer #3

The authors have addressed the reviewer's comments effectively. The manuscript has shown significant improvement, and I recommend it for publication.

We sincerely thank Reviewer #3 for the positive feedback and recommendation for publication.

Response to Reviewer #4

The authors have addressed my comments comprehensively. Additional compositions were synthesized based on the method to support the broader applicability of the synthetic approach. Also, more comparative electrochemical measurements based on different carbon coating and carbon mixing parameters were applied to justify the synthesis of 200 nm particles. The other small errors, e.g. labeling and units, etc., were also corrected. For me, the paper is good to go.

We thank Reviewer #4 for the supportive comments and are grateful for the recommendation for publication.